# Are Graph Neural Networks Optimal Approximation Algorithms?

**Morris Yau**
MIT CSAIL
morrisy@mit.edu

**Nikolaos Karalias**
MIT CSAIL
stalence@mit.edu

**Eric Lu**
Harvard University
ericlu01@g.harvard.edu

**Jessica Xu**
Independent researcher formerly at MIT
jessica.peng.xu@gmail.com

**Stefanie Jegelka**
TUM[*] and MIT[†]
stefje@mit.edu

## Abstract

In this work we design graph neural network architectures that capture optimal approximation algorithms for a large class of combinatorial optimization problems, using powerful algorithmic tools from semidefinite programming (SDP). Concretely, we prove that polynomial-sized message-passing GNN's can learn the most powerful polynomial time algorithms for Max Constraint Satisfaction Problems assuming the Unique Games Conjecture. We leverage this result to construct efficient graph neural network architectures, OptGNN, that obtain high-quality approximate solutions on landmark combinatorial optimization problems such as Max-Cut, Min-Vertex-Cover, and Max-3-SAT. Our approach achieves strong empirical results across a wide range of real-world and synthetic datasets against solvers and neural baselines. Finally, we take advantage of OptGNN's ability to capture convex relaxations to design an algorithm for producing bounds on the optimal solution from the learned embeddings of OptGNN.

## 1 Introduction

Combinatorial Optimization (CO) is the class of problems that optimize functions subject to constraints over discrete search spaces. They are often NP-hard to solve and to approximate, owing to their typically exponential search spaces over nonconvex domains. Nevertheless, their important applications in science and engineering (Gardiner et al., 2000; Zaki et al., 1997; Smith et al., 2004; Du et al., 2017) has engendered a long history of study rooted in the following simple insight. In practice, CO instances are endowed with domain-specific structure that can be exploited by specialized algorithms (Hespe et al., 2020; Walteros & Buchanan, 2019; Ganesh & Vardi, 2020). In this context, neural networks are natural candidates for learning and then exploiting patterns in the data distribution over CO instances.

The emerging field at the intersection of machine learning (ML) and combinatorial optimization (CO) has led to novel algorithms with promising empirical results for several CO problems. However, similar to classical approaches to CO, ML pipelines have to manage a tradeoff between efficiency and optimality. Indeed, prominent works in this line of research forego optimality and focus on parametrizing heuristics (Li et al., 2018; Khalil et al., 2017; Yolcu & Póczos, 2019; Chen & Tian, 2019) or by employing specialized models (Zhang et al., 2023; Nazari et al., 2018; Toenshoff et al., 2019; Xu et al., 2021; Min et al., 2022) and task-specific loss functions (Amizadeh et al., 2018;

---
[*]CIT, MCML, MDSI
[†]EECS and CSAIL

38th Conference on Neural Information Processing Systems (NeurIPS 2024).

Karalias & Loukas, 2020; Wang et al., 2022; Karalias et al., 2022; Sun et al., 2022). Exact ML solvers that can guarantee optimality often leverage general techniques like branch and bound (Gasse et al., 2019; Paulus et al., 2022) and constraint programming (Parjadis et al., 2021; Cappart et al., 2019), which offer the additional benefit of providing approximate solutions together with a bound on the distance to the optimal solution. The downside of those methods is their exponential worst-case time complexity. Striking the right balance between efficiency and optimality is quite challenging, which leads us to the central question of this paper:

*Are there neural architectures for efficient combinatorial optimization that can learn to adapt to a data distribution over instances yet capture algorithms with **optimal** worst-case approximation guarantees?*

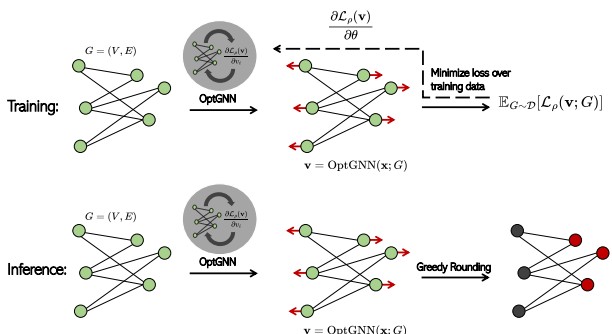

Figure 1: Schematic representation of OptGNN. During training, OptGNN produces node embeddings $\mathbf{v}$ using message passing updates on the graph $G$. These embeddings are used to compute the penalized objective $\mathcal{L}_p(\mathbf{v}; G)$. OptGNN is trained by minimizing the average loss over the training set. At inference time, the fractional solutions (embeddings) $\mathbf{v}$ for an input graph $G$ produced by OptGNN are rounded using randomized rounding.

To answer this question, we build on the extensive literature on approximation algorithms and semidefinite programming (SDP) which has led to breakthrough results for NP-hard combinatorial problems, such as the Goemans-Williamson approximation algorithm for Max-Cut (Goemans & Williamson, 1995) and the use of the Lovász theta function to find the maximum independent set on perfect graphs (Lovász, 1979; Grötschel et al., 1981). For several problems, it is known that if the Unique Games Conjecture (UGC) is true, then the approximation guarantees obtained through SDP relaxations are indeed the best that can be achieved (Raghavendra, 2008; Barak & Steurer, 2014). The key insight of our work is that a polynomial time message-passing algorithm (see Definition G) approximates the solution of an SDP with the optimal integrality gap for the class of Maximum Constraint Satisfaction Problems (Max-CSP), assuming the UGC. This algorithm can be naturally parameterized to build a graph neural network, which we call OptGNN.

Our contributions can be summarized as follows:

- We construct a polynomial-time message passing algorithm (see Definition G) for solving the SDP of Raghavendra (2008) for the broad class of maximum constraint satisfaction problems (including Max-Cut, Max-SAT, etc.), that is optimal barring the possibility of significant breakthroughs in the field of approximation algorithms.

- We construct a graph neural network architecture which we call OptGNN, to capture this message-passing algorithm. We show that OptGNN is PAC-learnable with polynomial sample complexity.

- We describe how to generate optimality certificates from the learned representations of OptGNN i.e., provable bounds on the optimal solution.

- Empirically, OptGNN is simple to implement (see pseudocode in Appendix 5) and we show that it achieves competitive results on 3 landmark CO problems and several datasets against classical heuristics, solvers and state-of-the-art neural baselines [3]. Furthermore, we provide out-of-distribution (OOD) evaluations and ablation studies for OptGNN that further validate our parametrized message-passing approach.

---

[3] Code is available at: `https://github.com/penlu/bespoke-gnn4do`.

## 2 Background and Related Work

**Optimal Approximation Algorithms.** Proving that an algorithm achieves the best approximation guarantee for NP-hard problems is an enormous scientific challenge as it requires ruling out that better algorithms exist (i.e., a hardness result). The Unique Games Conjecture (UGC) (Khot, 2002) is a striking development in the theory of approximation algorithms because it addresses this obstacle. If true, it implies approximation hardness results for a large number of hard combinatorial problems that match the approximation guarantees of the best-known algorithms (Raghavendra & Steurer, 2009b; Raghavendra et al., 2012). For that reason, those algorithms are also sometimes referred to as UGC-optimal. More importantly, the UGC implies that for all Max-CSPs there is a general UGC-optimal algorithm based on semidefinite programming (Raghavendra, 2008). For a complete exposition on the topic of UGC and approximation algorithms, we refer the reader to Barak & Steurer (2014).

**Neural approximation algorithms and their limitations.** There has been important progress in characterizing the combinatorial capabilities of modern deep learning architectures, including bounds on the approximation guarantees achievable by GNNs on bounded degree graphs (Sato et al., 2019, 2021) and conditional impossibility results for solving classic combinatorial problems such as Max-Independent-Set and Min-Spanning-Tree (Loukas, 2019). It has been shown that a GNN can implement a distributed local algorithm that straightforwardly obtains a 1/2-approximation for Max-SAT (Liu et al., 2021), which is also achievable through a simple randomized algorithm (Johnson, 1973). Recent work proves there are barriers to the approximation power of GNNs for combinatorial problems including Max-Cut and Min-Vertex-Cover (Gamarnik, 2023) for constant depth GNNs. Other approaches to obtaining approximation guarantees propose avoiding the dependence of the model on the size of the instance with a divide-and-conquer strategy; the problem is subdivided into smaller problems which are then solved with a neural network (McCarty et al., 2021; Kahng et al., 2023).

**Convex Relaxations and Machine Learning.** Convex relaxations are crucial in the design of approximation algorithms. In this work, we show how SDP-based approximation algorithms can be incorporated into the architecture of neural networks. We draw inspiration from the algorithms that are used for solving low-rank SDPs (Burer & Monteiro, 2003; Wang et al., 2017; Wang & Kolter, 2019; Boumal et al., 2020). Beyond approximation algorithms, there is work on designing differentiable Max-SAT solvers via SDPs to facilitate symbolic reasoning in neural architectures (Wang et al., 2019). This approach uses a fixed algorithm for solving a novel SDP relaxation for Max-SAT, and aims to learn the structure of the SAT instance. In our case, the instance is given, but our algorithm is learnable, and we seek to predict the solution. SDPs have found numerous applications in machine learning including neural network verification (Brown et al., 2022), differentiable learning with discrete functions (Karalias et al., 2022), kernel methods (Rudi et al., 2020; Jethava et al., 2013) and quantum information tasks (Kriváchy et al., 2021). Convex relaxation are also instrumental in integer programming where branch-and-bound tree search is guaranteed to terminate with optimal integral solutions to Mixed Integer Linear Programs (MILP). Proposals for incorporating neural networks into the MILP pipeline include providing a "warm start" (Benidis et al., 2023) to the solver, learning branching heuristics (Gasse et al., 2019; Nair et al., 2020; Gupta et al., 2020; Paulus et al., 2022), and learning cutting plane protocols (Paulus et al., 2022). A recent line of work studies the capabilities of neural networks to solve linear programs (Chen et al., 2022; Qian et al., 2023). It is shown that GNNs can represent LP solvers, which may in turn explain the success of learning branching heuristics (Qian et al., 2023). In a similar line of work, neural nets are used to learn branching heuristics for CDCL SAT solvers (Selsam & Bjørner, 2019; Kurin et al., 2020; Wang et al., 2021). Finally, convex optimization has also found applications (Numeroso et al., 2023) in the neural algorithmic reasoning paradigm (Veličković et al., 2022) where neural networks are trained to solve problems by learning to emulate discrete algorithms in higher dimensional spaces.

**Learning frameworks for CO.** A common approach to neural CO is to use supervision either in the form of execution traces of expert algorithms or labeled solutions (Li et al., 2018; Selsam et al., 2018; Prates et al., 2019; Vinyals et al., 2015; Joshi et al., 2019, 2020; Gasse et al., 2019; Ibarz et al., 2022; Georgiev et al., 2023). Obtaining labels for combinatorial problems can be computationally costly which has led to the development of neural network pipelines that can be trained without access to labels or partial solutions. This includes approaches based on Reinforcement Learning (Ahn et al., 2020; Böther et al., 2022; Barrett et al., 2020, 2022; Bello et al., 2016; Khalil et al., 2017; Yolcu &

Póczos, 2019; Chen & Tian, 2019), and other self-supervised methods (Brusca et al., 2023; Karalias et al., 2022; Karalias & Loukas, 2020; Tönshoff et al., 2022; Schuetz et al., 2022a,b; Amizadeh et al., 2019; Dai et al., 2020; Sun et al., 2022; Wang et al., 2022; Amizadeh et al., 2018; Gaile et al., 2022). Our work falls into the latter category since only the problem instance is sufficient for training and supervision signals are not required. For a complete overview of the field, we refer the reader to the relevant survey papers (Cappart et al., 2023; Bengio et al., 2021).

# 3 Optimal Approximation Algorithms with Neural Networks

We begin by showing that solving the Max-Cut problem using a vector (low-rank SDP) relaxation and a simple projected gradient descent scheme amounts to executing a message-passing algorithm on the input graph. We then generalize this insight to the entire class of Max-CSPs. We reformulate the UGC-optimal SDP for Max-CSP in SDP 1. Our main Theorem 3.1 exhibits a message passing algorithm (Algorithm 1) for solving SDP 1. We then capture Algorithm 1 via a message passing GNN with learnable weights (see Definition G for definition of Message Passing GNN). Thus, by construction OptGNN captures algorithms with UGC-optimal approximation guarantees for Max-CSP. Furthermore, we prove that OptGNN is efficiently PAC-learnable (see Lemma 3.1) as a step towards explaining its empirical performance.

## 3.1 Solving Combinatorial Optimization Problems with Message Passing

In the Max-Cut problem, we are given a graph $G = (V, E)$ with $N$ vertices $V$ and edges $E$. The goal is to divide the vertices into two sets that maximize the number of edges going between them. This corresponds to the quadratic integer program

$$\max_{x_1, x_2, \ldots, x_N} \sum_{(i,j) \in E} \frac{1}{2}(1 - x_i x_j) \quad \text{subject to:} \quad x_i^2 = 1 \quad \forall i \in [N].$$

The global optimum of the integer program is the Max-Cut. Noting that discrete variables are not amenable to the tools of continuous optimization, a standard technique is to 'lift' the quadratic integer problem: replace the integer variables $x_i$ with vectors $v_i \in \mathbb{R}^r$ and constrain $v_i$ to lie on the unit sphere

$$\min_{v_1, v_2, \ldots, v_N} \quad -\sum_{(i,j) \in E} \frac{1}{2}(1 - \langle v_i, v_j \rangle) \quad \text{subject to:} \quad \|v_i\| = 1 \quad \forall i \in [N] \quad v_i \in \mathbb{R}^r. \quad (1)$$

This nonconvex relaxation of the problem admits an efficient algorithm Burer & Monteiro (2003). Furthermore, all local minima are approximately global minima (Ge et al., 2016) and variations of stochastic gradient descent converge to its optimum (Bhojanapalli et al., 2018; Jin et al., 2017) under a variety of smoothness and compactness assumptions. Specifically, for large enough $r$ (Boumal et al., 2020; O'Carroll et al., 2022), simple algorithms such as block coordinate descent (Erdogdu et al., 2019) can find an approximate global optimum of the objective. Projected gradient descent is a natural approach for solving the minimization problem in equation 1. In iteration $t$ (for $T$ iterations), update vector $v_i$ as

$$v_i^{t+1} = \text{NORMALIZE}\left(v_i^t - \eta \sum_{j \in N(i)} v_j^t\right), \quad (2)$$

where NORMALIZE enforces unit Euclidean norm, $\eta \in \mathbb{R}^+$ is an adjustable step size, and $N(i)$ the neighborhood of node $i$. The gradient updates to the vectors are local, each vector is updated by aggregating information from its neighboring vectors (i.e., it is a message-passing algorithm).

**OptGNN for Max-Cut.** Our main contribution in this paper builds on the following observation. We may generalize the dynamics described above by considering an overparametrized version of the gradient descent updates in equations 2. Let $M_1, M_2, \ldots, M_T \in \mathbb{R}^{r \times 2r}$ be a set of $T$ learnable matrices corresponding to $T$ layers of a neural network. Then for layer $t$ and embedding $v_i$ we define a GNN update

$$v_i^{t+1} := \text{NORMALIZE}\left(M_t\left(\begin{bmatrix} v_i^t \\ \sum_{j \in N(i)} v_j^t \end{bmatrix}\right)\right). \quad (3)$$

More generally, we can write the dynamics as $v_i^{t+1} := \text{NORMALIZE}(M_t(\text{AGG}(v_i^t, \{v_j^t\}_{j \in N(i)})))$, where AGG is a function of the node embedding and its neighboring embeddings. We will present a message passing algorithm 1 that generalizes the dynamics of 2 to the entire class of Max-CSPs (see example derivations in Appendix A and Appendix B), which provably converges in polynomial iterations for a reformulation of the canonical SDP relaxation of Raghavendra (2008) (see SDP 1). Parametrizing this general message-passing algorithm will lead to the definition of OptGNN (see Definition 3.3).

## 3.2 Message Passing for Max-CSPs

Given a set of constraints over variables, Max-CSP asks to find a variable assignment that maximizes the number of satisfied constraints. Formally, a Constraint Satisfaction Problem $\Lambda = (\mathcal{V}, \mathcal{P}, q)$ consists of a set of $N$ variables $\mathcal{V} := \{x_i\}_{i \in [N]}$ each taking values in an alphabet $[q]$ and a set of predicates $\mathcal{P} := \{P_z\}_{z \subset \mathcal{V}}$ where each predicate is a payoff function over $k$ variables $X_z = \{x_{i_1}, x_{i_2}, ..., x_{i_k}\}$. Here we refer to $k$ as the arity of the Max-k-CSP. We adopt the normalization that each predicate $P_z$ returns outputs in $[0, 1]$. We index each predicate $P_z$ by its domain $z$. The goal of Max-k-CSP is to maximize the payoff of the predicates.

$$\text{OPT}(\Lambda) := \max_{(x_1,...,x_N) \in [q]^N} \frac{1}{|\mathcal{P}|} \sum_{P_z \in \mathcal{P}} P_z(X_z), \tag{4}$$

where we normalize by the number of constraints so that the total payoff is in $[0, 1]$. Therefore we can unambiguously define an $\epsilon$-approximate assignment as an assignment achieving a payoff of OPT $- \epsilon$. Since our result depends on a message-passing algorithm, we will need to define an appropriate graph structure over which messages will be propagated. To that end, we will leverage the constraint graph of the CSP instance: Given a Max-k-CSP instance $\Lambda = (\mathcal{V}, \mathcal{P}, q)$ a *constraint graph* $G_\Lambda = (V, E)$ is comprised of vertices $V = \{v_{\phi,\zeta}\}$ for every subset of variables $\phi \subseteq z$ for every predicate $P_z \in \mathcal{P}$ and every assignment $\zeta \in [q]^{|z|}$ to the variables in $z$. The edges $E$ are between any pair of vectors $v_{\phi,\zeta}$ and $v_{\phi',\zeta'}$ such that the variables in $\phi$ and $\phi'$ appear in a predicate together. For instance, for a SAT clause $(x_1 \vee x_2) \wedge x_1 \wedge x_3$ there are four nodes $v_1, v_{12}, v_3$ and $v_\emptyset$ with a complete graph between $\{v_1, v_{12}, v_\emptyset\}$ and $v_3$ an isolated node.

Let SDP$(\Lambda)$ be the optimal value of the SDP 1 on instance $\Lambda$. The *approximation ratio* for the Max-k-CSP problem achieved by the SDP 1 is $\min_{\Lambda \in \text{Max-k-CSP}} \frac{\text{OPT}(\Lambda)}{\text{SDP}(\Lambda)}$, where the minimization is taken over all instances $\Lambda$ with arity $k$. There is no polynomial time algorithm that can achieve a larger approximation ratio assuming the truth of the UGC Raghavendra (2008). We construct our message passing algorithm as follows. First we introduce the definition of the vector form SDP and its associated quadratically penalized Lagrangian.

**Definition** (Quadratically Penalized Lagrangian). Any standard form SDP $\Lambda$ can be expressed as the following vector form SDP for some matrix $V = [v_1, v_2, \ldots, v_N] \in R^{N \times N}$.

$$\min_{V \in \mathbb{R}^{N \times N}} \quad \langle C, V^T V \rangle \quad \text{subject to:} \quad \langle A_i, V^T V \rangle = b_i \quad \forall i \in [\mathcal{F}]. \tag{5}$$

For any SDP in vector form we define the $\rho$ quadratically penalized Lagrangian to be

$$\mathcal{L}_\rho(V) := \langle C, V^T V \rangle + \rho \sum_{i \in \mathcal{F}} \left( \langle A_i, V^T V \rangle - b_i \right)^2. \tag{6}$$

We show that gradient descent on this Lagrangian $\mathcal{L}_\rho(V)$ for the Max-CSP SDP 1 takes the form of a message-passing algorithm on the constraint graph that can provably converge to an approximate global optimum for the SDP (see algorithm 1). We see that gradient descent on $\mathcal{L}_\rho$ takes the form of a simultaneous message passing update on the constraint graph. See equation 60 and algorithm 3 for analytic form of the Max-CSP message passing update. See appendix A and B for analytic form of Min-Vertex-Cover and Max-3-SAT message passing updates. Our main theorem is then following.

**Theorem 3.1.** *[Informal] Given a Max-k-CSP instance $\Lambda$ represented in space $\Phi = O(|\mathcal{P}|q^k)$, there is a message passing Algorithm 3 on constraint graph $G_\Lambda$ with a per iteration update time of $O(\Phi)$ that computes in $O(\epsilon^{-4}\Phi^4)$ iterations an $\epsilon$-approximate solution (solution satisfies constraints to error $\epsilon$ achieving objective value within $\epsilon$ of optimum) to SDP 1. For the formal theorem and proof see Theorem C.1.*

### 3.3 OptGNN for Max-CSP

Next we define OptGNN for Max-CSP. See A and B for OptGNN for Vertex Cover and 3-SAT.

**Definition** (OptGNN for Max-CSP). Let $\Lambda$ be a Max-CSP instance on a constraint graph $G_\Lambda$ with $N$ nodes. Let $U$ be an input matrix of dimension $r \times N$ for $N$ nodes with embedding dimension $r$. Let $\mathcal{L}_\rho$ be the penalized lagrangian loss defined as in equation 6 associated with the Max-CSP instance $\Lambda$. Let $M$ be the OptGNN weights which are a set of matrices $M := \{M_1, M_2, ..., M_T\} \in \mathbb{R}^{r \times 2r}$. Let $\text{LAYER}_{M_i} : \mathbb{R}^{r \times N} \to \mathbb{R}^{r \times N}$ be the function $\text{LAYER}_{M_i}(U) = M_i(\text{AGG}(U))$, where $\text{AGG} : \mathbb{R}^{r \times N} \to \mathbb{R}^{2r \times N}$ is the aggregation function $\text{AGG}(U) := [U, \nabla\mathcal{L}_\rho(U)]$. We define $\text{OptGNN}_{(M,\Lambda)} : \mathbb{R}^{r \times N} \to \mathbb{R}$ to be the function

$$\text{OptGNN}_{(M,\Lambda)}(U) = \mathcal{L}_\rho \circ \text{LAYER}_{M_T} \circ \cdots \circ \circ\text{LAYER}_{M_1}(U). \tag{7}$$

The per iteration update time of OptGNN is $O(\Phi r^\omega)$ where $\omega$ is the matrix multiplication exponent. We update the parameters of OptGNN by inputting the output of the final layer $\text{LAYER}_{M_T}$ into the Lagrangian $\mathcal{L}_\rho$ and backpropagate its derivatives. We emphasize the data is the instance $\Lambda$ and not the collection of vectors $U$ which can be chosen entirely at random. The form of the gradient $\nabla\mathcal{L}_\rho$ is a message passing algorithm over the nodes of the constraint graph $G_\Lambda$. Therefore, OptGNN is a message passing GNN over $G_\Lambda$ (see Definition G). This point is of practical importance as it is what informs out implementation of the OptGNN architecture. We then arrive at the following corollary.

**Corollary 1** (Informal). Given a Max-k-CSP instance $\Lambda$ represented in space $\Phi = O(|\mathcal{P}|q^k)$, there is an $\text{OptGNN}_{(M,\Lambda)}$ with $T = O(\epsilon^{-4}\Phi^4)$ layers, and embeddings of dimension $\Phi$ such that there is an instantiation of learnable parameters $M = \{M_t\}_{t \in [T]}$ that outputs a set of vectors $V$ satisfying the constraints of SDP 1 and approximating its optimum to error $\epsilon$. See formal statement 2

Moving on from our result on representing approximation algorithms, we ask whether OptGNN is learnable. That is to say, does OptGNN approximate the value of SDP 1 when given a polynomial amount of data? We provide a perturbation analysis to establish the polynomial sample complexity of PAC-learning OptGNN. The key idea is to bound the smoothness of the polynomial circuit AGG used in the OptGNN layer which is a cubic polynomial analogous to linear attention. We state the informal version below. For the formal version see Lemma E.5.

**Lemma 3.1** (PAC learning). Let $\mathcal{Q}$ be a dataset of Max-k-CSP instances over an alphabet of size $[q]$ with each instance represented in space $\Phi$. Here the dataset $\mathcal{D} := \Lambda_1, \Lambda_2, ..., \Lambda_\Gamma \sim \mathcal{D}$ is drawn i.i.d from a distribution over instances $\mathcal{D}$. Let $M$ be a set of parameters $M = \{M_1, M_2, ..., M_T\}$ in a parameter space $\Theta$. Then for $T = O(\epsilon^{-4}\Phi^4)$, for $\Gamma = O(\epsilon^{-4}\Phi^6 \log^4(\delta^{-1}))$, let $\widehat{M}$ be the empirical loss minimizing weights for arbitrary choice of initial embeddings $U$ in a bounded norm ball. Then we have that OptGNN is $(\Gamma, \epsilon, \delta)$-PAC learnable. That is to say the empirical loss minimizer $\text{EMP}(\mathcal{Q})$ is within $\epsilon$ from the distributional loss with probability greater than $1 - \delta$:

$$\Pr\left[\left|\text{EMP}(\mathcal{Q}) - \mathbb{E}_{\Lambda \sim \mathcal{D}}[\text{OptGNN}_{(\widehat{M},\Lambda)}(U)]\right| \leq \epsilon\right] \geq 1 - \delta.$$

We note that this style of perturbation analysis is akin to the VC theory on neural networks adapted to our unsupervised setting. Although it's insufficient to explain the empirical success of backprop, we believe our analysis sheds light on how architectures that capture gradient iterations can successfully generalize.

**OptGNN in practice**. Figure 1 summarizes the OptGNN pipeline for solving CO problems. OptGNN computes node embeddings $V \in \mathbb{R}^{N \times r}$ as per equation 7 which feeds into the loss $\mathcal{L}_\rho$. For pseudocode, please refer to the appendix (Max-Cut: algorithm 4 and general SDPs: algorithm 5). We use Adam (Kingma & Ba, 2014) to update the parameter matrices $M$. Given a training distribution $\mathcal{D}$, the network is trained in a completely unsupervised fashion by minimizing $\mathbb{E}_{G \sim \mathcal{D}}[\mathcal{L}(\mathbf{v}; G)]$ with a standard automatic differentiation package like PyTorch (Paszke et al., 2019). A practical benefit of our approach is that users do not need to reimplement the network to handle each new problem. Users need only implement the appropriate loss, and our implementation uses automatic differentiation to compute the messages in the forward pass. At inference time, the output embeddings must be rounded to a discrete solution. To do this, we select a random hyperplane vector $y \in \mathbb{R}^r$, and for each node with embedding vector $v_i$, we calculate its discrete assignment $x_i \in \{-1, 1\}$ as $x_i = \text{sign}(v_i^\top y)$. We use multiple hyperplanes and pick the best resulting solution.

**SDP 1** SDP for Max-k-CSP (Equivalent to UGC-optimal)

SDP Vector Formulation $\Lambda = (\mathcal{V}, \mathcal{P}, \{0, 1\})$.
Multilinear formulation of objective.

$$\min_{V} \sum_{P_z \in \mathcal{P}} \sum_{s \subseteq z} \frac{w_s}{\mathcal{C}(s)} \sum_{g, g' \subseteq s : \zeta(g, g') = s} \langle v_g, v_{g'} \rangle \qquad (8)$$

$$\text{subject to: } \|v_s\|^2 = 1 \quad \forall s \subseteq \mathcal{S}(P), \forall P \in \mathcal{P} \qquad (9)$$

$$\langle v_g, v_{g'} \rangle = \langle v_h, v_{h'} \rangle \quad \forall \zeta(g, g') = \zeta(h, h') \text{ s.t}$$
$$g \cup g' \subseteq \mathcal{S}(P), \; \forall P \in \mathcal{P} \qquad (10)$$

**Notation:** For a Max-CSP instance $\Lambda$, there is a canonical boolean polynomial representing the objective denoted $P(x_1, ..., x_N) = \sum_{P_z \in \mathcal{P}} P_z(X)$ where $P_z$ is the canonical polynomial associated to the predicate over the variables $z$. Conversely, for each predicate $P \in \mathcal{P}$ we let $\mathcal{S}(P)$ denote the variables in the domain of $P$. We write out the polynomial $P_z$ with coefficients $\{w_s\}_{s \subseteq z}$ as follows $P_z = \sum_{s \subseteq z} w_s \prod_{i \in s} x_i$. We introduce the set notation $\zeta(A, B) := A \cup B / A \cap B$ and use $\mathcal{C}(s)$ to denote the size of the set $\{g, g' \subseteq s : \zeta(g, g') = s\}$.

---

**Algorithm 1** Message Passing for Max-CSP (informal).

**Input: Max-CSP Instance** $\Lambda$ with constraint graph $G_\Lambda$
$\mathbf{v}^0 \leftarrow \text{Uniform}(\mathcal{S}^{n-1})$ {Initialization on the unit sphere}
**for** $t = 1, 2, \ldots, T$ **do**
  **for** $v_i^t \in \mathbf{v}^t$ {message passing on the constraint graph} **do**

$$v_i^{t+1} \leftarrow v_i^t - \eta \sum_{e, e' \in E(i)} \frac{\partial \mathcal{Y}_\rho(e^t, e'^t)}{\partial v_i^t}$$

  **end for**
**end for**
**Output: vector solution** $\mathbf{v}^t$ **for SDP 1**
**Notation:** $\mathcal{L}_\rho(V)$ can be written as the sum of functions across pairs of edges in the constraint graph $G_\Lambda$ i.e $\mathcal{L}_\rho(V) = \sum_{i \in V} \sum_{e, e' \in E(i)} \mathcal{Y}_\rho(e, e')$ where $E(i)$ is the set of edges involving vertex $i$, and $e$ denotes the vector embeddings of its two corresponding vertices. For analytic form of the messages see algorithm 3

---

**Obtaining neural certificates.** We construct dual certificates of optimality from the output embeddings of OptGNN. The certificates provide a bound on the optimal solution of the SDP. The key idea is to estimate the dual variables of SDP 1. Since we use a quadratic penalty for constraints, the natural dual estimate is one step of the augmented method of Lagrange multipliers on the SDP solution which can be obtained straightforwardly from the primal problem variables $V$. We then analytically bound the error in satisfying the KKT conditions of SDP 1. See C.1 for derivations and extended discussion, and Figure 3 for an experimental demonstration.

**Adaptation to Data vs. Theoretical Result**. The theory result concerns worst case optimality where the embedding dimension is as large as $N$. In practice we deploy OptGNN with varying choices of the embedding dimension to obtain the best performance. For low dimensional embeddings, the corresponding low rank SDP objective is NP-hard to optimize. As such, training OptGNN is like training any other GNN for CO in that the weights adapt to the data. This is highlighted in our experiments where OptGNN generally outperforms SDP solvers.

## 4 Experiments

First, we provide a comprehensive experimental evaluation of the OptGNN pipeline on several problems and datasets and compare it against heuristics, solvers, and neural baselines. We then describe a model ablation study, an out-of-distribution performance (OOD) study, and an experiment with our neural certification scheme. We empirically test the performance of OptGNN on NP-Hard combinatorial optimization problems: *Maximum Cut*, *Minimum Vertex Cover*, and *Maximum 3-SAT*. We obtain results for several datasets and compare against greedy algorithms, local search, a state-of-the-art MIP solver (Gurobi), and various neural baselines. For Max-3-SAT and Min-Vertex-Cover we adopt the quadratically penalized Lagrangian loss of their respective SDP relaxations. For details of the setup see Appendix D.

**Min-Vertex-Cover experiments.** We evaluated OptGNN on forced RB instances, which are hard vertex cover instances from the RB random CSP model that contain hidden optimal solutions (Xu et al., 2007). We use two distributions specified in prior work (Wang & Li, 2023), RB200 and RB500. The results are in Figure 2b, which also includes the performance of several neural and classical baselines as reported in (Wang & Li, 2023; Brusca et al., 2023). OptGNN consistently outperforms state-of-the-art unsupervised baselines on this task and is able to match the performance of Gurobi with a 0.5s time limit.

**Max-3-SAT experiment and ablation.** The purpose of this experiment is twofold: to demonstrate the

Table 1: Performance of OptGNN, Greedy, and Gurobi 0.1s, 1s, and 8s on Maximum Cut. For each approach and dataset, we report the average cut size measured on the test slice. Here, higher score is better. In parentheses, we include the average runtime in *milliseconds* for OptGNN.

| Dataset | OptGNN | Greedy | Gurobi 0.1s | Gurobi 1.0s |
|---|---|---|---|---|
| BA[a] (400,500) | 2197.99 (66) | 1255.22 | 2208.11 | 2208.11 |
| ER[a] (400,500) | 16387.46 (225) | 8622.34 | 16476.72 | 16491.60 |
| HK[a] (400,500) | 2159.90 (61) | 1230.98 | 2169.46 | 2169.46 |
| WC[a] (400,500) | 1166.47 (78) | 690.19 | 1173.45 | 1175.97 |
| ENZYMES[b] | 81.37 (14) | 48.53 | 81.45 | 81.45 |
| COLLAB[b] | 2622.41 (22) | 1345.70 | 2624.32 | 2624.57 |
| REDDIT-M-12K[c] | 568.00 (89) | 358.40 | 567.71 | 568.91 |
| REDDIT-M-5K[c] | 786.09 (133) | 495.02 | 785.44 | 787.48 |

Table 2: Average number of unsatisfied clauses for Max-3-SAT on random instances with $N = 100$ variables and clause ratios $r = 4.00, 4.15, 4.30$. Standard deviation of the ratio over the test set is reported in superscript. In parentheses, we report the average time per instance on the test set in seconds.

| Dataset | $r = 4.00$ | $r = 4.15$ | $r = 4.30$ |
|---|---|---|---|
| ErdősGNN | $5.46^{\pm 1.91}$ (0.01) | $6.14^{\pm 2.01}$ (0.01) | $6.79^{\pm 2.03}$ (0.01) |
| Walksat (100 restarts) | $0.14^{\pm 0.36}$ (0.12) | $0.36^{\pm 0.52}$ (0.25) | $0.68^{\pm 0.65}$ (0.40) |
| Walksat (1 restart) | $0.94^{\pm 0.92}$ (0.01) | $1.46^{\pm 1.11}$ (0.01) | $1.97^{\pm 1.28}$ (0.01) |
| Survey Propagation | $3.32^{\pm 0.81}$ (0.001) | $3.87^{\pm 0.79}$ (0.001) | $3.94^{\pm 0.93}$ (0.001) |
| OptGNN | $4.46^{\pm 1.68}$ (0.01) | $5.15^{\pm 1.76}$ (0.01) | $5.84^{\pm 2.18}$ (0.01) |
| Autograd SDP | $6.85^{\pm 2.33}$ (6.80) | $7.52^{\pm 2.38}$ (6.75) | $8.32^{\pm 2.50}$ (6.77) |
| Low-Rank SDP ((Wang & Kolter, 2019)) | $12.38^{\pm 1.06}$ (0.66) | $13.32^{\pm 1.09}$ (0.67) | $14.27^{\pm 1.08}$ (0.69) |

viability of OptGNN on the Max-3-SAT problem and to examine the role of overparameterization in OptGNN. We generate 3-SAT formulae on the fly with 100 variables and a random number of clauses in the $[400, 430]$ interval, and train OptGNN for 100,000 iterations. We then test on instances with 100 variables and $\{400, 415, 430\}$ clauses. The results are Table 2. We compare with WalkSAT, a classic local search algorithm for SAT, a low-rank SDP solver (Wang et al., 2019), Survey Propagation (Braunstein et al., 2005), and ErdosGNN Karalias & Loukas (2020), a neural baseline trained in the same way. We also compare with a simple baseline reported as "Autograd" that directly employs gradient descent on the penalized Lagrangian using the autograd functionality of Pytorch. For details see D.2. OptGNN is able to outperform ErdősGNN consistently and improves significantly over Autograd, which supports the overparameterized message passing of OptGNN. OptGNN performs better than the low-rank SDP solver, though does not beat WalkSAT/Survey Prop. It is worth noting that the performance of OptGNN could likely be further improved without significant computational cost by applying a local search post-processing step to its solutions but we did not pursue this further in order to emphasize the simplicity of our approach.

**Max-Cut experiments.** Table 1 presents a comparison between OptGNN, a greedy algorithm, and Gurobi for Max-Cut. OptGNN clearly outperforms greedy on all datasets and is competitive with Gurobi when Gurobi is restricted to a similar runtime. For results on more datasets see subsection D.3. Following the experimental setup of ANYCSP Tönshoff et al. (2022), we also tested OptGNN on the GSET benchmark instances (Benlic & Hao, 2013). We trained an OptGNN for 20k iterations on generated Erdős-Renyi graphs $\mathcal{G}_{n,p}$ for $n \in [400, 500]$ and $p = 0.15$. Figure 2a shows the results. We have included an additional low-rank SDP baseline to the results, while the rest of the baselines are reproduced as they appear in the original ANYCSP paper. These include state-of-the-art neural baselines, the Goemans-Williamson algorithm, and a greedy heuristic. We can see that OptGNN outperforms the SDP algorithms and the greedy algorithm, while also being competitive with the

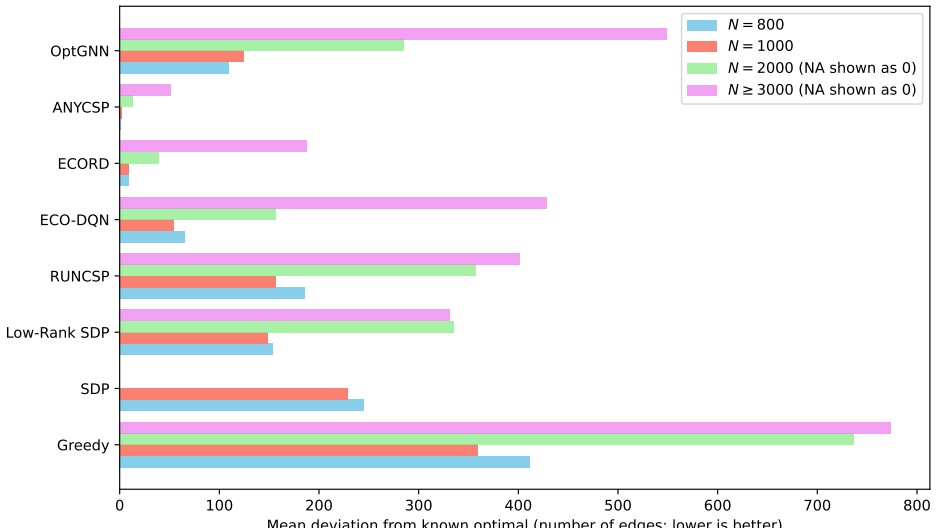

(a) Comparison on GSET Max-Cut instances against state-of-the-art neural baselines. Numbers reported are the mean (over the graphs in the test set) deviations from the best-known Max-Cut values, reported in Benlic & Hao (2013).

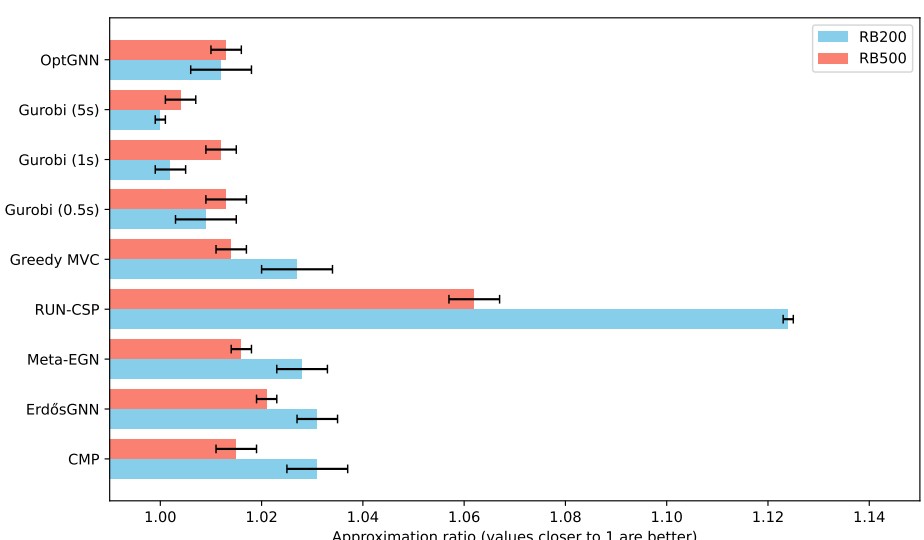

(b) Average approximation ratio and standard deviation over the test set for vertex covers on forced RB instances. A ratio of 1.000 represents finding the minimum vertex cover.

Figure 2: Results for Max-Cut and Minimum Vertex Cover.

neural baselines. However, OptGNN does not manage to outperform ANYCSP, which to the best of our knowledge achieves the current state-of-the-art results for neural networks.

**Out of distribution generalization.** We test OptGNN's ability to perform on data distributions (for the same optimization problem) that it's not trained on. The results can be seen in table 7 and subsection D.6. The results show that the model is capable of performing well even on datasets it has not been trained on.

**Model ablation.** We train modern GNN architectures from the literature with the same loss function and compare them against OptGNN. Please see Appendix D.4 for more details and results on multiple datasets for two different problems. Overall, OptGNN is the best performing model on both problems across all datasets.

**Experimental demonstration of neural certificates.** Next, we provide a simple experimental example of our neural certificate scheme on small synthetic instances. Deploying this scheme on Max-Cut on random graphs, we find this dual certificate to be remarkably tight. figure 3 shows an example. For 100 node graphs with 1000 edges our certificates deviate from the SDP certificate by about 20 nodes but are dramatically faster to produce. The runtime is dominated by the feedforward of OptGNN which is 0.02 seconds vs. the SDP solve time which is 0.5 seconds on cvxpy. See C.1 for extensive discussion and additional results.

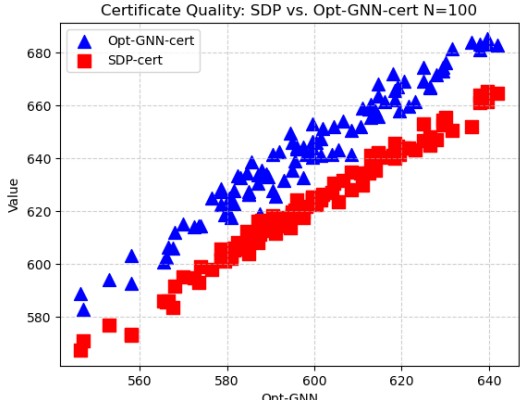

Figure 3: Experimental comparison of SDP versus OptGNN Dual Certificates on random graphs of 100 nodes for the Max-Cut problem. Our OptGNN certificates track closely with the SDP certificates while taking considerably less time to generate.

## 5 Conclusion

We presented OptGNN, a GNN that can capture provably optimal message passing algorithms for a large class of combinatorial optimization problems. OptGNN achieves the appealing combination of obtaining approximation guarantees while also being able to adapt to the data to achieve improved results. Empirically, we observed that the OptGNN architecture achieves strong performance on a wide range of datasets and on multiple problems. OptGNN opens up several directions for future exploration, such as the design of powerful rounding procedures that can secure approximation guarantees, the construction of neural certificates that improve upon the ones we described in Appendix C.1, and the design of neural SDP-based branch and bound solvers.

## 6 Acknowledgment

The authors would like to thank Ankur Moitra, Sirui Li, and Zhongxia Yan for insightful discussions in the preparation of this work. Nikolaos Karalias is funded by the SNSF, in the context of the project "General neural solvers for combinatorial optimization and algorithmic reasoning" (SNSF grant number: P500PT_217999). Stefanie Jegelka and Nikolaos Karalias acknowledge support from NSF AI Institute TILOS (NSF CCF-2112665). Stefanie Jegelka acknowledges support from NSF award 2134108.

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

# A Min-Vertex-Cover OptGNN

Min-Vertex-Cover can be written as the following integer program

$$\text{Minimize:} \quad \sum_{i \in [N]} \frac{1 + x_i}{2} \tag{11}$$

$$\text{Subject to:} \quad (1 - x_i)(1 - x_j) = 0 \quad \forall (i, j) \in E \tag{12}$$

$$x_i^2 = 1 \quad \forall i \in [N] \tag{13}$$

To deal with the constraint on the edges $(1 - x_i)(1 - x_j) = 0$, we add a quadratic penalty to the objective with a penalty parameter $\rho > 0$ yielding

$$\text{Minimize:} \quad \sum_{i \in [N]} \frac{1 + x_i}{2} + \rho \sum_{(i,j) \in E} (1 - x_i - x_j + x_i x_j)^2 \tag{14}$$

$$\text{Subject to:} \quad x_i^2 = 1 \quad \forall i \in [N] \tag{15}$$

Analogously to Max-Cut, we adopt a natural low rank vector formulation for vectors $\mathbf{v} = \{v_i\}_{i \in [N]}$ in $r$ dimensions.

$$\text{Minimize:} \quad \sum_{i \in [N]} \frac{1 + \langle v_i, v_\emptyset \rangle}{2} + \rho \sum_{(i,j) \in E} (1 - \langle v_i, v_\emptyset \rangle - \langle v_j, v_\emptyset \rangle + \langle v_i, v_j \rangle)^2 \tag{16}$$

$$\text{Subject to:} \quad \|v_i\| = 1 \quad v_i \in \mathbb{R}^r \quad \forall i \in [N] \tag{17}$$

Now we can design a simple projected gradient descent scheme as follows. For iteration $t$ in max iterations $T$, and for vector $v_i$ in $\mathbf{v}$ we perform the following update.

$$\hat{v}_i^{t+1} := v_i^t - \eta \big( v_\emptyset + 2\rho \sum_{j \in N(i)} (1 - \langle v_i^t, v_\emptyset \rangle - \langle v_j^t, v_\emptyset \rangle + \langle v_i^t, v_j^t \rangle)(-v_\emptyset + v_j^t) \big) \tag{18}$$

$$v_i^{t+1} := \frac{\hat{v}_i^{t+1}}{\|\hat{v}_i^{t+1}\|} \tag{19}$$

We can then define a $\text{OptGNN}_{(M,G)}$ analogously with learnable matrices $M = \{M_t\}_{t \in [T]} \in \mathbb{R}^{r \times 2r}$ which are each sets of $T$ learnable matrices corresponding to $T$ layers of neural network. Let the message from node $v_j$ to node $v_i$ be

$$\text{MESSAGE}[v_j \to v_i] := 2\rho(1 - \langle v_i^t, v_\emptyset \rangle - \langle v_j^t, v_\emptyset \rangle + \langle v_i^t, v_j^t \rangle)(-v_\emptyset + v_j^t)$$

Let the aggregation function AGG be defined as

$$\text{AGG}(\{v_j^t\}_{j \in N(i)}) := \begin{bmatrix} v_i^t \\ \sum_{j \in N(i)} \text{MESSAGE}[v_j \to v_i] \end{bmatrix}$$

Then for layer $t$ in max iterations $T$, for $v_i$ in $\mathbf{v}$, we have

$$\hat{v}_i^{t+1} := M \big( \text{AGG}(v_i^t) \big) \tag{20}$$

$$v_i^{t+1} := \frac{\hat{v}_i^{t+1}}{\|\hat{v}_i^{t+1}\|} \tag{21}$$

This approach can be straightforwardly adopted to compute the maximum clique and the maximum independent set.

# B   Max-3-SAT OptGNN

Our formulation for Max 3-SAT directly translates the OptGNN architecture from Definition 3.3. Let $\Lambda$ be a 3-SAT instance with a set of clauses $\mathcal{P}$ over a set of binary literals $\mathcal{V} = \{x_1, x_2, ..., x_N\} \in \{-1, 1\}$. Here $-1$ corresponds to assigning the literal to False whilst $1$ corresponds to True. Each clause $P_z \in \mathcal{P}$ can be specified by three literals $(x_i, x_j, x_k)$ and a set of three 'tokens' $(\tau_i, \tau_j, \tau_k) \in \{-1, 1\}^3$ which correspond to whether the variable $x_i, x_j, x_k$ are negated in the clause. For instance the clause $(x_1 \vee \neg x_2 \vee x_3)$ is translated into three literals $(x_1, x_2, x_3)$ and tokens $(1, -1, 1)$.

The 3-SAT problem on instance $\Lambda$ is equivalent to maximizing the following polynomial

$$\sum_{(x_i, x_j, x_k, \tau_i, \tau_j, \tau_k) \in \mathcal{P}} \left(1 - \frac{1}{8}(1 + \tau_i x_i)(1 + \tau_j x_j)(1 + \tau_k x_k)\right) \tag{22}$$

Subject to the constraint $x_1, x_2, ..., x_N \in \{1, -1\}$. Now we're reading to define the OptGNN for 3-SAT

**Objective:**   First we define the set of vector embeddings. For every literal $x_i$ we associate an embedding $v_i \in \mathbb{R}^r$. For every pair of literals that appear in a clause $(x_i, x_j)$ we associate a variable $v_{ij} \in \mathbb{R}^r$. Finally we associate a vector $v_\emptyset$ to represent 1. Then the unconstrained objective SAT is defined as

$$\text{SAT}(\Lambda) := \sum_{(x_i, x_j, x_k, \tau_i, \tau_j, \tau_k) \in \mathcal{P}} \frac{1}{8} \Bigg( -\tau_i \tau_j \tau_k \frac{1}{3} \left[ \langle v_i, v_{jk} \rangle + \langle v_j, v_{ik} \rangle + \langle v_k, v_{ij} \rangle \right]$$

$$- \tau_i \tau_j \frac{1}{2} \left[ \langle v_i, v_j \rangle + \langle v_{ij}, v_\emptyset \rangle \right] - \tau_i \tau_k \frac{1}{2} \left[ \langle v_i, v_k \rangle + \langle v_{ik}, v_\emptyset \rangle \right] - \tau_j \tau_k \frac{1}{2} \left[ \langle v_j, v_k \rangle + \langle v_{jk}, v_\emptyset \rangle \right]$$

$$- \tau_i \langle v_i, v_\emptyset \rangle - \tau_j \langle v_j, v_\emptyset \rangle - \tau_k \langle v_k, v_\emptyset \rangle + 7 \Bigg) \tag{23}$$

**Constraints:**   The constraints are then as follows. For every clause involving variables $(x_i, x_j, x_k)$ we impose the following constraints on $v_i, v_j, v_k, v_{ij}, v_{ik}, v_{jk}$ and $v_\emptyset$. Note that these constraints are exactly the ones listed in algorithm 2 which we organize here for convenience. The naming convention is the degree of the polynomial on the left to the degree of the polynomial on the right.

1. **pair-to-pair**

$$\langle v_i, v_j \rangle = \langle v_{ij}, v_\emptyset \rangle \tag{24}$$
$$\langle v_i, v_k \rangle = \langle v_{ik}, v_\emptyset \rangle \tag{25}$$
$$\langle v_j, v_k \rangle = \langle v_{jk}, v_\emptyset \rangle \tag{26}$$

2. **triplets-to-triplet**

$$\langle v_{ij}, v_k \rangle = \langle v_{ik}, v_j \rangle = \langle v_{jk}, v_i \rangle \tag{27}$$

3. **triplet-to-single**

$$\langle v_i, v_{ij} \rangle = \langle v_j, v_\emptyset \rangle \tag{28}$$
$$\langle v_j, v_{ij} \rangle = \langle v_i, v_\emptyset \rangle \tag{29}$$
$$\langle v_j, v_{jk} \rangle = \langle v_k, v_\emptyset \rangle \tag{30}$$
$$\langle v_k, v_{jk} \rangle = \langle v_j, v_\emptyset \rangle \tag{31}$$
$$\langle v_i, v_{ik} \rangle = \langle v_k, v_\emptyset \rangle \tag{32}$$
$$\langle v_k, v_{ik} \rangle = \langle v_i, v_\emptyset \rangle \tag{33}$$
$$\tag{34}$$

4. **quad-to-pair**

$$\langle v_{ij}, v_{jk} \rangle = \langle v_i, v_k \rangle \tag{35}$$
$$\langle v_{ij}, v_{ik} \rangle = \langle v_j, v_k \rangle \tag{36}$$
$$\langle v_{ik}, v_{jk} \rangle = \langle v_i, v_j \rangle \tag{37}$$
$$\tag{38}$$

5. **unit norm**

$$\|v_i\| = \|v_j\| = \|v_k\| = \|v_{ij}\| = \|v_{ik}\| = \|v_{jk}\| = \|v_\emptyset\| = 1 \tag{39}$$

Then for completeness we write out the full penalized objective. We perform this exercise in such great detail to inform the reader of how this could be set up for other Max-CSP's.

**Definition** (OptGNN for 3-SAT). Let $\Lambda$ be a 3-SAT instance. Let $\mathcal{L}_\rho$ be defined as

$$
\mathcal{L}_\rho(V) := \sum_{(x_i, x_j, x_k, \tau_i, \tau_j, \tau_k) \in \mathcal{P}} \left[ -\frac{1}{8} \left( -\tau_i \tau_j \tau_k \frac{1}{3} \left[ \langle v_i, v_{jk} \rangle + \langle v_j, v_{ik} \rangle + \langle v_k, v_{ij} \rangle \right] \right. \right.
$$
$$
- \tau_i \tau_j \frac{1}{2} \left[ \langle v_i, v_j \rangle + \langle v_{ij}, v_\emptyset \rangle \right] - \tau_i \tau_k \frac{1}{2} \left[ \langle v_i, v_k \rangle + \langle v_{ik}, v_\emptyset \rangle \right] - \tau_j \tau_k \frac{1}{2} \left[ \langle v_j, v_k \rangle + \langle v_{jk}, v_\emptyset \rangle \right]
$$
$$
\left. - \tau_i \langle v_i, v_\emptyset \rangle - \tau_j \langle v_j, v_\emptyset \rangle - \tau_k \langle v_k, v_\emptyset \rangle + 7 \right)
$$
$$
+ \rho \left[ (\langle v_i, v_j \rangle - \langle v_{ij}, v_\emptyset \rangle)^2 + (\langle v_i, v_k \rangle - \langle v_{ik}, v_\emptyset \rangle)^2 + (\langle v_j, v_k \rangle - \langle v_{jk}, v_\emptyset \rangle)^2 \right]
$$
$$
+ \rho \left[ (\langle v_{ij}, v_k \rangle - \langle v_{ik}, v_j \rangle)^2 + (\langle v_{ik}, v_j \rangle - \langle v_{jk}, v_i \rangle)^2 + (\langle v_{jk}, v_i \rangle - \langle v_{ij}, v_k \rangle)^2 \right]
$$
$$
+ \rho \left[ (\langle v_{ij}, v_{jk} \rangle - \langle v_i, v_k \rangle)^2 + (\langle v_{ij}, v_{ik} \rangle - \langle v_j, v_k \rangle)^2 + (\langle v_{ik}, v_{jk} \rangle - \langle v_i, v_j \rangle)^2 \right]
$$
$$
\rho \left[ (\langle v_i, v_{ij} \rangle - \langle v_j, v_\emptyset \rangle)^2 + (\langle v_j, v_{ij} \rangle - \langle v_i, v_\emptyset \rangle)^2 + (\langle v_j, v_{jk} \rangle - \langle v_k, v_\emptyset \rangle)^2 + (\langle v_k, v_{jk} \rangle - \langle v_j, v_\emptyset \rangle)^2 \right.
$$
$$
\left. + (\langle v_i, v_{ik} \rangle - \langle v_k, v_\emptyset \rangle)^2 + (\langle v_k, v_{ik} \rangle - \langle v_i, v_\emptyset \rangle)^2 \right]
$$
$$
\left. + \rho \left[ (\|v_i\| - 1)^2 + (\|v_j\| - 1)^2 + (\|v_k\| - 1)^2 + (\|v_{ij}\| - 1)^2 + (\|v_{ik}\| - 1)^2 + (\|v_{jk}\| - 1)^2 + (\|v_\emptyset\| - 1)^2 \right] \right]
$$
$$\tag{40}$$

Then taking the gradient of $\mathcal{L}$ gives us the precise form of the message passing. We list the forms of the messages from adjacent nodes in the constraint graph $G_\Lambda$.

**Message: Pair to Singles for Single in Pair.** This is the message MESSAGE$[v_{ij} \to v_i]$ for each pair node $v_{ij}$ to a single $v_i$ node.

$$\text{MESSAGE}[v_{ij} \to v_i] = |\{c \in \mathcal{P} : x_i, x_j \in c\}| 2\rho(\langle v_i, v_{ij} \rangle - \langle v_j, v_\emptyset \rangle) v_{ij} \tag{41}$$

**Message: Pair to Singles for Single not in Pair.** This is the message MESSAGE$[v_{ij} \to v_k]$ for each pair node $v_{ij}$ to a single $v_k$ node.

$$
\text{MESSAGE}[v_{ij} \to v_k] = \sum_{c \in \mathcal{P}:(x_i, x_j, x_k) \in c} \left[ -\frac{1}{8} \left( -\tau_i \tau_j \tau_k \frac{1}{3} v_{ij} \right) \right.
$$
$$
\left. + 2\rho \left[ (\langle v_k, v_{ij} \rangle - \langle v_i, v_{jk} \rangle) v_{ij} + (\langle v_i, v_{jk} \rangle - \langle v_k, v_{ij} \rangle)(-v_{ij}) \right] \right] \tag{42}
$$

**Message: Single to Pair for Single in Pair.** This is the message MESSAGE$[v_i \to v_{ij}]$ for each single node $v_i$ to a pair $v_k$

$$\text{MESSAGE}[v_i \to v_{ij}] = |\{c \in \mathcal{P} : x_i, x_j \in c\}| 2\rho(\langle v_i, v_{ij} \rangle - \langle v_j, v_\emptyset \rangle) v_i \tag{43}$$

**Message: Single to Pair for Single not in Pair.** This is the message MESSAGE$[v_{ij} \to v_k]$ for each pair node $v_{ij}$ to a single $v_k$ node.

$$
\text{MESSAGE}[v_{ij} \to v_k] = \sum_{c \in \mathcal{P}:(x_i, x_j, x_k) \in c} \left[ -\frac{1}{8} \left( -\tau_i \tau_j \tau_k \frac{1}{3} v_k \right) \right. \tag{44}
$$
$$
\left. + 2\rho \left[ (\langle v_k, v_{ij} \rangle - \langle v_i, v_{jk} \rangle) v_k + (\langle v_k, v_{ij} \rangle - \langle v_j, v_{ik} \rangle) v_k \right] \right] \tag{45}
$$

**Message: Single to Single** This is the message $\text{MESSAGE}[v_i \to v_j]$ for each single node $v_i$ to a single $v_j$ node.

$$\text{MESSAGE}[v_i \to v_j] = \sum_{c \in \mathcal{P}:(x_i,x_j,x_k)\in c} \left[ -\frac{1}{8}\left( -\tau_i \tau_j \frac{1}{2}v_i \right) \right. \tag{46}$$

$$\left. +2\rho\left[ (\langle v_i, v_j\rangle - \langle v_{ij}, v_\emptyset\rangle)v_i + (\langle v_{ik}, v_{jk}\rangle - \langle v_i, v_j\rangle)(-v_i) \right] \right] \tag{47}$$

**Message: Pair to Pair** This is the message $\text{MESSAGE}[v_{ij} \to v_{jk}]$ for each pair node $v_{ij}$ to a pair $v_{jk}$ node.

$$\text{MESSAGE}[v_{ij} \to v_{jk}] = |\{c \in \mathcal{P} : x_i, x_j, x_k \in c\}|2\rho\left[ (\langle v_{ij}, v_{jk}\rangle - \langle v_i, v_k\rangle)v_{ij} \right] \tag{48}$$

Then the $\text{OptGNN}_{(M,G_\Lambda)}$ is defined with the following functions UPDATE, AGGREGATE, and NONLINEAR. For any node $v$, the OptGNN update is

$$\text{AGGREGATE}(\{v_\zeta\}_{\zeta\in N(v)}) := \sum_{v_\zeta\in N(v)} \text{MESSAGE}[v_\zeta \to v] \tag{49}$$

$$\text{UPDATE}(v) = M\left( \begin{bmatrix} v \\ \text{AGGREGATE}(\{v_\zeta\}_{\zeta\in N(v)}) \end{bmatrix} \right) \tag{50}$$

$$\text{NONLINEAR}(v) = \frac{v}{\|v\|} \tag{51}$$

Finally, we note that many of the signs in the forms of messages could have been chosen differently i.e $\langle v_i, v_j\rangle - \langle v_{ij}, v_\emptyset\rangle$ produces different gradients from $\langle v_{ij}, v_\emptyset\rangle - \langle v_i, v_j\rangle$. We leave small choices like this to the reader.

## C   Optimality of Message Passing for Max-CSP

Our primary theoretical result is that a polynomial time message passing algorithm on an appropriately defined constraint graph computes the approximate optimum of SDP 2 which is notable for being an SDP that achieves the Unique Games optimal integrality gap.

Our proof roadmap is simple. First, we design an SDP relaxation SDP 2 for Max-k-CSP that is provably equivalent to the SDP of Raghavendra (2008) and therefore inherits its complexity theoretic optimality. Finally, we design a message passing algorithm to approximately solve SDP 2 in polynomial time to polynomial precision. Our message passing algorithm has the advantage of being formulated on an appropriately defined constraint graph. For a Max-k-CSP instance $\Lambda$ with $N$ variables, $|\mathcal{P}|$ predicates, over an alphabet of size $q$, it takes $|\mathcal{P}|q^k$ space to represent the Max-CSP. To declutter notation, we let $\Phi$ be the size of the Max-CSP which is equal to $|\mathcal{P}|q^k$ . Our message passing algorithm achieves an additive $\epsilon$ approximation in time $poly(\epsilon^{-1}, \Phi, \log(\delta^{-1}))$ which is then polynomial in the size of the CSP and inverse polynomial in the precision.

Here we briefly reiterate the definition of Max-k-CSP. A Max-k-CSP instance $\Lambda = (\mathcal{V}, \mathcal{P}, q)$ consists of a set of $N$ variables $\mathcal{V} := \{x_i\}_{i\in[N]}$ each taking values in an alphabet $[q]$ and a set of predicates $\mathcal{P} := \{P_z\}_{z\subset\mathcal{V}}$ where each predicate is a payoff function over $k$ variables denoted $z = \{x_{i_1}, x_{i_2}, ..., x_{i_k}\}$. Here we refer to $k$ as the arity of the Max-k-CSP, and we adopt the normalization that each predicate $P_z$ returns outputs in $[0, 1]$. We index each predicate $P_z$ by its domain $z$ and we will use the notation $\mathcal{S}(P)$ to denote the domain of a predicate $P$. The goal of Max-k-CSP is to maximize the payoff of the predicates.

$$\max_{(x_1,...,x_N)\in[q]^N} \frac{1}{|\mathcal{P}|} \sum_{P_z\in\mathcal{P}} P_z(X_z) \tag{52}$$

Where $X_z$ denotes the assignment of variables $\{x_i\}_{i\in z}$.

There is an SDP relaxation of equation 52 that is the "qualitatively most powerful assuming the Unique Games conjecture" Raghavendra (2008). More specifically, the integrality gap of the SDP achieves the Unique Games optimal approximation ratio. Furthermore, there exists a rounding that achieves its integrality gap.

**SDP Reformulation:** Next we will introduce the SDP formulation we adopt in this paper. For the sake of exposition and notational simplicity, we will work with binary Max-k-CSP's where $q = \{0, 1\}$. The extension to general $q$ is straightforward and detailed in the appendix.

We will adopt the standard pseudoexpectation and pseudodistribution formalism in describing our SDP. Let $\tilde{\mathbb{E}}_\mu[\mathbf{x}]$ be a matrix in dimension $\mathbb{R}^{(N+1)^{d/2} \times (N+1)^{d/2}}$ of optimization variables defined as follows

$$\tilde{\mathbb{E}}_\mu[\mathbf{x}] := \tilde{\mathbb{E}}_\mu[(1, x_1, x_2, ..., x_N)^{\otimes d/2}((1, x_1, x_2, ..., x_N)^{\otimes d/2})^T] \tag{53}$$

Where we use $\otimes$ to denote tensor product. It is convenient to think of $\tilde{\mathbb{E}}_\mu[\mathbf{x}]$ as a matrix of variables denoting the up to $d$ multilinear moments of a distribution $\mu$ over the variables $\mathcal{V}$. A multilinear polynomial is a polynomial of the form $X_\phi := \prod_{i \in \phi} x_i$ for some subset of the variables $\phi \subseteq \mathcal{V}$. We index the variables of the matrix $\tilde{\mathbb{E}}_\mu[\mathbf{x}]$ by the multilinear moment that it represents. Notice that this creates repeat copies as their are multiple entries representing the same monomial. This is dealt with by constraining the repeated copies to be equal with linear equality constraints.

Specifically, let $z$ be a subset of the CSP variables $z \subset \{x_i\}_{i \in [N]}$ of size $k$. Let $X_z$ denote the multilinear moment $X_z := \prod_{i \in z} x_i$. Then $\tilde{\mathbb{E}}_\mu[X_z]$ denotes the SDP variable corresponding to the multilinear moment $\mathbb{E}_\mu[X_z]$. Of course optimizing over the space of distributions $\mu$ over $\mathcal{V}$ is intractable, and so we opt for optimizing over the space of low degree pseudodistributions and their associated pseudoexpecation functionals. See Barak & Steurer (2014) for references therein.

In particular, for any subset of variables $X_z := \{x_{i_1}, ..., x_{i_k}\} \in \mathcal{V}$ we let $\tilde{\mathbb{E}}_\mu[\mathbf{x}]\big|_{z,d}$ denote the matrix of the up to degree up to $d$ multilinear moments of the variables in $z$.

$$\tilde{\mathbb{E}}_\mu[\mathbf{x}]\big|_{z,d} := \tilde{\mathbb{E}}_\mu[(1, x_{i_1}, x_{i_2}, ..., x_{i_k})^{\otimes d/2}((1, x_{i_1}, x_{i_2}, ..., x_{i_k})^{\otimes d/2})^T] \tag{54}$$

We refer to the above matrix as a degree $d$ pseudoexpectation funcitonal over $X_z$. Subsequently, we describe a pseudoexpectation formulation of our SDP followed by a vector formulation.

**Multilinear Formulation:** A predicate for a boolean Max-k-CSP $P_z(X_z)$ can be written as a multilinear polynomial

$$P_z(X_z) := \sum_{\tau = (\tau_1, ..., \tau_k) \in \{-1, 1\}^k} w_{z,\tau} \prod_{x_i \in z} \frac{1 + \tau_i x_i}{2} := \sum_{s \subseteq z} y_s X_s \tag{55}$$

For some real valued weights $w_{z,\tau}$ and $y_s$ which are simply the fourier coefficients of the function $P_z$. Then the pseudoexpectation formulation of our SDP is as follows

$$\max_{\tilde{\mathbb{E}}_\mu[\mathbf{x}]} \frac{1}{|\mathcal{P}|} \sum_{P_z \in \mathcal{P}} \tilde{\mathbb{E}}_\mu[P_z(X_z)] \tag{56}$$

subject to the following constraints

1. **Unit:** $\tilde{\mathbb{E}}_\mu[1] = 1$, $\tilde{\mathbb{E}}_\mu[x_i^2] = 1$ for all $x_i \in \mathcal{V}$, and $\tilde{\mathbb{E}}_\mu[\prod_{i \in s} x_i^2 \prod_{j \in s'} x_j] = \tilde{\mathbb{E}}_\mu[\prod_{j \in s'} x_j]$ for all $s, s' \subseteq \mathcal{S}(P)$ for every predicate $P \in \mathcal{P}$ such that $2s + s' \le k$. In expectation, the squares of all multilinear polynomials are equal to 1.

2. **Positive Semidefinite:** $\tilde{\mathbb{E}}_\mu[\mathbf{x}]|_{\mathcal{V},2} \succeq 0$ i.e the degree two pseudoexpectation is positive semidefinite. $\tilde{\mathbb{E}}_\mu[\mathbf{x}]\big|_{z,2k} \succeq 0$ for all $z = \mathcal{S}(P)$ for all $P \in \mathcal{P}$. The moment matrix for the multilinear polynomials corresponding to every predicate is positive semidefinite.

Equivalently we can view the SDP in terms of the vectors in the cholesky decomposition of $\tilde{\mathbb{E}}_\mu[\mathbf{x}]$. We rewrite the above SDP accordingly. For this purpose it is useful to introduce the notation $\zeta(A, B) := A \cup B / A \cap B$. It is also useful to introduce the notation $\mathcal{C}(s)$ for the size of the set $\{g, g' \subseteq s : \zeta(g, g') = s\}$.

**SDP 2** SDP for Max-k-CSP (Equivalent to UGC-optimal)

---

SDP Vector Formulation $\Lambda = (\mathcal{V}, \mathcal{P}, \{0,1\})$. Multilinear formulation of objective.

$$\min \frac{1}{|\mathcal{P}|} \sum_{P_z \subset \mathcal{P}} \tilde{\mathbb{E}}_\mu[-P_z(X_z)] := \sum_{P_z \in \mathcal{P}} \sum_{s \subseteq z} w_s \frac{1}{|\mathcal{C}(s)|} \sum_{g,g' \subseteq s: \zeta(g,g')=s} \langle v_g, v_{g'} \rangle \tag{57}$$

$$\text{subject to: } \|v_s\|^2 = 1 \quad \forall s \subseteq \mathcal{S}(P), \forall P \in \mathcal{P} \tag{58}$$

$$\tilde{\mathbb{E}}_\mu[X_{\zeta(g,g')}] := \langle v_g, v_{g'} \rangle$$
$$= \langle v_h, v_{h'} \rangle \quad \forall \zeta(g,g') = \zeta(h,h') \text{ s.t } g \cup g' \subseteq \mathcal{S}(P), \forall P \in \mathcal{P} \tag{59}$$

First constraint is the square of multilinear polynomials are unit.
Second constraint are degree $2k$ SoS constraints for products of multilinear polynomials.

---

**Lemma C.1.** For Max-k-CSP instance $\Lambda$, The SDP of SDP 2 is at least as tight as the SDP of Raghavendra (2008).

*Proof.* The SDP of Raghavendra (2008) is a degree 2 SoS SDP augmented with $k$-local distributions for every predicate $P \in \mathcal{P}$. By using the vectors of the cholesky decomposition and constraining them to be unit vectors we automatically capture degree 2 SoS. To capture $k$ local distributions we simply enforce degree $2k$ SoS on the boolean hypercube for the domain of every predicate. This can be done with the standard vector formulation written in SDP 2. See Barak & Steurer (2014) for background and references. $\square$

Moving forward, the goal of Algorithm 3 is to minimize the loss $\mathcal{L}_\rho(\mathbf{v})$ which is a function of the Max-CSP instance $\Lambda$.

$$\mathcal{L}_\rho(\mathbf{v}) = \frac{1}{|\mathcal{P}|} \Bigg[ \sum_{P_z \in \mathcal{P}} \sum_{s \subseteq z} y_s \frac{1}{|\mathcal{C}(s)|} \sum_{\substack{g,g' \subseteq s \\ \text{s.t } \zeta(g,g')=s}} \langle v_g, v_{g'} \rangle$$

$$+ \rho \Bigg[ \sum_{P_z \in \mathcal{P}} \sum_{\substack{g,g',h,h' \subseteq z \\ \text{s.t } \zeta(g,g')=\zeta(h,h')}} \left( \langle v_g, v_{g'} \rangle - \langle v_h, v_{h'} \rangle \right)^2$$

$$+ \sum_{v_s \in \mathbf{v}} (\|v_s\|^2 - 1)^2 \Bigg] \Bigg] \tag{60}$$

The loss $\mathcal{L}_\rho$ has gradient of the form

$$\frac{\partial \mathcal{L}_\rho(\mathbf{v})}{\partial v_w} = \frac{1}{|\mathcal{P}|} \Bigg[ \sum_{\substack{P_z \in \mathcal{P} \\ \text{s.t } w \subseteq z}} \sum_{\substack{s \subseteq z \\ \text{s.t } w \subseteq s}} y_s \frac{1}{|\mathcal{C}(s)|} \sum_{\substack{w' \subseteq s \\ \text{s.t } \zeta(w,w')=s}} v_{w'}$$

$$+ 2\rho \Bigg[ \sum_{\substack{P_z \in \mathcal{P} \\ \text{s.t } w \subseteq z}} \sum_{\substack{w',h,h' \subseteq s \\ \text{s.t } \zeta(w,w')=\zeta(h,h')}} \left( \langle v_w, v_{w'} \rangle - \langle v_h, v_{h'} \rangle \right) v'_w$$

$$+ (\|v_w\|^2 - 1) v_w \Bigg] \Bigg] \tag{61}$$

Noticing that the form of the gradient depends only on the vectors in the neighborhood of the constraint graph $G_\Lambda$ we arrive at our message passing algorithm. The key to our proof is bounding the number of iterations required to optimize equation 60 to sufficient accuracy to be an approximate global optimum of SDP 2.

**Theorem C.1.** *Algorithm 3 computes in $O(\epsilon^{-4} \Phi^4 \log(\delta^{-1}))$ iterations a set of vectors $\mathbf{v} := \{\hat{v}_s\}$ for all $s \subseteq \mathcal{S}(P)$ for all $P \in \mathcal{P}$ that satisfy the constraints of SDP 2 to error $\epsilon$ and approximates the*

---

**Algorithm 3** Message Passing for Max-CSP

---
1: **Inputs: Max-CSP instance** $\Lambda$
2: $n \leftarrow \Phi \log(\delta^{-1})$
3: $\eta, \psi, \sigma \leftarrow n^{-2}$ {Initialize step size, noise threshold, and noise variance}
4: $\mathbf{v}^0 = \{v_s\}_{s \subseteq z : P_z \in \mathcal{P}} \leftarrow Uniform(\mathcal{S}^{n-1})$ {Initialize vectors to uniform on the unit sphere}
5: **for** $t \in [O(\epsilon^{-4}\Phi^4 \log(\delta^{-1}))]$ **do**
6:    **for** $v_w^t \in \mathbf{v}^t$ {Iterate over vectors and update each vector with neighboring vectors in constraint graph} **do**
7:

$$
v_w^{t+1} \leftarrow v_w^t - \eta \frac{1}{|\mathcal{P}|} \Bigg[ \sum_{\substack{P_z \in \mathcal{P} \\ \text{s.t } w \subseteq z}} \sum_{\substack{s \subseteq z \\ \text{s.t } w \subseteq s}} y_s \frac{1}{|\mathcal{C}(s)|} \sum_{\substack{w' \subseteq s \\ \text{s.t } \zeta(w,w')=s}} v_{w'}^t \tag{62}
$$

$$
+ 2\rho \Bigg[ \sum_{\substack{P_z \in \mathcal{P} \\ \text{s.t } w \subseteq z}} \sum_{\substack{w',h,h' \subseteq s \\ \text{s.t } \zeta(w,w')=\zeta(h,h')}} \left( \langle v_w^t, v_{w'}^t \rangle - \langle v_h^t, v_{h'}^t \rangle \right) v_w^{\prime t} + (\|v_w^t\|^2 - 1)v_w^t \Bigg] \Bigg]
$$

8:      **if** $\|v_w^{t+1} - v_w^t\| \le \psi$ **then**
9:        $\zeta \leftarrow N(0, \sigma I)$
10:     **else**
11:        $\zeta \leftarrow 0$
12:     **end if**
13:     $v_w^{t+1} \leftarrow v_w^{t+1} + \zeta$
14:    **end for**
15: **end for**
16: **return** $\mathbf{v}^t$
17: **Output: vectors corresponding to solution to** SDP 2

---

*optimum of SDP 2 to error $\epsilon$ with probability $1 - \delta$*

$$
\Big| \sum_{P_z \in \mathcal{P}} \tilde{\mathbb{E}}_{\hat{\mu}}[P_z(X_z)] - SDP(\Lambda) \Big| \le \epsilon
$$

*where $SDP(\Lambda)$ is the optimum of SDP 2.*

*Proof.* We begin by writing down the objective penalized by a quadratic on the constraints.

$$
\mathcal{L}_\rho(\mathbf{v}) := \frac{1}{|\mathcal{P}|} \Bigg[ \sum_{P_z \in \mathcal{P}} \tilde{\mathbb{E}}_\mu[P_z(X_z)]
$$

$$
+ \rho \Bigg[ \sum_{\substack{P_z \in \mathcal{P} \\ \text{s.t } \zeta(g,g')=\zeta(h,h')}} \sum_{\substack{g,g',h,h' \subseteq z}} \left( \langle v_g, v_{g'} \rangle - \langle v_h, v_{h'} \rangle \right)^2 + \sum_{v_s \in \mathbf{v}} (\|v_s\|^2 - 1)^2 \Bigg] \Bigg] \tag{63}
$$

For any monomial $X_s = \prod_{i \in s} x_i$ in $P_z(X_z)$ we write

$$
\tilde{\mathbb{E}}_\mu[X_s] := \frac{1}{|\mathcal{C}(s)|} \sum_{\substack{g,g' \subseteq s \\ \text{s.t } \zeta(g,g')=s}} \langle v_g, v_{g'} \rangle \tag{64}
$$

Where $\mathcal{C}(s)$ is the size of the set $\{g, g' \subseteq s : \zeta(g, g') = s\}$. In a small abuse of notation, we regard this as the definition of $\tilde{\mathbb{E}}_\mu[X_s]$ but realize that we're referring to the iterates of the algorithm before they've converged to a pseudoexpectation. Now recall equation 55, we can expand the polynomial $P_z(X_z)$ along its standard monomial basis

$$P_z(X_z) = \sum_{s \subseteq z} y_s X_s \tag{65}$$

where we have defined coefficients $y_s$ for every monomial in $P_z(X_z)$. Plugging equation 64 and equation 65 into equation 63 we obtain

$$\mathcal{L}_\rho(\mathbf{v}) = \frac{1}{|\mathcal{P}|}\Bigg[ \sum_{P_z \in \mathcal{P}} \sum_{s \subseteq z} y_s \frac{1}{|\mathcal{C}(s)|} \sum_{\substack{g,g' \subseteq s \\ \text{s.t } \zeta(g,g')=s}} \langle v_g, v_{g'} \rangle$$

$$+ \rho \Bigg[ \sum_{P_z \in \mathcal{P}} \sum_{\substack{g,g',h,h' \subseteq z \\ \text{s.t } \zeta(g,g')=\zeta(h,h')}} \big( \langle v_g, v_{g'} \rangle - \langle v_h, v_{h'} \rangle \big)^2$$

$$+ \sum_{v_s \in \mathbf{v}} (\|v_s\|^2 - 1)^2 \Bigg]\Bigg] \tag{66}$$

Taking the derivative with respect to any $v_w \in \mathbf{v}$ we obtain

$$\frac{\partial \mathcal{L}_\rho(\mathbf{v})}{\partial v_w} = \frac{1}{|\mathcal{P}|}\Bigg[ \sum_{\substack{P_z \in \mathcal{P} \\ \text{s.t } w \subseteq z}} \sum_{\substack{s \subseteq z \\ \text{s.t } w \subseteq s}} y_s \frac{1}{|\mathcal{C}(s)|} \sum_{\substack{w' \subseteq s \\ \text{s.t } \zeta(w,w')=s}} v_{w'}$$

$$+ 2\rho \Bigg[ \sum_{\substack{P_z \in \mathcal{P} \\ \text{s.t } w \subseteq z}} \sum_{\substack{w',h,h' \subseteq s \\ \text{s.t } \zeta(w,w')=\zeta(h,h')}} \big( \langle v_w, v_{w'} \rangle - \langle v_h, v_{h'} \rangle \big) v_w'$$

$$+ (\|v_w\|^2 - 1) v_w \Bigg]\Bigg] \tag{67}$$

The gradient update is then what is detailed in Algorithm 3

$$v_w^{t+1} = v_w^t - \eta \frac{\partial \mathcal{L}_\rho(\mathbf{v})}{\partial v_w} \tag{68}$$

Thus far we have established the form of the gradient. To prove the gradient iteration converges we reference the literature on convergence of perturbed gradient descent (Jin et al., 2017) which we rewrite in Theorem F.1. First we note that the SDP 2 has $\ell$ smooth gradient for $\ell \leq poly(\rho, B)$ and has $\gamma$ lipschitz Hessian for $\gamma = poly(\rho, B)$ which we arrive at in Lemma E.2 and Lemma F.1. The proofs of Lemma E.2 and Lemma F.1 are technically involved and form the core hurdle in arriving at our proof.

Then by Theorem F.1 the iteration converges to an $(\epsilon', \gamma^2)$-SOSP Definition F.1 in no more than $\tilde{O}(\frac{1}{\epsilon'^2})$ iterations with probability $1 - \delta$. Now that we've established that the basic gradient iteration converges to approximate SOSP, we need to then prove that the approximate SOSP are approximate global optimum. We achieve this by using the result of Bhojanapalli et al. (2018) lemma 3 which we restate in Lemma C.2 for convenience.

The subsequent presentation adapts the proof of Lemma C.2 to our setting. To show that $(\epsilon', \gamma^2)$-SOSP are approximately global optimum, we have to work with the original SDP loss as opposed to the nonconvex vector loss. For subsequent analysis we need to define the penalized loss which we denote $\mathcal{H}_\rho(\tilde{\mathbb{E}}[\mathbf{x}])$ in terms of the SDP moment matrix $\tilde{\mathbb{E}}[\mathbf{x}]$.

$$\mathcal{H}_\rho(\tilde{\mathbb{E}}[\mathbf{x}]) := \frac{1}{|\mathcal{P}|}\Bigg[ \sum_{P_z \in \mathcal{P}} \tilde{\mathbb{E}}_{\hat{\mu}}[P_z(X_z)]$$

$$+ \rho \Bigg[ \sum_{P_z \in \mathcal{P}} \sum_{\substack{g,g',h,h' \subseteq z \\ \text{s.t } \zeta(g,g')=\zeta(h,h')}} \big( \tilde{\mathbb{E}}_{\hat{\mu}}[X_{\zeta(g,g')}] - \tilde{\mathbb{E}}_{\hat{\mu}}[X_{\zeta(h,h')}] \big)^2 + \sum_{\substack{X_s \text{ s.t } s \subset \mathcal{S}(P) \\ |s| \leq k, \forall P \in \mathcal{P}}} \big( \tilde{\mathbb{E}}_{\hat{\mu}}[X_s^2] - 1 \big)^2 \Bigg]\Bigg] \tag{69}$$

Here we use the notation $\tilde{\mathbb{E}}_{\hat{\mu}}[X_{\zeta(g,g')}]$ and $\tilde{\mathbb{E}}_{\hat{\mu}}[X_{\zeta(h,h')}]$ to denote $\langle v_g, v'_g \rangle$ and $\langle v_h, v'_h \rangle$ respectively. Note that although by definition $\mathcal{H}_\rho(\tilde{\mathbb{E}}[\mathbf{x}]) = \mathcal{L}_\rho(\mathbf{v})$, their gradients and hessians are distinct because $\mathcal{L}_\rho(\mathbf{v})$ is overparameterized. This is the key point. It is straightforward to argue that SOSP of a convex optimization are approximately global, but we are trying to make this argument for SOSP of the overparameterized loss $\mathcal{L}_\rho(\mathbf{v})$.

Let the global optimum of SDP 2 be denoted $\tilde{\mathbb{E}}_{\tilde{\mu}}[\tilde{\mathbf{x}}]$ with a cholesky decomposition $\tilde{\mathbf{v}}$. Let $\hat{\mathbf{v}}$ be the set of vectors outputted by Algorithm 3 with associated pseudoexpectation $\tilde{\mathbb{E}}_{\hat{\mu}}[\hat{\mathbf{x}}]$. Then, we can bound

$$\mathcal{L}_\rho(\hat{\mathbf{v}}) - \mathcal{L}_\rho(\tilde{\mathbf{v}}) = \mathcal{H}_\rho(\tilde{\mathbb{E}}[\hat{\mathbf{x}}]) - \mathcal{H}_\rho(\tilde{\mathbb{E}}[\tilde{\mathbf{x}}]) \leq \left\langle \nabla \mathcal{H}_\rho(\tilde{\mathbb{E}}[\hat{\mathbf{x}}]), \tilde{\mathbb{E}}[\hat{\mathbf{x}}] - \tilde{\mathbb{E}}[\tilde{\mathbf{x}}] \right\rangle \tag{70}$$

Here the first equality is by definition, and the inequality is by the convexity of $\mathcal{H}_\rho$. Moving on, we use the fact that the min eigenvalue of the hessian of overparameterized loss $\nabla^2 \mathcal{L}_\rho(\hat{\mathbf{v}}) \succeq -\gamma\sqrt{\epsilon'}$ implies the min eigenvalue of the gradient of the convex loss $\lambda_{\min}(\nabla \mathcal{H}_\rho(\tilde{\mathbb{E}}[\hat{\mathbf{x}}])) \geq -\gamma\sqrt{\epsilon'}$. This fact is invoked in Bhojanapalli et al. (2018) lemma 3 which we adapt to our setting in Lemma C.2. Subsequently, we adapt the lines of their argument in Lemma C.2 most relevant to our analysis which we detail here for the sake of completeness.

$$\begin{aligned}
70 &\leq -\lambda_{\min}(\nabla \mathcal{H}_\rho(\tilde{\mathbb{E}}[\hat{\mathbf{x}}])) \operatorname{Tr}(\tilde{\mathbb{E}}[\hat{\mathbf{x}}]) - \left\langle \nabla \mathcal{H}_\rho(\tilde{\mathbb{E}}[\hat{\mathbf{x}}]), \tilde{\mathbb{E}}[\tilde{\mathbf{x}}] \right\rangle \\
&\leq -\lambda_{\min}(\nabla \mathcal{H}_\rho(\tilde{\mathbb{E}}[\hat{\mathbf{x}}])) \operatorname{Tr}(\tilde{\mathbb{E}}[\hat{\mathbf{x}}]) + \|\nabla \mathcal{H}_\rho(\tilde{\mathbb{E}}[\hat{\mathbf{x}}])\|_F \|\tilde{\mathbb{E}}[\tilde{\mathbf{x}}]\|_F \\
&\leq \gamma\sqrt{\epsilon'} \operatorname{Tr}(\tilde{\mathbb{E}}[\mathbf{x}]) + \epsilon'\|\tilde{\mathbf{v}}\|_F \leq \gamma\sqrt{\epsilon'}\Phi + \epsilon'\Phi \leq \epsilon
\end{aligned}$$
$$\tag{71}$$

Here the first inequality follows by a standard inequality of frobenius inner product, the second inequality follows by Cauchy-Schwarz, the third inequality follows by the $(\epsilon', \gamma^2)$-SOSP conditions on both the min eigenvalue of the hessian and the norm of the gradient, the final two inequalities follow from knowing the main diagonal of $\tilde{\mathbb{E}}[\hat{\mathbf{x}}]$ is the identity and that every vector in $\tilde{\mathbf{v}}$ is a unit vector up to inverse polynomial error $poly(\rho^{-1}, 2^k)$. For this last point see the proof in Lemma E.1. Therefore if we set $\epsilon' = poly(\epsilon, 2^{-k})$ we arrive at any $\epsilon$ error. Therefore we have established our estimate $\hat{v}$ is approximates the global optimum of the quadratically penalized objective i.e $\mathcal{H}_\rho(\tilde{\mathbb{E}}[\hat{\mathbf{x}}]) - \mathcal{H}_\rho(\tilde{\mathbb{E}}[\tilde{\mathbf{x}}]) \leq \epsilon$. To finish our proof, we have to bound the distance between the global optimum of the quadratically penalized objective $\mathcal{H}_\rho(\tilde{\mathbb{E}}[\tilde{\mathbf{x}}])$ and SDP($\Lambda$) the optimum of SDP 2. This is established for $\rho$ a sufficiently large $poly(\epsilon^{-1}, 2^k)$ in Lemma E.1. This concludes our proof that the iterates of Algorithm 3 converge to the solution of SDP 2. $\qquad\square$

In our proof above, the bound on approximate SOSP being approximate global optimum is built on the result of Bhojanapalli et al. (2018) which we rephrase for our setting below.

**Lemma C.2.** [Bhojanapalli et al. (2018) lemma 3 rephrased] Let $\mathcal{L}_\rho(\cdot)$ be defined as in equation 63 and let $\mathcal{H}_\rho(\cdot)$ be defined as in equation 69. Let $U \in \mathbb{R}^{\Phi \times \Phi}$ be an $(\epsilon, \gamma)$-SOSP of $\mathcal{L}_\rho(\cdot)$, then

$$\lambda_{\min}(\nabla \mathcal{H}_\rho(\tilde{\mathbb{E}}[\hat{\mathbf{x}}])) \geq -\gamma\sqrt{\epsilon}$$

Furthermore, the global optimum $\tilde{\mathbf{X}}$ obeys the optimality gap

$$\mathcal{H}_\rho(UU^T) - \mathcal{H}_\rho(\tilde{\mathbf{X}}) \leq \gamma\sqrt{\epsilon} \operatorname{Tr}(\tilde{\mathbf{X}}) + \frac{1}{2}\epsilon\|U\|_F$$

The following Lemma E.1 establishes that for a sufficiently large penalty parameter $\rho = poly(\epsilon^{-1}, 2^k)$ the optimum of the penalized problem and the exact solution to SDP 2 are close.

**Lemma C.3.** Let $\Lambda$ be a Max-k-CSP instance, and let SDP($\Lambda$) be the optimum of SDP 2. Let $\mathcal{L}_\rho(\mathbf{v})$ be the quadratically penalized objective

$$\mathcal{L}_\rho(\mathbf{v}) := \frac{1}{|\mathcal{P}|}\left[ \sum_{P_z \in \mathcal{P}} \sum_{s \subseteq z} y_s \frac{1}{|\mathcal{C}(s)|} \sum_{\substack{g,g' \subseteq s \\ \text{s.t } \zeta(g,g')=s}} \langle v_g, v_{g'} \rangle + \rho\left[ \sum_{P_z \in \mathcal{P}} \sum_{\substack{g,g',h,h' \subseteq z \\ \zeta(g,g')=\zeta(h,h')}} \left( \langle v_g, v_{g'} \rangle - \langle v_h, v_{h'} \rangle \right)^2 \right.\right.$$
$$\left.\left. + \sum_{v_s \in \mathbf{v}} (\|v_s\|^2 - 1)^2 \right]\right] \tag{72}$$

Let $\tilde{v}$ be the argmin of the unconstrained minimization

$$\tilde{v} := \underset{\mathbf{v} \in \mathbb{R}^{|\mathcal{P}|^2(2^{2k})}}{\arg\min} \mathcal{L}_\rho(\mathbf{v})$$

Then we have

$$\mathcal{L}_\rho(\tilde{\mathbf{v}}) - \mathrm{SDP}(\Lambda) \le \epsilon$$

for $\rho = poly(\epsilon^{-1}, 2^k)$

*Proof.* We begin the analysis with the generic equality constrained semidefinite program of the form

$$\text{Minimize:} \quad \langle C, X \rangle \tag{73}$$
$$\text{Subject to:} \quad \langle A_i, X \rangle = b_i \quad \forall i \in \mathcal{F} \tag{74}$$
$$X \succeq 0 \tag{75}$$
$$X \in \mathbb{R}^{d \times d} \tag{76}$$

For an objective matrix $C$ and constraint matrices $\{A_i\}_{i \in \mathcal{F}}$ in some constraint set $\mathcal{F}$. We will invoke specific properties of SDP 2 to enable our analysis. First we define the penalized objective in this generic form

$$\mathcal{H}_\rho(X) := \langle C, X \rangle + \rho \sum_{i \in \mathcal{F}} (\langle A_i, X \rangle - b_i)^2$$

Let $\tilde{X}$ be the minimizer of the penalized problem.

$$\tilde{X} := \underset{X \in \mathbb{R}^{d \times d}}{\arg\min} \mathcal{L}_\rho(X)$$

Let $X^*$ be the minimizer of the constrained problem equation 87. Let $\tau_i$ be the error $\tilde{X}$ has in satisfying constraint $\langle A_i, \tilde{X} \rangle = b_i$.

$$\tau_i := |\langle A_i, \tilde{X} \rangle - b_i|$$

We will show that $\tau_i$ scales inversely with $\rho$. That is, $\tau_i \le poly(k, \rho^{-1})$. Notice that the quadratic penalty on the violated constraints must be smaller than the decrease in the objective for having violated the constraints. So long as the objective is not too sensitive i.e 'robust' to perturbations in the constraint violations the quadratic penalty should overwhelm the decrease in the objective. To carry out this intuition, we begin with the fact that the constrained minimum is larger than the penalized minimum.

$$\mathcal{H}_\rho(X^*) - \mathcal{H}_\rho(\tilde{X}) \le 0 \tag{77}$$

Applying the definitions of $\mathcal{H}_\rho(X^*)$ and $\mathcal{H}_\rho(\tilde{X})$ we obtain

$$\langle C, X^* \rangle - \left( \langle C, \tilde{X} \rangle + \rho \sum_{i \in \mathcal{F}} \tau_i^2 \right) \le 0 \tag{78}$$

Rearranging LHS and RHS and dividing by $|\mathcal{F}|$ we obtain

$$\rho \frac{1}{|\mathcal{F}|} \sum_{i \in \mathcal{F}} \tau_i^2 \le \frac{1}{|\mathcal{F}|} \langle C, \tilde{X} - X^* \rangle \tag{79}$$

We upper bound the RHS using the robustness theorem of Raghavendra & Steurer (2009a) restated in the appendix Theorem F.2 which states that an SDP solution that violates the constraints by a small perturbation changes the objective by a small amount. Thus we obtain,

$$RHS \le \left( \frac{1}{|\mathcal{F}|} \sum_{i \in \mathcal{F}} \tau_i \right)^{1/2} poly(k) \tag{80}$$

Therefore combining equation 79 with equation 80 we obtain

$$\rho \frac{1}{|\mathcal{F}|} \sum_i \tau_i^2 \le \left( \frac{1}{|\mathcal{F}|} \sum_i \tau_i \right)^{1/2} poly(k)$$

Taking Cauchy-Schwarz on the RHS we obtain

$$\rho \frac{1}{|\mathcal{F}|} \sum_i \tau_i^2 \leq \left( \frac{1}{|\mathcal{F}|} \sum_i \tau_i^2 \right)^{1/4} poly(k)$$

Noticing that $\frac{1}{|\mathcal{F}|} \sum_i \tau_i^2$ appears in both LHS and RHS we rearrange to obtain

$$\frac{1}{|\mathcal{F}|} \sum_i \tau_i^2 \leq \frac{poly(k)}{\rho^{4/3}} \leq \epsilon$$

Where for $\rho = \frac{poly(k)}{\epsilon^{3/4}}$ we have the average squared error of the constraints is upper bounded by $\epsilon$. Then via Markov's no more than $\sqrt{\epsilon}$ fraction of the constraints can be violated by more than squared error $\sqrt{\epsilon}$. We label these 'grossly' violated constraints. The clauses involved in these grossly violated constraints contributes no more than $poly(k)\sqrt{\epsilon}$ to the objective. On the other hand, the constraints that are violated by no more than squared error $\sqrt{\epsilon}$ contributed no more than $poly(k)\epsilon^{1/8}$ to the objective which follows from the robustness theorem Theorem F.2. Taken together we conclude that

$$\mathcal{L}_\rho(\tilde{\mathbf{v}}) - \text{SDP}(\Lambda) \leq \epsilon$$

For $\rho = poly(k, \epsilon^{-1})$ as desired. $\qquad \square$

Finally we show it's not hard to generalize our algorithm to alphabets of size $[q]$.

**Notation for General Alphabet.** For any predicate $P \in \mathcal{P}$, let $\mathcal{D}(P)$ be the set of all variable assignment tuples indexed by a set of variables $s \subseteq \mathcal{S}(P)$ and an assignment $\tau \in [q]^{|s|}$. Let $x_{(i,a)}$ denote an assignment of value $a \in [q]$ to variable $x_i$.

---

**SDP 3** SDP Vector Formulation for Max-k-CSP General Alphabet (Equivalent to UGC optimal)

---

SDP Vector Formulation General Alphabet $\Lambda = (\mathcal{V}, \mathcal{P}, q)$.
Pseudoexpectation formulation of the objective.

$$\min_{x_1, x_2, \ldots, x_N} \sum_{P_z \subset \mathcal{P}} \tilde{\mathbb{E}}_\mu[-P_z(X_z)] \tag{81}$$

subject to: $\tilde{\mathbb{E}}_\mu[(x_{(i,a)}^2 - x_{(i,a)}) \prod_{(j,b) \in \phi} x_{(j,b)}] = 0 \quad \forall i \in \mathcal{V}, \forall a \in [q], \forall \phi \subseteq \mathcal{D}(P), \forall P \in \mathcal{P}$

$$\tag{82}$$

$$\tilde{\mathbb{E}}_\mu[(\sum_{a \in [q]} x_{ia} - 1) \prod_{(j,b) \in \phi} x_{(j,b)}] = 0 \quad \forall i \in \mathcal{V}, \forall \phi \subseteq \mathcal{D}(P), \forall P \in \mathcal{P}, \tag{83}$$

$$\tilde{\mathbb{E}}_\mu[x_{(i,a)} x_{(i,a')} \prod_{(j,b) \in \phi} x_{(j,b)}] = 0 \quad \forall i \in \mathcal{V}, \forall a \neq a' \in [q], \forall \phi \subseteq \mathcal{D}(P), \forall P \in \mathcal{P},$$

$$\tag{84}$$

$$\tilde{\mathbb{E}}[SoS_{2kq}(X_\phi)] \geq 0 \quad \forall \phi \subseteq \mathcal{D}(P), \forall P \in \mathcal{P}, \tag{85}$$

$$\tilde{\mathbb{E}}[SoS_2(\mathbf{x})] \geq 0. \tag{86}$$

First constraint corresponds to booleanity of each value in the alphabet.
Second constraint corresponds to a variable taking on only one value in the alphabet.
Third constraint corresponds to a variable taking on only one value in the alphabet.
Fourth constraint corresponds to local distribution on the variables in each predicate.
Fifth constraint corresponds to the positivity of every degree two sum of squares of polynomials.

---

**Lemma C.4.** There exists a message passing algorithm that computes in $poly(\epsilon^{-1}, \Phi, \log(\delta^{-1}))$ iterations a set of vectors $\mathbf{v} := \{\hat{v}_{(i,a)}\}$ for all $(i, a) \in \phi$, for all $\phi \subseteq \mathcal{D}(P)$, for all $P \in \mathcal{P}$ that satisfy

the constraints of Algorithm 3 to error $\epsilon$ and approximates the optimum of Algorithm 3 to error $\epsilon$ with probability $1 - \delta$

$$\left| \sum_{P_z \in \mathcal{P}} \tilde{\mathbb{E}}_{\hat{\mu}}[P_z(X_z)] - \text{SDP}(\Lambda) \right| \leq \epsilon$$

where $\text{SDP}(\Lambda)$ is the optimum of Algorithm 3.

*Proof.* The proof is entirely parallel to the proof of Theorem C.1. We can write Algorithm 3 entirely in terms of the vector of its cholesky decomposition where once again we take advantage of the fact that SoS degree $2kq$ distributions are actual distributions over subsets of $kq$ variables over each predicate. Given the overparameterized vector formulation, we observe that once again we are faced with equality constraints that can be added to the objective with a quadratic penalty. Perturbed gradient descent induces a message passing algorithm over the constraint graph $G_\Lambda$, and in no more than $poly(\epsilon^{-1}\Phi)$ iterations reaches an $(\epsilon, \gamma)$-SOSP. The analysis of optimality goes along the same lines as Lemma E.1. For sufficiently large penalty $\rho = poly(\epsilon^{-1}, q^k)$ the error in satisfying the constraints is $\epsilon$ and the objective is robust to small perturbations in satisfying the constraint. That concludes our discussion of generalizing to general alphabets. $\qquad\square$

## C.1  Neural Certification Scheme

An intriguing aspect of OptGNN is that the embeddings can be interpreted as the solution to a low rank SDP which leaves open the possibility that the embeddings can be used to generate a dual certificate i.e a lower bound on a convex relaxation. First, we define the primal problem

$$\text{Minimize:} \quad \langle C, X \rangle \tag{87}$$
$$\text{Subject to:} \quad \langle A_i, X \rangle = b_i \quad \forall i \in [\mathcal{F}] \tag{88}$$
$$X \succeq 0 \tag{89}$$

**Lemma C.5.** Let OPT be the minimizer of the SDP equation 87. Then for any $\tilde{X} \in \mathbb{R}^{N \times N} \succeq 0$ and any $\lambda^* \in \mathbb{R}^{|\mathcal{F}|}$, we define $F_{\lambda^*}(X)$ to be

$$F_{\lambda^*}(\tilde{X}) := \langle C, \tilde{X} \rangle + \sum_{i \in \mathcal{F}} \lambda_i^* \left( \langle A_i, \tilde{X} \rangle - b_i \right)$$

We require SDP to satisfy a bound on its trace $Tr(X) \leq \mathcal{Y}$ for some $\mathcal{Y} \in \mathbb{R}^+$. Then the following is a lower bound on OPT.

$$OPT \geq F_{\lambda^*}(\tilde{X}) - \langle \nabla F_{\lambda^*}(\tilde{X}), \tilde{X} \rangle + \lambda_{\min}\left( \nabla F_{\lambda^*}(\tilde{X}) \right) \mathcal{Y}$$

*Proof.* Next we introduce lagrange multipliers $\lambda \in \mathbb{R}^k$ and $Q \succeq 0$ to form the lagrangian

$$\mathcal{L}(\lambda, Q, X) = \langle C, X \rangle + \sum_{i \in \mathcal{F}} \lambda_i \left( \langle A_i, X \rangle - b_i \right) - \langle Q, X \rangle$$

We lower bound the optimum of OPT defined to be the minimizer of equation 87

$$OPT := \min_{X \succeq 0} \max_{\lambda \in \mathbb{R}, Q \succeq 0} \mathcal{L}(\lambda, Q, X) \geq \min_{V \in \mathbb{R}^{N \times N}} \max_{\lambda} \langle C, VV^T \rangle + \sum_{i \in \mathcal{F}} \lambda_i \left( \langle A_i, VV^T \rangle - b_i \right) \tag{90}$$

$$= \max_{\lambda} \min_{V \in \mathbb{R}^{N \times N}} \langle C, VV^T \rangle + \sum_{i \in \mathcal{F}} \lambda_i \left( \langle A_i, VV^T \rangle - b_i \right) \tag{91}$$

$$= \min_{V \in \mathbb{R}^{N \times N}} \langle C, VV^T \rangle + \sum_{i \in \mathcal{F}} \lambda_i^* \left( \langle A_i, VV^T \rangle - b_i \right). \tag{92}$$

Where in the first inequality we replaced $X \succeq 0$ with $VV^T$ which is a lower bound as every psd matrix admits a cholesky decomposition. In the second inequality we flipped the order of min and max, and in the final inequality we chose a specific set of dual variables $\lambda^* \in \mathbb{R}^{|\mathcal{F}|}$ which lower bounds the maximization over dual variables. The key is to find a good setting for $\lambda^*$.

Next we establish that for any choice of $\lambda^*$ we can compute a lower bound on inequality 92 as follows. Let $F_{\lambda^*}(VV^T)$ be defined as the funciton in the RHS of 92.

$$F_{\lambda^*}(VV^T) := \langle C, X \rangle + \sum_{i \in \mathcal{F}} \lambda_i^* \left( \langle A_i, X \rangle - b_i \right)$$

Then equation 92 can be rewritten as

$$OPT \geq \min_{V \in \mathbb{R}^{N \times N}} F_{\lambda^*}(VV^T) := \langle C, X \rangle + \sum_{i \in \mathcal{F}} \lambda_i^* \left( \langle A_i, X \rangle - b_i \right)$$

Now let $V^*$ be the minimizer of equation 92 and let $X^* = V^*(V^*)^T$. We have by convexity that

$$F_{\lambda^*}(X) - F_{\lambda^*}(X^*) \leq \langle \nabla F_{\lambda^*}(X), X - X^* \rangle = \langle \nabla F_{\lambda^*}(X), X \rangle + \langle -\nabla F_{\lambda^*}(X), X^* \rangle \quad (93)$$

$$\leq \langle \nabla F_{\lambda^*}(X), X \rangle - \lambda_{min}\left(\nabla F_{\lambda^*}(X)\right) \text{Tr}(X^*) \quad (94)$$

$$\leq \langle \nabla F_{\lambda^*}(X), X \rangle - \lambda_{min}(\nabla F_{\lambda^*}(X))N \quad (95)$$

In the first inequality we apply the convexity of $F_{\lambda^*}$. In the second inequality we apply a standard inequality of frobenius inner product. In the last inequality we use the fact that $\text{Tr}(X^*) = N$. Rearranging we obtain for any $X$

$$OPT \geq F_\lambda(X^*) \geq F_{\lambda^*}(X) - \langle \nabla F_{\lambda^*}(X), X \rangle + \lambda_{min}\left(\nabla F_{\lambda^*}(X)\right)N \quad (96)$$

Therefore it suffices to upper bound the two terms above $\langle \nabla F_{\lambda^*}(X), X \rangle$ and $\lambda_{min}(\nabla F_{\lambda^*}(X))$ which is an expression that holds for any $X$. Given the output embeddings $\tilde{V}$ of OptGNN (or indeed any set of vectors $\tilde{V}$) let $\tilde{X} = \tilde{V}\tilde{V}^T$. Then we have concluded

$$OPT \geq F_\lambda(X^*) \geq F_{\lambda^*}(\tilde{X}) - \langle \nabla F_{\lambda^*}(\tilde{X}), \tilde{X} \rangle + \lambda_{min}(\nabla F_{\lambda^*}(\tilde{X}))N \quad (97)$$

as desired. $\qquad\square$

Up to this point, every manipulation is formal proof. Subsequently we detail how to estimate the dual variables $\lambda^*$. Although any estimate will produce a bound, it won't produce a tight bound. To be clear, solving for the optimal $\lambda^*$ would be the same as building an SDP solver which would bring us back into the expensive primal dual procedures that are involved in solving SDP's. We are designing quick and cheap ways to output a dual certificate that may be somewhat looser. Our scheme is simply to set $\lambda^*$ such that $\|\nabla F_{\lambda^*}(\tilde{X})\|$ is minimized, ideally equal to zero. The intuition is that if $(\tilde{X}, \lambda^*)$ were a primal dual pair, then the lagrangian would have a derivative with respect to $X$ evaluated at $\tilde{X}$ would be equal to zero. Let $H_\lambda(V)$ be defined as follows

$$H_{\lambda^*}(\tilde{V}) := \langle C, \tilde{V}\tilde{V}^T \rangle + \sum_{i \in \mathcal{F}} \lambda_i^* \left( \langle A_i, \tilde{V}\tilde{V}^T \rangle - b_i \right)$$

We know the gradient of $H_\lambda(\tilde{V})$

$$\nabla H_\lambda(\tilde{V}) = 2 \left( C + \sum_{i \in \mathcal{F}} \lambda_i^* A_i \right) \tilde{V} = 2 \nabla F_\lambda(\tilde{V}\tilde{V}^T)\tilde{V}$$

Therefore it suffices to find a setting of $\lambda^*$ such that $\|\nabla F_\lambda(\tilde{X})\tilde{V}\|$ is small, ideally zero. This would be a simple task, indeed a regression, if not for the unfortunate fact that OptGNN explicitly projects the vectors in $\tilde{V}$ to be unit vectors. This creates numerical problems such that minimizing the norm of $\|\nabla F_\lambda(\tilde{X})\tilde{V}\|$ does not produce a $\nabla F_\lambda(\tilde{X})$ with a large minimum eigenvalue.

To fix this issue, let $R_{\eta,\rho}(V)$ denote the penalized lagrangian with quadratic penalties for constraints of the form $\langle A_i, X \rangle = b_i$ and linear penalty $\eta_i$ for constraints along the main diagonal of $X$ of the form $\langle e_i e_i^T, X \rangle = 1$.

$$R_{\eta,\rho}(V) := \langle C, VV^T \rangle + \sum_{i \in \mathcal{J}} \rho(\langle A_i, VV^T \rangle - b_i)^2 + \sum_{i=1}^{N} \eta_i(\langle e_i e_i^T, VV^T \rangle - 1)$$

Taking the gradient of $R_{\eta,\rho}(V)$ we obtain

$$\nabla R_{\eta,\rho}(V) := 2CV + \sum_{i \in \mathcal{J}} 2\rho(\langle A_i, VV^T \rangle - b_i)A_i V + \sum_{i=1}^{N} 2\eta_i e_i e_i^T V$$

Our rule for setting dual variables $\delta_i$ for $i \in \mathcal{J}$ is

$$\delta_i := 2\rho \left( \langle A_i, \tilde{V}\tilde{V}^T \rangle - b_i \right)$$

our rule for setting dual variables $\eta_j$ for $j \in [N]$ is

$$\eta_j := \frac{1}{2} \left\| e_j^T \left( C + \sum_{i \in \mathcal{F}} 2\rho(\langle A_i, VV^T \rangle - b_i)A_i \right) V \right\|$$

Then our full set of dual variables $\lambda^*$ is simply the concatenation $(\delta, \eta)$. Writing out everything explicitly we obtain the following matrix for $\nabla F_{\lambda^*}(\tilde{V}\tilde{V}^T)$

$$\nabla F_\lambda(\tilde{V}\tilde{V}^T) = C + \sum_{i \in \mathcal{F}} \rho(\langle A_i, \tilde{V}\tilde{V}^T \rangle - b_i)A_i + \sum_{j \in [N]} \frac{1}{2} \left\| e_j^T \left( C + \sum_{i \in \mathcal{F}} 2\rho(\langle A_i, \tilde{V}\tilde{V}^T \rangle - b_i)A_i \right) \tilde{V} \right\| e_i e_i^T$$

Plugging this expression into Lemma C.5 the final bound we evaluate in our code is

$$OPT \geq \langle C, \tilde{V}\tilde{V}^T \rangle + \sum_{i \in \mathcal{F}} 2\rho(\langle A_i, \tilde{V}\tilde{V}^T \rangle - b_i)^2$$

$$- \left\langle C + \sum_{i \in \mathcal{F}} \rho(\langle A_i, \tilde{V}\tilde{V}^T \rangle - b_i)A_i + \sum_{j \in [N]} \frac{1}{2} \left\| e_j^T \left( C + \sum_{i \in \mathcal{F}} 2\rho(\langle A_i, \tilde{V}\tilde{V}^T \rangle - b_i)A_i \right) \tilde{V} \right\| e_i e_i^T, \tilde{V}\tilde{V}^T \right\rangle$$

$$+ \lambda_{\min} \left( C + \sum_{i \in \mathcal{F}} \rho(\langle A_i, \tilde{V}\tilde{V}^T \rangle - b_i)A_i + \sum_{j \in [N]} \frac{1}{2} \left\| e_j^T \left( C + \sum_{i \in \mathcal{F}} 2\rho(\langle A_i, \tilde{V}\tilde{V}^T \rangle - b_i)A_i \right) \tilde{V} \right\| e_i e_i^T \right) N \tag{98}$$

Which is entirely computed in terms of $\tilde{V}$ the output embeddings of OptGNN. The resulting plot is as follows.

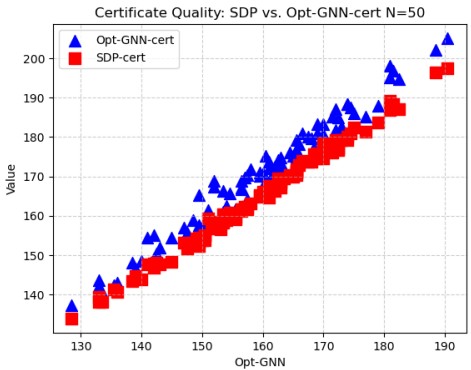

Figure 4: N=50 p=0.1 SDP vs OptGNN Dual Certificate

**Note:** The reason for splitting the set of dual variables is because the projection operator onto the unit ball is hard coded into the architecture of the lift network. Satisfying the constraint set via projection is different from the soft quadratic penalties on the remaining constraints and require separate handling.

**Max-Cut Certificate**  For Max-Cut our dual variables are particularly simple as there are no constraints $\langle A_i, X \rangle = b_i$ for $b_i \neq 0$. The dual variables for Max-Cut take on the form for all $i \in [N]$

$$\lambda_i^* = \frac{1}{2} \left\| \sum_{j \in N(i)} w_{ij} v_j \right\|$$

It's certainly possible to come up with tighter certification schemes which we leave to future work.

**Intuition:**  Near global optimality one step of the augmented method of lagrange multipliers ought to closely approximate the dual variables. After obtaining a estimate for the penalized lagrange multipliers we estimate the lagrange multipliers for the norm constraint by approximating $\nabla R_\lambda(V) = 0$. The alternative would have been to solve the linear system for all the lagrange multipliers at once but this runs into numerical issues and degeneracies.

**Certificate Experiment:**  We run our certification procedure which we name OptGNN-cert and compare it to the SDP certificate. Note, that mathematically we will always produce a larger (i.e inferior) dual certificate in comparison to the SDP because we are bounding the distance to the SDP optimum with error in the gradients and hessians of the output embeddings of OptGNN. Our advantage is in the speed of the procedure. Without having to go through a primal dual solver, the entire time of producing OptGNN-cert is in the time required to feedforward through OptGNN. In this case we train an OptGNN-Max-Cut with 10 layers, on 1000 Erdos-Renyi graphs, with $N = 100$ nodes and edge density $p = 0.1$. We plot the OptGNN Max-Cut value (an actual integer cut) on the x-axis and in the y-axis we plot the dual certificate value on the same graph where we compare the SDP certificate with the OptGNN-cert. See 4 for the $N = 50$ graphs and 3 for the $N = 100$ graphs.

Note of course the dual certificate for any technique must be larger than the cut value outputted by OptGNN so the scatter plot must be above the $x = y$ axis of the plot. We see as is mathematically necessary, the OptGNN-cert is not as tight as the SDP certificate but certainly competitive and more importantly it is arrived at dramatically faster. Without any runtime optimizations, the OptGNN feedforward and certification takes no more than 0.02 seconds whereas the SDP takes 0.5 seconds on $N = 100$ node graphs.

## D  Experiment details

In this section we give further details on our experimental setup.

**Datasets.**  Our experiments span a variety of randomly generated and real-world datasets. Our randomly generated datasets contain graphs from several random graph models, in particular Erdős-Rényi (with $p = 0.15$), Barabási–Albert (with $m = 4$), Holme-Kim (with $m = 4$ and $p = 0.25$), Watts-Strogatz (with $k = 4$ and $p = 0.25$), and forced RB (with two sets of parameters, RB200 and RB500). Our real-world datasets are ENZYMES, PROTEINS, MUTAG, IMDB-BINARY, COLLAB (which we will together call **TU-small**), and REDDIT-BINARY, REDDIT-MULTI-5K, and REDDIT-MULTI-12K (which we will call **TU-REDDIT**).

We abbreviate the generated datasets using their initials and the range of vertex counts. For example, by ER (50,100) we denote Erdős-Rényi random graphs with a vertex count drawn uniformly at random from [50, 100]. In tables, we mark generated datasets with superscript [a], **TU-small** with [b], and **TU-REDDIT** with [c].

For Figure 2b, we display results for forced RB instances drawn from two different distributions. For RB200, we select $N$ uniformly in $[6, 15]$ and $K$ uniformly in $[12, 21]$. For RB500, we select $N$ uniformly in $[20, 34]$ and $K$ uniformly in $[10, 29]$.

In Table 2, random 3-SAT instances are generated by drawing three random variables for each clause and negating each variable with $p = 0.5$. We trained on instances with 100 variables and clause count drawn uniformly from the interval $[400, 430]$, and tested on instances with 100 variables and 400, 415, and 430 clauses respectively.

**Baselines.**  We compare the performance of our approach against known classical algorithms. In terms of classical baselines, we run Gurobi with varying timeouts and include SDP results on smaller datasets.

| Parameter | ER, BA, WS, HK | RB200 | RB500 | 3-SAT | **TU-small** | **TU-REDDIT** |
|---|---|---|---|---|---|---|
| Gradient steps | 20,000 | 200,000 | 100,000 | 100,000 | 100,000 | 100,000 |
| Validation freq | 1,000 | 2,000 | 2,000 | 2,000 | 1,000 | 2,000 |
| Batch size | 16 | 32 | 32 | 32 | 16 | 16 |
| Ranks | 4, 8, 16, 32 | 32, 64 | 32, 64 | 32, 64 | 4, 8, 16, 32 | 4, 8, 16, 32 |
| Layer counts | 8, 16 | 16 | 16 | 16 | 8, 16 | 8, 16 |
| Positional encodings | RW | none | none | none | LE, RW | RW |
| **Run count** | 8 | 2 | 2 | 2 | 16 | 8 |

Figure 5: Hyperparameter range explored for each group of datasets. For each NN architecture, when training on a dataset, we explored every listed hyperparameter combination in the corresponding column.

We also include a greedy baseline, which is the function `one_exchange` (for Max-Cut) or `min_weighted_vertex_cover` (for Min-Vertex-Cover) from `networkx` (Hagberg et al., 2008).

**Validation and test splits.** For each dataset we hold out a validation and test slice for evaluation. In our generated graph experiments we set aside 1000 graphs each for validation and testing. Each step of training ran on randomly generated graphs. For **TU-small**, we used a train/validation/test split of 0.8/0.1/0.1. For **TU-REDDIT**, we set aside 100 graphs each for validation and testing.

**Scoring.** To measure a model's score on a graph, we first run the model on the graph with a random initial vector assignment to generate an SDP output, and then round this output to an integral solution using 1,000 random hyperplanes. For the graph, we retain the best score achieved by any hyperplane.

We ran validation periodically during each training run and retained the model that achieved the highest validation score. Then for each model and dataset, we selected the hyperparameter setting that achieved the highest validation score, and we report the average score measured on the test slice. Please see D for further details on the hyperparameter ranges used.

In Figure 2b and Figure 2a, at test time, we use 100 random initial vector assignments instead of just 1, and retain the best score achieved by any hyperplane on any random initial vector assignment. We use 1 random initial vector assignment for validation as in other cases.

**Hardware.** Our training runs used 20 cores of an Intel Xeon Gold 6248 (for data loading and random graph generation) and a NVIDIA Tesla V100 GPU. Our Gurobi runs use 8 threads on a Intel Xeon Platinum 8260. Our KaMIS runs use an Intel Core i9-13900H. Our LwD and DGL-TREESEARCH runs use an Intel Core i9-13900H and an RTX 4060.

**Hyperparameters.** We ran each experiment on a range of hyperparameters. See Figure 5 for the hyperparameter listing. For all training runs, we used the Adam optimizer Kingma & Ba (2014) with a learning rate of 0.001. We used Laplacian eigenvector Dwivedi et al. (2020) (LE) or random walk Dwivedi et al. (2021) (RW) positional encoding with dimensionality of half the rank, except for rank 32 where we used 8 dimensions. For Max-3-SAT, we set the penalty term $\rho = 0.003$.

### D.1 Max-Cut

**Low-rank SDP.** We have included an additional baseline that solves a low-rank SDP using coordinate descent (Wang & Kolter, 2019) to the Max-Cut benchmark that originally appears in Any-CSP (Tönshoff et al., 2022). We have used the publicly available implementation provided by the authors. For each instance, the rank is automatically selected by the package to be $\lceil \sqrt{2N} \rceil$.

### D.2 Max-3-SAT

**ErdősGNN**. The ErdősGNN baseline accepts as input a clause-variable bipartite graph of the SAT formula and the node degrees as input attributes. Each variable $x_i$ is set to TRUE with probability $p_i$. A graph neural network (GatedGCNN) is trained until convergence ($\sim$50k iterations) with randomly generated formulae with clause to variable ratiot in the range $[4, 4.3]$. The neural network is trained to produce output probabilities $p_i$ by minimizing the expected number of unsatisfied clauses. The exact closed form of the expectation can be found in (Karalias, 2023). At inference time, the probabilities are rounded to binary assignments using the method of conditional expectation.

**Survey propagation**. We report the results of survey propagation Braunstein et al. (2005). Typically, algorithms like survey propagation are accompanied by a local search algorithm like WalkSAT. Here we report the results obtained by the algorithm without running any local search on its output.

**Low-rank SDP**. This is identical to the baseline that we used for Max-3-SAT (see Max-Cut section above).

**WalkSAT.** We use a publicly available python implementation of WalkSAT and we set the max number of variable flips to $4 \times$ number of variables, for a total of 400 flips on the instances we tested.

**Autograd.** The autograd comparison measures the performance of autograd on the same loss as OptGNN. Starting with some initial node embeddings for the instance, the Lagrangian is computed for the problem and the node embeddings are updated by minimizing the Lagrangian using Adam. This process is run for several iterations. After it is concluded, the vectors are rounded with hyperplane rounding, yielding a binary assignment for the variables of the formula.

**Autograd parameters.** We use the Adam optimizer with learning rate 0.1 for 1000 epochs with penalty 0.01 and round with 10,000 hyperplanes, which is 10 times as many as that used by OptGNN.

## D.3    Additional Results

### D.3.1    Comparisons with Gurobi and Greedy

Table 4 and Table 3 contain comparisons on additional datasets with Gurobi and Greedy.

| Dataset | OptGNN | Greedy | Gurobi 0.1s | Gurobi 1.0s | Gurobi 8.0s |
|---|---|---|---|---|---|
| BA[a] (50,100) | 42.88 (27) | 51.92 | 42.82 | 42.82 | 42.82 |
| BA[a] (100,200) | 83.43 (25) | 101.42 | 83.19 | 83.19 | 83.19 |
| BA[a] (400,500) | 248.74 (27) | 302.53 | 256.33 | 246.49 | 246.46 |
| ER[a] (50,100) | 55.25 (21) | 68.85 | 55.06 | 54.67 | 54.67 |
| ER[a] (100,200) | 126.52 (18) | 143.51 | 127.83 | 123.47 | 122.76 |
| ER[a] (400,500) | 420.70 (41) | 442.84 | 423.07 | 423.07 | 415.52 |
| HK[a] (50,100) | 43.06 (25) | 51.38 | 42.98 | 42.98 | 42.98 |
| HK[a] (100,200) | 84.38 (25) | 100.87 | 84.07 | 84.07 | 84.07 |
| HK[a] (400,500) | 249.26 (27) | 298.98 | 247.90 | 247.57 | 247.57 |
| WC[a] (50,100) | 46.38 (26) | 72.55 | 45.74 | 45.74 | 45.74 |
| WC[a] (100,200) | 91.28 (21) | 143.70 | 89.80 | 89.80 | 89.80 |
| WC[a] (400,500) | 274.21 (31) | 434.52 | 269.58 | 269.39 | 269.39 |
| MUTAG[b] | 7.79 (18) | 12.84 | 7.74 | 7.74 | 7.74 |
| ENZYMES[b] | 20.00 (24) | 27.35 | 20.00 | 20.00 | 20.00 |
| PROTEINS[b] | 25.29 (18) | 33.93 | 24.96 | 24.96 | 24.96 |
| IMDB-BIN[b] | 16.78 (18) | 17.24 | 16.76 | 16.76 | 16.76 |
| COLLAB[b] | 67.50 (23) | 71.74 | 67.47 | 67.46 | 67.46 |
| REDDIT-BIN[c] | 82.85 (38) | 117.16 | 82.81 | 82.81 | 82.81 |
| REDDIT-M-12K[c] | 81.55 (25) | 115.72 | 81.57 | 81.52 | 81.52 |
| REDDIT-M-5K[c] | 107.36 (33) | 153.24 | 108.73 | 107.32 | 107.32 |

Table 3: Performance of OptGNN, Greedy, and Gurobi 0.1s, 1s, and 8s on Min-Vertex-Cover. For each approach and dataset, we report the average Vertex-Cover size measured on the test slice. Here, lower score is better. In parentheses, we include the average runtime in *milliseconds* for OptGNN.

### D.3.2    Ratio tables

In Figure 6 and Figure 7 we supply the performance of OptGNN as a ratio against the integral value achieved by Gurobi running with a time limit of 8 seconds. These tables include the standard deviation in the ratio. We note that for Maximum Cut, OptGNN comes within 1.1% of the Gurobi 8s value, and for Min-Vertex-Cover, OptGNN comes within 3.1%.

| Dataset | OptGNN | Greedy | Gurobi 0.1s | Gurobi 1.0s | Gurobi 8.0s |
|---|---|---|---|---|---|
| BA[a] (50,100) | 351.49 (18) | 200.10 | 351.87 | 352.12 | 352.12 |
| BA[a] (100,200) | 717.19 (20) | 407.98 | 719.41 | 719.72 | 720.17 |
| BA[a] (400,500) | 2197.99 (66) | 1255.22 | 2208.11 | 2208.11 | 2212.49 |
| ER[a] (50,100) | 528.95 (18) | 298.55 | 529.93 | 530.03 | 530.16 |
| ER[a] (100,200) | 1995.05 (24) | 1097.26 | 2002.88 | 2002.88 | 2002.93 |
| ER[a] (400,500) | 16387.46 (225) | 8622.34 | 16476.72 | 16491.60 | 16495.31 |
| HK[a] (50,100) | 345.74 (18) | 196.23 | 346.18 | 346.42 | 346.42 |
| HK[a] (100,200) | 709.39 (23) | 402.54 | 711.68 | 712.26 | 712.88 |
| HK[a] (400,500) | 2159.90 (61) | 1230.98 | 2169.46 | 2169.46 | 2173.88 |
| WC[a] (50,100) | 198.29 (18) | 116.65 | 198.74 | 198.74 | 198.74 |
| WC[a] (100,200) | 389.83 (24) | 229.43 | 390.96 | 392.07 | 392.07 |
| WC[a] (400,500) | 1166.47 (78) | 690.19 | 1173.45 | 1175.97 | 1179.86 |
| MUTAG[b] | 27.95 (9) | 16.95 | 27.95 | 27.95 | 27.95 |
| ENZYMES[b] | 81.37 (14) | 48.53 | 81.45 | 81.45 | 81.45 |
| PROTEINS[b] | 102.15 (12) | 60.74 | 102.28 | 102.36 | 102.36 |
| IMDB-BIN[b] | 97.47 (11) | 51.85 | 97.50 | 97.50 | 97.50 |
| COLLAB[b] | 2622.41 (22) | 1345.70 | 2624.32 | 2624.57 | 2624.62 |
| REDDIT-BIN[c] | 693.33 (186) | 439.79 | 693.02 | 694.10 | 694.14 |
| REDDIT-M-12K[c] | 568.00 (89) | 358.40 | 567.71 | 568.91 | 568.94 |
| REDDIT-M-5K[c] | 786.09 (133) | 495.02 | 785.44 | 787.48 | 787.92 |

Table 4: Performance of OptGNN, Greedy, and Gurobi 0.1s, 1s, and 8s on Maximum Cut. For each approach and dataset, we report the average cut size measured on the test slice. Here, higher score is better. In parentheses, we include the average runtime in *milliseconds* for OptGNN.

| Dataset | OptGNN |
|---|---|
| BA[a] (50,100) | 0.998 ± 0.002 |
| BA[a] (100,200) | 0.996 ± 0.003 |
| BA[a] (400,500) | 0.993 ± 0.003 |
| ER[a] (50,100) | 0.998 ± 0.002 |
| ER[a] (100,200) | 0.996 ± 0.002 |
| ER[a] (400,500) | 0.993 ± 0.001 |
| HK[a] (50,100) | 0.998 ± 0.002 |
| HK[a] (100,200) | 0.995 ± 0.003 |
| HK[a] (400,500) | 0.994 ± 0.003 |
| WC[a] (50,100) | 0.998 ± 0.003 |

| Dataset | OptGNN |
|---|---|
| WC[a] (100,200) | 0.995 ± 0.003 |
| WC[a] (400,500) | 0.989 ± 0.003 |
| MUTAG[b] | 1.000 ± 0.000 |
| ENZYMES[b] | 0.999 ± 0.003 |
| PROTEINS[b] | 1.000 ± 0.002 |
| IMDB-BIN[b] | 1.000 ± 0.001 |
| COLLAB[b] | 0.999 ± 0.002 |
| REDDIT-BIN[c] | 1.000 ± 0.001 |
| REDDIT-M-12K[c] | 0.999 ± 0.002 |
| REDDIT-M-5K[c] | 0.999 ± 0.002 |

Figure 6: Performance of OptGNN on Max-Cut compared to Gurobi running under an 8 second time limit, expressed as a ratio. For each dataset, we take the ratio of the integral values achieved by OptGNN and Gurobi 8s on each of the graphs in the test slice. We present the average and standard deviation of these ratios. Here, higher is better. This table demonstrates that OptGNN achieves nearly the same performance, missing on average 1.1% of the cut value in the worst measured case.

| Dataset | OptGNN |
|---|---|
| BA[a] (50,100) | $1.001 \pm 0.005$ |
| BA[a] (100,200) | $1.003 \pm 0.005$ |
| BA[a] (400,500) | $1.008 \pm 0.011$ |
| ER[a] (50,100) | $1.010 \pm 0.015$ |
| ER[a] (100,200) | $1.031 \pm 0.012$ |
| ER[a] (400,500) | $1.013 \pm 0.006$ |
| HK[a] (50,100) | $1.002 \pm 0.007$ |
| HK[a] (100,200) | $1.004 \pm 0.013$ |
| HK[a] (400,500) | $1.007 \pm 0.011$ |
| WC[a] (50,100) | $1.014 \pm 0.016$ |

| Dataset | OptGNN |
|---|---|
| WC[a] (100,200) | $1.016 \pm 0.013$ |
| WC[a] (400,500) | $1.018 \pm 0.007$ |
| MUTAG[b] | $1.009 \pm 0.027$ |
| ENZYMES[b] | $1.000 \pm 0.000$ |
| PROTEINS[b] | $1.010 \pm 0.021$ |
| IMDB-BIN[b] | $1.002 \pm 0.016$ |
| COLLAB[b] | $1.001 \pm 0.003$ |
| REDDIT-BIN[c] | $1.000 \pm 0.002$ |
| REDDIT-M-12K[c] | $1.000 \pm 0.001$ |
| REDDIT-M-5K[c] | $1.000 \pm 0.001$ |

Figure 7: Performance of OptGNN on Min-Vertex-Cover compared to Gurobi running under an 8 second time limit, expressed as a ratio. For each dataset, we take the ratio of the integral values achieved by OptGNN and Gurobi 8s on each of the graphs in the test slice. We present the average and standard deviation of these ratios. Here, lower is better. This table demonstrates that OptGNN achieves nearly the same performance, producing a cover on average 3.1% larger than Gurobi 8s in the worst measured case.

## D.4 Model ablation study

Here we provide the evaluations of several models that were trained on the same loss as OptGNN. We see that OptGNN consistently achieves the best performance among different neural architectures. Note that while OptGNN was consistently the best model, other models were able to perform relatively well; for instance, GatedGCNN achieves average cut values within a few percent of OptGNN on nearly all the datasets (excluding COLLAB). This points to the overall viability of training using an SDP relaxation for the loss function.

| Dataset | GAT | GCNN | GIN | GatedGCNN | OptGNN |
|---|---|---|---|---|---|
| ER[a] (50,100) | 525.92 (25) | 500.94 (17) | 498.82 (14) | 526.78 (14) | **528.95** (18) |
| ER[a] (100,200) | 1979.45 (20) | 1890.10 (26) | 1893.23 (23) | 1978.78 (21) | **1995.05** (24) |
| ER[a] (400,500) | 16317.69 (208) | 15692.12 (233) | 15818.42 (212) | 16188.85 (210) | **16387.46** (225) |
| MUTAG[b] | 27.84 (19) | 27.11 (12) | 27.16 (13) | **27.95** (14) | **27.95** (9) |
| ENZYMES[b] | 80.73 (17) | 74.03 (12) | 73.85 (16) | 81.35 (9) | **81.37** (14) |
| PROTEINS[b] | 100.94 (14) | 92.01 (19) | 92.62 (17) | 101.68 (10) | **102.15** (12) |
| IMDB-BIN[b] | 81.89 (18) | 70.56 (21) | 81.50 (10) | 97.11 (9) | **97.47** (11) |
| COLLAB[b] | 2611.83 (22) | 2109.81 (21) | 2430.20 (23) | 2318.19 (18) | **2622.41** (22) |

Table 5: Performance of various model architectures for selected datasets on Maximum Cut. Here, higher is better. GAT is the Graph Attention network (Veličković et al., 2018), GIN is the Graph Isomorphism Network (Xu et al., 2019), GCNN is the Graph Convolutional Neural Network (Morris et al., 2019), and GatedGCNN is the gated version (Li et al., 2015).

Table 6 presents the performance of alternative neural network architectures on Min-Vertex-Cover.

| Dataset | GAT | GCNN | GIN | GatedGCNN | OptGNN |
|---|---|---|---|---|---|
| ER[a] (50,100) | 58.78 (20) | 64.42 (23) | 64.18 (20) | 56.17 (14) | **55.25** (21) |
| ER[a] (100,200) | 129.47 (20) | 141.94 (17) | 140.06 (20) | 130.32 (20) | **126.52** (18) |
| ER[a] (400,500) | 443.93 (43) | 444.12 (33) | 442.11 (31) | 440.90 (28) | **420.70** (41) |
| MUTAG[b] | **7.79** (19) | 8.11 (16) | 7.95 (20) | **7.79** (17) | **7.79** (18) |
| ENZYMES[b] | 21.93 (24) | 25.42 (18) | 25.80 (28) | 20.28 (14) | **20.00** (24) |
| PROTEINS[b] | 28.19 (23) | 31.07 (19) | 32.28 (21) | **25.25** (19) | 25.29 (18) |
| IMDB-BIN[b] | 17.62 (21) | 19.22 (19) | 19.03 (23) | 16.79 (15) | **16.78** (18) |
| COLLAB[b] | 68.23 (23) | 73.32 (17) | 73.82 (26) | 72.92 (13) | **67.50** (23) |

Table 6: Performance of various model architectures compared to OptGNN for selected datasets on Min-Vertex-Cover. Here, lower is better.

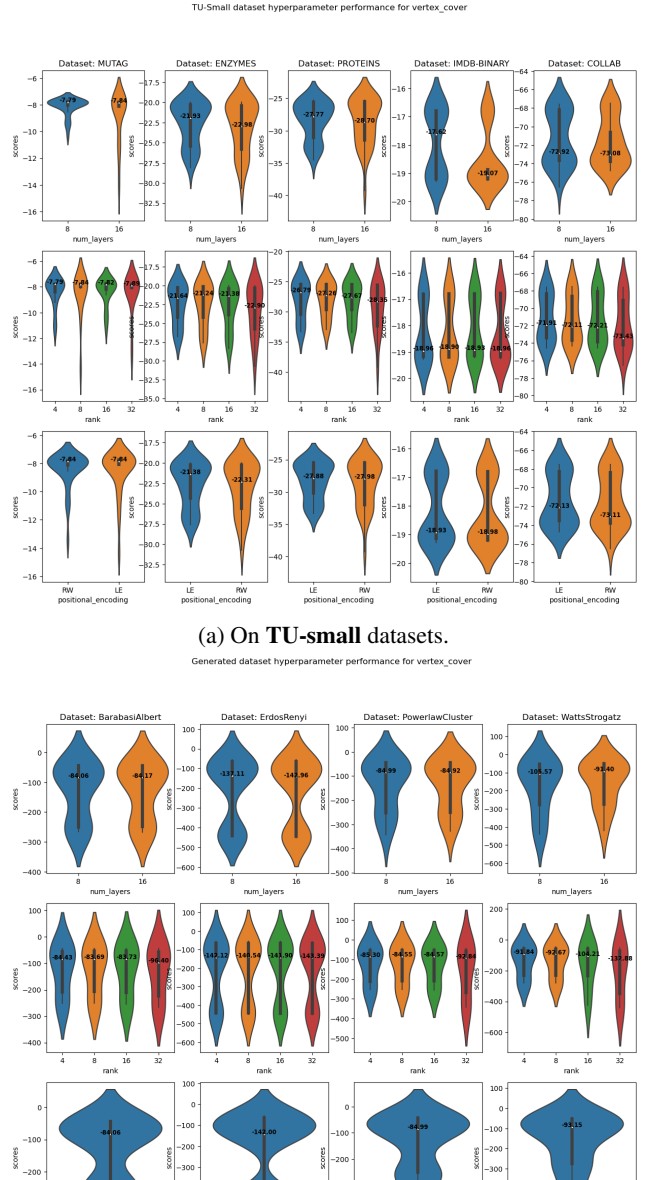

(a) On **TU-small** datasets.

(b) On generated datasets.

Figure 8: Trends in model performance on Min-Vertex-Cover with respect to the number of layers, hidden size, and positional encoding of the models.

## D.5 Effects of hyperparameters on performance

Figure 8 and Figure 9 present overall trends in model performance across hyperparameters.

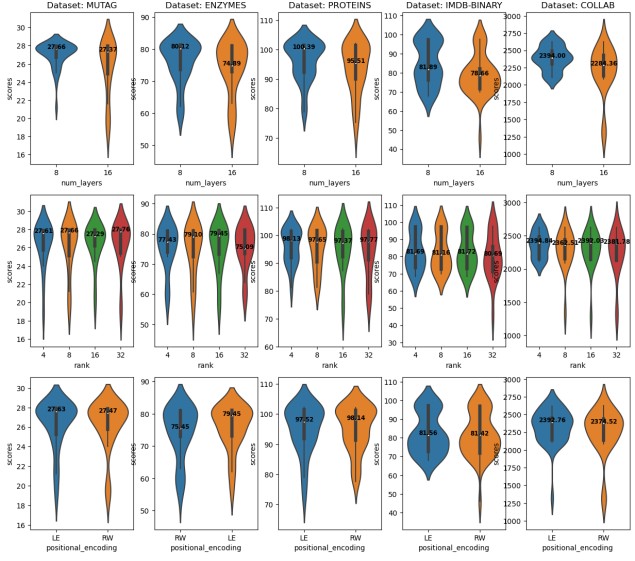

(a) On **TU-small** datasets.

Generated dataset hyperparameter performance for max_cut

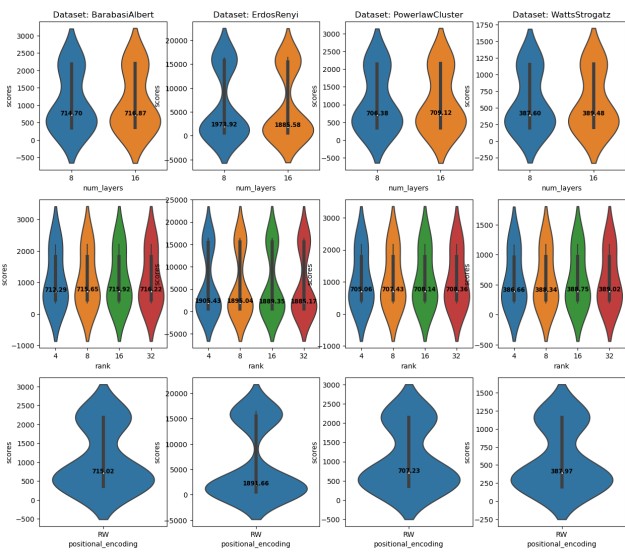

(b) On generated datasets.

Figure 9: Trends in model performance on Max-Cut with respect to the number of layers, hidden size, and positional encoding of the models.

## D.6 Out of distribution testing

OptGNN trained on one dataset performed quite well on other datasets without any finetuning, suggesting that the model can generalize to examples outside its training distribution. For each dataset in our collection, we train a model and then test the trained model on a subset of datasets in the collection. The results are shown in Table 7. It is apparent from the results that the model performance generalizes well to different datasets. Interestingly, we frequently observe that the model reaches its peak performance on a given test dataset even when trained on a different one. This suggests that the model indeed is capturing elements of a more general process instead of just overfitting the training data.

| Train Dataset | MUTAG | ENZYMES | PROTEINS | IMDB-BIN | COLLAB |
|---|---|---|---|---|---|
| BA (50,100) | **7.74** | 20.12 | 27.66 | 17.57 | 74.15 |
| BA (100,200) | **7.74** | 20.35 | 26.03 | 16.86 | 69.29 |
| BA (400,500) | 8.05 | 21.00 | 26.54 | 17.34 | 70.17 |
| ER (50,100) | **7.74** | 20.37 | 28.17 | 16.86 | 69.07 |
| ER (100,200) | 8.05 | 21.52 | 27.72 | 16.89 | 68.83 |
| ER (400,500) | 7.79 | 21.55 | 28.60 | 16.78 | 68.74 |
| HK (50,100) | **7.74** | 20.42 | 25.60 | 17.05 | 69.17 |
| HK (100,200) | 7.84 | 20.43 | 27.30 | 17.01 | 70.20 |
| HK (400,500) | 7.95 | 20.63 | 26.30 | 17.15 | 69.91 |
| WC (50,100) | 7.89 | **20.13** | 25.46 | 17.38 | 70.14 |
| WC (100,200) | 7.79 | 20.30 | 25.45 | 17.91 | 71.16 |
| WC (400,500) | 8.05 | 20.48 | 25.79 | 17.12 | 70.16 |
| MUTAG | **7.74** | 20.83 | 26.76 | 16.92 | 70.09 |
| ENZYMES | **7.74** | 20.60 | 28.29 | 16.79 | 68.40 |
| PROTEINS | 7.89 | 20.22 | **25.29** | 16.77 | 70.26 |
| IMDB-BIN | 7.95 | 20.97 | 27.06 | **16.76** | 68.03 |
| COLLAB | 7.89 | 20.35 | 26.13 | **16.76** | **67.52** |

Table 7: Models for Min-Vertex-Cover trained on "dataset" were tested on a selection of the TU datasets (ENZYMES, PROTEINS, MUTAG, IMDB-BINARY, and COLLAB). We observe that the performance of the models generalizes well even when they are taken out of their training context.

### D.7 Pseudocode for OptGNN training and inference

In algorithm 4, we present pseudocode for OptGNN in the Max-Cut case and in algorithm 5 pseudocode for the forward pass of a general SDP.

---

**Algorithm 4** OptGNN pseudocode for Max-Cut forward pass

---

**Require:** graph $G$
1: $\mathbf{v}^0 = \{v_1, v_2, \ldots, v_N\}$ (random initial feature vectors and/or positional encodings)
2: **for** $t = 1, 2, 3, \ldots, T$ **do**
3:     **for** $v_i^t \in \mathbf{v}^t$ **do**
4:         $v_i^{t+1} \leftarrow v_i^t + \sum_{j \in N(i)} v_j^t$
5:         $\hat{v}_i^{t+1} \leftarrow \text{Linear}(\frac{v_i^{t+1}}{|v_i^{t+1}|})$
6:     **end for**
7: **end for**

---

**Algorithm 5** OptGNN pseudocode for implementing a general SDP forward pass

---

**Require:** graph $G$
1: $\mathbf{v}^0 = \{v_1, v_2, \ldots, v_N\}$ (random initial feature vectors and/or positional encodings)
2: **for** $t = 1, 2, 3, \ldots, T$ **do**
3:     **for** $v_i^t \in \mathbf{v}^t$ **do**
4:         $v_i^{t+1} \leftarrow v_i^t + \text{Autograd}(\mathcal{L}(\mathbf{v}_i^t; G))$
5:         $\hat{v}_i^{t+1} \leftarrow \text{Linear}(\frac{v_i^{t+1}}{|v_i^{t+1}|})$
6:     **end for**
7: **end for**

---

# E  Generalization Analysis

In this section we produce a generalization bound for OptGNN. First we restate our result that a penalized loss approximates the optimum of SDP 2. Our analysis follows from a standard perturbation argument where the key is to bound the lipschitzness of the OptGNN aggregation function. Here we will have to be more precise with the exact polynomial dependence of the lipschitzness of the gradient $\nabla \mathcal{L}_\rho$ and the smoothness of the hessian $\nabla^2 \mathcal{L}_\rho$.

**Notation:**  For the convenience of the proofs in this section, with a slight abuse of notation, we will define loss functions $\mathcal{L}_\rho(V)$ that take matrix arguments $V$ instead of collections of vectors $\mathbf{v}$ where $V$ is simply the vectors in $\mathbf{v}$ concatenated row by row. We will also refer to rows of $V$ by indexing them with set notation $v_i \in V$ where $v_i$ denotes the $i$'th row of $V$. Furthermore, let every vector be bounded in norm by some absolute constant $B$.

We begin by recomputing precisely the number of iterations required for algorithm 3 to approximate the global optimum of SDP 2. Note that this was done in the proof of Theorem C.1 but we do it here with explicit polynomial dependence.

**Lemma E.1** (gradient descent lemma restated). Algorithm 3 computes in $O(\Phi^4 \epsilon^{-4} \log^4(\delta^{-1}))$ iterations a set of vectors $\mathbf{v} := \{\hat{v}_s\}$ for all $s \subseteq \mathcal{S}(P)$ for all $P \in \mathcal{P}$ that satisfy the constraints of SDP 2 to average error $\epsilon$ and approximates the optimum of SDP 2 to error $\epsilon$ with probability $1 - \delta$

$$\Big| \sum_{P_z \in \mathcal{P}} \tilde{\mathbb{E}}_{\hat{\mu}}[P_z(X_z)] - \mathrm{SDP}(\Lambda) \Big| \leq \epsilon$$

where $\mathrm{SDP}(\Lambda)$ is the optimum of SDP 2.

*Proof.* To apply the gradient descent lemma of Jin et al. (2017) Theorem F.1 we need a bound on the lipschitzness of the gradient and the smoothness of the hessian of the loss equation 72. By the lipschitz gradient Lemma E.2 we have that the loss is $\ell := \mathrm{poly}(B)\rho$ lipschitz, and by the smooth hessian Lemma F.1 we have the loss is $\gamma := \mathrm{poly(B)}\rho$ smooth. Then we have by Theorem F.1 that perturbed gradient descent in

$$O\left( \frac{(f(X_0) - f^*)\ell}{\epsilon'^2} \log^4\left( \frac{d\ell\Delta_f}{\epsilon'^2 \delta} \right) \right)$$

iterations can achieve a $(\gamma^2, \epsilon')-$SOSP. In our setting $|f(X_0) - f^*| \leq 1$ because the loss is normalized between $[0, 1]$. Our desired accuracy $\epsilon'$ is $\mathrm{poly}(B^{-1}, \rho^{-1})\Phi^{-2} = \mathrm{poly}(B^{-1}, \epsilon^{-1})\Phi^{-2}$ where we take $\rho = \mathrm{poly}(2^k, \epsilon^{-1})$ as in Lemma E.1. For these settings we achieve an $\epsilon$ approximation in $\tilde{O}(\Phi^4 \epsilon^{-4} \log^4(\delta^{-1}))$ iterations. $\qquad \square$

Next we move on to prove the lipschitzness of the Max-CSP gradient. This is important for two reasons. First we need it to bound the number of iterations required in the proof of Lemma E.1. Secondly, the lipschitzness of the hessian will be the key quantity

**Lemma E.2** (Lipschitz Gradient Lemma Max-CSP). For a Max-CSP instance $\Lambda$, Let $\mathcal{L}_\rho(\mathbf{v})$ be the normalized loss defined in equation 72. Then the gradient satisfies

$$\|\nabla \mathcal{L}_\rho(V) - \nabla \mathcal{L}_\rho(\hat{V})\|_F \leq O(B^4 \rho)\|V - \hat{V}\|_F$$

*Proof.* We begin with the form of the Max-CSP gradient.

$$\left\| \nabla \mathcal{L}_\rho(V) - \nabla \mathcal{L}_\rho(\hat{V}) \right\|_F = \sqrt{ \sum_{w \in \mathcal{F}} \left\| \frac{\partial \mathcal{L}_\rho(V)}{\partial v_w} - \frac{\partial \mathcal{L}_\rho(\hat{V})}{\partial \hat{v}_w} \right\|_F^2 } \qquad (99)$$

Where recall

$$
\frac{\partial \mathcal{L}_\rho(\mathbf{v})}{\partial v_w} = \frac{1}{|\mathcal{P}|} \left[ \sum_{\substack{P_z \in \mathcal{P} \\ \text{s.t } w \subseteq z}} \sum_{\substack{s \subseteq z \\ \text{s.t } w \subseteq s}} y_s \frac{1}{|\mathcal{C}(s)|} \sum_{\substack{w' \subseteq s \\ \text{s.t } \zeta(w,w')=s}} v_{w'} \right.
$$

$$
+ 2\rho \left[ \sum_{\substack{P_z \in \mathcal{P} \\ \text{s.t } w \subseteq z}} \sum_{\substack{w',h,h' \subseteq s \\ \text{s.t } \zeta(w,w')=\zeta(h,h')}} \left( \langle v_w, v_{w'} \rangle - \langle v_h, v_{h'} \rangle \right) v'_w \right.
$$

$$
\left. \left. + (\|v_w\|^2 - 1)v_w \right] \right] \quad (100)
$$

We break the gradient $\nabla \mathcal{L}_\rho(V)$ up into three terms $\mathcal{T}_1(V), \mathcal{T}_2(V),$ and $\mathcal{T}_3(V)$ such that $\partial \mathcal{L}_\rho(V)/\partial v_w = \mathcal{T}_1(V)|_w + \mathcal{T}_2(V)|_w + \mathcal{T}_3(V)|_w$ Where $\mathcal{T}_1(V)|_w, \mathcal{T}_2(V)|_w, \mathcal{T}_3(V)|_w$ are defined as follows

$$
\mathcal{T}_1(V)|_w := \frac{1}{|\mathcal{P}|} \left[ \sum_{\substack{P_z \in \mathcal{P} \\ \text{s.t } w \subseteq z}} \sum_{\substack{s \subseteq z \\ \text{s.t } w \subseteq s}} y_s \frac{1}{|\mathcal{C}(s)|} \sum_{\substack{w' \subseteq s \\ \text{s.t } \zeta(w,w')=s}} v_{w'} \right] \quad (101)
$$

$$
\mathcal{T}_2(V)|_w = \frac{2\rho}{|\mathcal{P}|} \left[ \sum_{\substack{P_z \in \mathcal{P} \\ \text{s.t } w \subseteq z}} \sum_{\substack{w',h,h' \subseteq s \\ \text{s.t } \zeta(w,w')=\zeta(h,h')}} \left( \langle v_w, v_{w'} \rangle - \langle v_h, v_{h'} \rangle \right) v'_w \right] \quad (102)
$$

$$
\mathcal{T}_3(V)|_w := \frac{2\rho}{|\mathcal{P}|} (\|v_w\|^2 - 1)v_w \quad (103)
$$

Such that by triangle inequality we have

$$
\left\| \nabla \mathcal{L}_\rho(V) - \nabla \mathcal{L}_\rho(\hat{V}) \right\|_F \leq \left\| \mathcal{T}_1(V) - \mathcal{T}_1(\hat{V}) \right\|_F + \left\| \mathcal{T}_2(V) - \mathcal{T}_2(\hat{V}) \right\|_F + \left\| \mathcal{T}_3(V) - \mathcal{T}_3(\hat{V}) \right\|_F \quad (104)
$$

Where the three terms in equation 104 are as follows.

$$
\left\| \mathcal{T}_1(V) - \mathcal{T}_1(\hat{V}) \right\|_F = \frac{1}{|\mathcal{P}|} \sqrt{ \sum_{w \in \mathcal{F}} \left\| \left[ \sum_{\substack{P_z \in \mathcal{P} \\ \text{s.t } w \subseteq z}} \sum_{\substack{s \subseteq z \\ \text{s.t } w \subseteq s}} y_s \frac{1}{|\mathcal{C}(s)|} \sum_{\substack{w' \subseteq s \\ \text{s.t } \zeta(w,w')=s}} (v_{w'} - \hat{v}_{w'}) \right] \right\|^2 } \quad (105)
$$

$$
\left\| \mathcal{T}_2(V) - \mathcal{T}_2(\hat{V}) \right\|_F
$$

$$
(106)
$$

$$
:= \frac{2\rho}{|\mathcal{P}|} \sqrt{ \sum_{w \in \mathcal{F}} \left\| \left[ \sum_{\substack{P_z \in \mathcal{P} \\ \text{s.t } w \subseteq z}} \sum_{\substack{w',h,h' \subseteq s \\ \text{s.t } \zeta(w,w')=\zeta(h,h')}} \left[ \left( \langle v_w, v_{w'} \rangle - \langle v_h, v_{h'} \rangle \right) v'_w - \left( \langle \hat{v}_w, \hat{v}_{w'} \rangle - \langle \hat{v}_h, \hat{v}_{h'} \rangle \right) \hat{v}'_w \right] \right] \right\|^2 } \quad (107)
$$

$$
\left\| \mathcal{T}_3(V) - \mathcal{T}_3(\hat{V}) \right\|_F := \frac{2\rho}{|\mathcal{P}|} \sqrt{ \sum_{w \in \mathcal{F}} \left\| \left[ (\|v_w\|^2 - 1)v_w - (\|\hat{v}_w\|^2 - 1)\hat{v}_w \right] \right\|^2 } \quad (108)
$$

We bound the terms one by one. First we bound term 1.

**Term 1:**

$$\left\|\mathcal{T}_1(V) - \mathcal{T}_1(\hat{V})\right\|_F = \frac{1}{|\mathcal{P}|}\sqrt{\sum_{w\in\mathcal{F}}\left\|\left[\sum_{\substack{P_z\in\mathcal{P}\\ \text{s.t } w\subseteq z}}\sum_{\substack{s\subseteq z\\ \text{s.t } w\subseteq s}} y_s \frac{1}{|\mathcal{C}(s)|}\sum_{\substack{w'\subseteq s\\ \text{s.t } \zeta(w,w')=s}}(v_{w'}-\hat{v}_{w'})\right]\right\|^2} \tag{109}$$

We will need to define some matrices so that we can write the above expression as the frobenius inner product of matrices. First we define $G_\Lambda$ to be the adjacency matrix of the constraint graph. In particular we denote the $(w,w')$ entry of $G_\Lambda$ as $G_\Lambda|_{w,w'}$ defined as follows.

$$G_\Lambda|_{w,w'} := \begin{cases} 1, & \text{if there exists } P_z \in \mathcal{P} \text{ s.t } \zeta(w,w') \subseteq z \\ 0, & \text{otherwise} \end{cases}$$

Furthermore, we define $M_{y_s/C(s)}$ to be a matrix comprised of a set of coefficients $y_s/|\mathcal{C}(s)|$ corresponding to every edge in the constraint graph $G_\Lambda$. The $(w,w')$ entry of $M_{y_s/C(s)}$ is denoted $M_{y_s/C(s)}|_{w,w'}$ and defined as follows.

$$M_{y_s/C(s)}|_{w,w'} := \begin{cases} y_s/|\mathcal{C}(s)|, & \text{if there exists } P_z \in \mathcal{P} \text{ s.t } \zeta(w,w') = s \subseteq z \\ 0, & \text{otherwise} \end{cases}$$

Then rewriting equation 109

$$\left\|\mathcal{T}_1(V) - \mathcal{T}_1(\hat{V})\right\|_F = \frac{1}{|\mathcal{P}|}\sqrt{\sum_{w\in\mathcal{F}}\left\|\left[e_w^T G_\Lambda \odot M_{y_s/C(s)}(V-\hat{V})\right]\right\|^2} \tag{110}$$

$$= \frac{1}{|\mathcal{P}|}\sqrt{\left\|(G_\Lambda \odot M_{y_s/\mathcal{C}(s)})(V-\hat{V})\right\|_F^2} \tag{111}$$

By Cauchy-Schwarz we obtain

$$\left\|\mathcal{T}_1(V) - \mathcal{T}_1(\hat{V})\right\|_F \leq \frac{1}{|\mathcal{P}|}\left\|G_\Lambda \odot M_{y_s/\mathcal{C}(s)}\right\|\left\|V-\hat{V}\right\|_F \tag{112}$$

Next we move on to bound term 2.

**Term 2:**

$$\left\|\mathcal{T}_2(V) - \mathcal{T}_2(\hat{V})\right\|_F = \tag{113}$$

$$\frac{2\rho}{|\mathcal{P}|}\sqrt{\sum_{w\in\mathcal{F}}\left\|\left[\sum_{\substack{P_z\in\mathcal{P}\\ \text{s.t } w\subseteq z}}\sum_{\substack{w',h,h'\subseteq s\\ \text{s.t } \zeta(w,w')=\zeta(h,h')}}\left[(\langle v_w,v_{w'}\rangle v'_w - \langle v_h,v_{h'}\rangle v'_w) - (\langle \hat{v}_w,\hat{v}_{w'}\rangle \hat{v}'_w - \langle \hat{v}_h,\hat{v}_{h'}\rangle \hat{v}'_w)\right]\right\|^2}$$
$$\tag{114}$$

Let the vector $\delta_{w'} = v_w - v_{w'}$ and let the scalar $\delta_{w,w'} = \langle v_w,v_{w'}\rangle - \langle \hat{v}_w,\hat{v}_{w'}\rangle$. Then we have by plugging definitions that

$$\left(\langle v_w,v_{w'}\rangle v'_w - \langle \hat{v}_w,\hat{v}_{w'}\rangle \hat{v}'_w\right) = \left(\langle v_w,v_{w'}\rangle v'_w - ((\langle v_w,v_{w'}\rangle + \delta_{w,w'})(v'_w + \delta_{w'}))\right) \tag{115}$$

Substituting equation 115 into equation 114 in the square root we obtain

$$= \frac{2\rho}{|\mathcal{P}|}\sqrt{\sum_{w\in\mathcal{F}}\left\|\sum_{\substack{P_z\in\mathcal{P}\\ \text{s.t } w\subseteq z}}\sum_{w'\subseteq s} -\langle v_w,v_{w'}\rangle\delta_{w'} - \delta_{w,w'}v'_w + \delta_{w,w'}\delta_{w'}\right\|^2} \tag{116}$$

Applying triangle inequality we obtain

$$\leq \frac{16\rho}{|\mathcal{P}|} \sqrt{ \sum_{w \in \mathcal{F}} \left\| \sum_{\substack{P_z \in \mathcal{P} \\ \text{s.t } w \subseteq z}} \sum_{w' \subseteq s} -\langle v_w, v_{w'} \rangle \delta_{w'} \right\|^2 + \sum_{w \in \mathcal{F}} \left\| \sum_{\substack{P_z \in \mathcal{P} \\ \text{s.t } w \subseteq z}} \sum_{w' \subseteq s} -\delta_{w,w'} v'_w \right\|^2 + \sum_{w \in \mathcal{F}} \left\| \sum_{\substack{P_z \in \mathcal{P} \\ \text{s.t } w \subseteq z}} \sum_{w' \subseteq s} \delta_{w,w'} \delta_{w'} \right\|^2 }$$

(117)

Let $M_{\langle v_w, v_{w'} \rangle} \in \mathbb{R}^{|\mathcal{P}|2^k \times |\mathcal{P}|2^k}$ be the matrix whose $w, w'$ entry is $\langle v_w, v'_w \rangle$. Let $M_{\delta_{w,w'}}$ be the matrix whose $w, w'$ entry is $\delta_{w,w'}$. Let $\odot$ denote the entrywise product of two matrices.

$$= \frac{16\rho}{|\mathcal{P}|} \sqrt{ \left\| G_\Lambda \odot M_{\langle v_w, v_{w'} \rangle}(V - \hat{V}) \right\|_F^2 + \left\| G_\Lambda \odot M_{\delta_{w,w'}}(V - \hat{V}) \right\|_F^2 + \left\| G_\Lambda \odot M_{\delta_{w,w'}}(V - \hat{V}) \right\|_F^2 }$$

(118)

Applying Cauchy-Schwarz we obtain

$$= \frac{16\rho}{|\mathcal{P}|} \sqrt{ \left\| G_\Lambda \odot M_{\langle v_w, v_{w'} \rangle} \right\|_F^2 \left\| (V - \hat{V}) \right\|_F^2 + \left\| G_\Lambda \odot M_{\delta_{w,w'}} \right\|_F^2 \left\| (V - \hat{V}) \right\|_F^2 + \left\| G_\Lambda \odot M_{\delta_{w,w'}} \right\|_F^2 \left\| (V - \hat{V}) \right\|_F^2 }$$

(119)

We apply entrywise upper bound $\langle v_w, v_{w'} \rangle \leq B^2$ which can be done because we're taking a frobenius norm so the sign of each entry does not matter. Likewise we apply a crude entrywise upper bound of $\delta_{w,w'} \leq B^2$ again because the sign of each entry does not matter in the frobenius norm (for each row this is euclidean norm).

$$= \frac{16B^4 \rho}{|\mathcal{P}|} \sqrt{ \|G_\Lambda\|_F^2 \left\| V - \hat{V} \right\|_F^2 + \|G_\Lambda\|_F^2 \left\| V - \hat{V} \right\|_F^2 + \|G_\Lambda\|_F^2 \left\| V - \hat{V} \right\|_F^2 }$$

(120)

Using the fact that $\|G_\lambda\|_F \leq 2^k \sqrt{|\mathcal{P}|}$

$$= \frac{2^k B^4 \rho}{\sqrt{|\mathcal{P}|}} \left\| V - \hat{V} \right\|_F \leq 2^k B^4 \rho \left\| V - \hat{V} \right\|_F$$

(121)

Therefore we have established

$$\left\| \mathcal{T}_2(V) - \mathcal{T}_2(\hat{V}) \right\|_F \leq 2^k B^4 \rho \left\| V - \hat{V} \right\|_F$$

(122)

Finally we move on to bound term 3.

**Term 3:**

$$\left\| \mathcal{T}_3(V) - \mathcal{T}_3(\hat{V}) \right\|_F := \frac{2\rho}{|\mathcal{P}|} \sqrt{ \sum_{w \in \mathcal{F}} \left\| \left[ (\|v_w\|^2 - 1)v_w - (\|\hat{v}_w\|^2 - 1)\hat{v}_w \right] \right\|^2 }$$

(123)

$$= \frac{2\rho}{|\mathcal{P}|} \sqrt{ \sum_{w \in \mathcal{F}} \left\| \left[ \|v_w\|^2 v_w - \|\hat{v}_w\|^2 \hat{v}_w + \hat{v}_w - v_w \right] \right\|^2 }$$

(124)

Applying triangle inequality we obtain

$$\leq \frac{2\rho}{|\mathcal{P}|} \sqrt{ \sum_{w \in \mathcal{F}} \left[ \left\| \|v_w\|^2 v_w - \|\hat{v}_w\|^2 \hat{v}_w \right\|^2 \right] } + \frac{2\rho}{|\mathcal{P}|} \sqrt{ \sum_{w \in \mathcal{F}} \left[ \|\hat{v}_w - v_w\|^2 \right] }$$

(125)

Using the fact that $\sum_{w \in \mathcal{F}} \left[ \|\hat{v}_w - v_w\|^2 \right] = \|V - \hat{V}\|_F$

$$= \frac{2\rho}{|\mathcal{P}|} \sqrt{ \sum_{w \in \mathcal{F}} \left[ \left\| \|v_w\|^2 v_w - \|\hat{v}_w\|^2 \hat{v}_w \right\|^2 \right] } + \frac{2\rho}{|\mathcal{P}|} \|V - \hat{V}\|_F$$

(126)

Let $\delta_{v_w} := v'_w - v_w$ then substituting the definition we obtain

$$= \frac{2\rho}{|\mathcal{P}|} \sqrt{\sum_{w \in \mathcal{F}} \left[ \left\| \|v_w\|^2 v_w - \|v_w + \delta_{v_w}\|^2 (v_w + \delta_{v_w}) \right\|^2 \right]} + \frac{2\rho}{|\mathcal{P}|} \|V - \hat{V}\|_F \tag{127}$$

expanding the expression in the square root we obtain

$$= \frac{2\rho}{|\mathcal{P}|} \sqrt{\sum_{w \in \mathcal{F}} \left[ \left\| -2\langle v_w, \delta_{v_w} \rangle v_w - \|\delta_{v_w}\|^2 v_w - \|v_w\|^2 \delta_{v_w} - 2\langle v_w, \delta_{v_w} \rangle \delta_{v_w} - \|\delta_{v_w}\|^2 \delta_{v_w} \right\|^2 \right]} \tag{128}$$

$$+ \frac{2\rho}{|\mathcal{P}|} \|V - \hat{V}\|_F \tag{129}$$

By triangle inequality we upper bound by

$$\leq \frac{2\rho}{|\mathcal{P}|} \sqrt{\sum_{w \in \mathcal{F}} \left[ \left\| -2\langle v_w, \delta_{v_w} \rangle v_w \right\|^2 \right]}$$

$$+ \frac{2\rho}{|\mathcal{P}|} \sqrt{\sum_{w \in \mathcal{F}} \left[ \left\| -\|\delta_{v_w}\|^2 v_w \right\|^2 \right]}$$

$$+ \frac{2\rho}{|\mathcal{P}|} \sqrt{\sum_{w \in \mathcal{F}} \left[ \left\| -\|v_w\|^2 \delta_{v_w} \right\|^2 \right]}$$

$$+ \frac{2\rho}{|\mathcal{P}|} \sqrt{\sum_{w \in \mathcal{F}} \left[ \left\| -2\langle v_w, \delta_{v_w} \rangle \delta_{v_w} \right\|^2 \right]}$$

$$+ \frac{2\rho}{|\mathcal{P}|} \sqrt{\sum_{w \in \mathcal{F}} \left[ \left\| -\|\delta_{v_w}\|^2 \delta_{v_w} \right\|^2 \right]}$$

$$+ \frac{2\rho}{|\mathcal{P}|} \|V - \hat{V}\|_F \tag{130}$$

For the first term consider by Cauchy-Schwarz

$$\| -2\langle v_w, \delta_{v_w} \rangle v_w \|^2 = 4\langle v_w, \delta_{v_w} \rangle^2 \|v_w\|^2 \leq 4B^4 \|\delta_{v_w}\|^2$$

Consider the second term which we upper bound via the norm bound on $\|v_w\| \leq B$

$$\left\| \|\delta_{v_w}\|^2 v_w \right\| = \|\delta_{v_w}\|^2 B^2$$

Consider the third term which we upper bound via the norm bound on $\|v_w\| \leq B$

$$\left\| -\|v_w\|^2 \delta_{v_w} \right\|^2 = \|v_w\|^2 \|\delta_{v_w}\|^2 \leq B^2 \|\delta_{v_w}\|^2$$

Consider the fourth term which we upper bound via Cauchy-Schwarz

$$\| -2\langle v_w, \delta_{v_w} \rangle \delta_{v_w} \|^2 = 4\langle v_w, \delta_{v_w} \rangle^2 \|\delta_{v_w}\|^2 \leq 4B^4 \|\delta_{v_w}\|^2$$

Consider the fifth term. We apply a crude upper bound of $\|\delta_{v_w}\|^2 \leq B^2$

$$\left\| -\|\delta_{v_w}\|^2 \delta_{v_w} \right\|^2 = \|\delta_{v_w}\|^2 \|\delta_{v_w}\|^2 \leq B^2 \|\delta_{v_w}\|^2$$

Therefore we conclude

$$130 \leq \frac{20B^2\rho}{|\mathcal{P}|} \sqrt{\sum_{w \in \mathcal{F}} \|\delta_{v_w}\|^2} + \frac{2\rho}{|\mathcal{P}|} \|V - \hat{V}\|_F \tag{131}$$

$$= \frac{20B^2\rho}{|\mathcal{P}|} \left\| V - \hat{V} \right\|_F + \frac{2\rho}{|\mathcal{P}|} \|V - \hat{V}\|_F \tag{132}$$

So we conclude our bound on term 3

$$\left\|\mathcal{T}_3(V) - \mathcal{T}_3(\hat{V})\right\|_F \leq 20B^2\rho \left\|V - \hat{V}\right\|_F \tag{133}$$

as desired. Putting our bounds for terms $1, 2$, and $3$ into equation 104 we obtain

$$\left\|\nabla\mathcal{L}_\rho(V) - \nabla\mathcal{L}_\rho(\hat{V})\right\|_F \leq O(B^2\rho) \left\|V - \hat{V}\right\|_F \tag{134}$$

as desired. $\qquad\square$

**Lemma E.3** (Max-CSP gradient perturbation analysis)**.** For any set of matrices $M := \{M_1, M_2, ..., M_T\} \in \mathbb{R}^{2r \times r}$ a perturbation $U := \{U_1, U_2, ..., U_T\} \in \mathbb{R}^{2r \times r}$ that satisfies $\|U_i\|_{1,1} \leq \epsilon$ for all $i \in [T]$. Let $M + U$ denote the elementwise addition of $M$ and $U$ as such $M+U := \{M_1+U_1, M_2+U_2, ..., M_T+U_T\}$. Then for a matrix $V \in \mathbb{R}^{r \times N}$ satisfying $\|V\|_F \leq \sqrt{d}$ we define the aggregation function AGG : $\mathbb{R}^{r \times N} \to \mathbb{R}^{2r \times N}$ as such

$$\text{AGG}(V) := \begin{bmatrix} V \\ \nabla\mathcal{L}_\rho(V) \end{bmatrix} \tag{135}$$

Furthermore, let $\text{LAYER}_{M_i}(V)$ denote

$$\text{LAYER}_{M_i}(V) = M_i(\text{AGG}(V))$$

Finally, let $\text{OptGNN}_M : \mathbb{R}^{r \times N} \to \mathbb{R}$ be defined as

$$\text{OptGNN}_M(V) = \mathcal{L}_\rho \circ \text{LAYER}_{M_T} \circ .... \circ \text{LAYER}_{M_2} \circ \text{LAYER}_{M_1}(V)$$

Here we feed the output of the final layer $\text{LAYER}_T$ to the loss function $\mathcal{L} : \mathbb{R}^{r \times N} \to \mathbb{R}$. Then

$$\left\|\text{OptGNN}_M(V) - \text{OptGNN}_{M+U}(V)\right\|_F \leq O(\epsilon B^{2T}\rho^{2T} r^{2T})$$

*Proof.* We begin by analyzing how much the gradient perturbs the input to a single layer. First we apply the definition of LAYER and AGG to obtain

$$\|\text{LAYER}_M(V) - \text{LAYER}_{M+U}(V)\|_F \tag{136}$$
$$\leq \|(M+U)V - MV\|_F + \|\nabla\mathcal{L}_\rho((M+U)V) - \nabla\mathcal{L}_\rho(MV)\|_F \tag{137}$$
$$= \|UV\|_F + \|\nabla\mathcal{L}_\rho((M+U)V) - \nabla\mathcal{L}_\rho(MV)\|_F \tag{138}$$
$$\leq \|U\|_F \|V\|_F + \|\nabla\mathcal{L}_\rho((M+U)V) - \nabla\mathcal{L}_\rho(MV)\|_F \tag{139}$$
$$\leq \epsilon r^{3/2} + \|\nabla\mathcal{L}_\rho((M+U)V) - \nabla\mathcal{L}_\rho(MV)\|_F \tag{140}$$

Here the first inequality follows by triangle inequality. The second inequality follows by Cauchy-Schwarz. Finally the frobenius norm of $U$ is $\epsilon r$ the frobenius norm of $V$ is Then applying the lipschitzness of the gradient Lemma E.2 we obtain.

$$\leq \epsilon r^{3/2} + O(B^2\rho) \|(M+U)V - MV\|_F = O(B^2\rho) \|UV\|_F \tag{141}$$
$$\leq \epsilon r^{3/2} + O(B^2\rho) \|U\|_F \|V\|_F \tag{142}$$
$$\leq \epsilon r^{3/2} + O(B^2\rho)\epsilon d \|V\|_F \tag{143}$$
$$= O(B^2\rho)\epsilon r^{3/2} \tag{144}$$

To conclude, we've established that

$$\|\text{LAYER}_M(V) - \text{LAYER}_{M+U}(V)\|_F = O(B^2\rho)\epsilon r^{3/2} \tag{145}$$

Next we upper bound the lipschitzness of the LAYER function by the frobenius norm of $\|M\|_F \leq O(r)$ multiplied by the lipschitzness of $\nabla\mathcal{L}_\rho$ which is $O(B^2\rho)$. Taken together we find the the lipschitzness of LAYER is upper bounded by $O(B^2\rho r)$.

Note that OptGNN is comprised of $T$ layers of LAYER functions followed by the evaluation of the loss $\mathcal{L}$. The $T$ layers contribute a geometric series of errors with the dominant term being $O(\epsilon r^{3/2}B^2\rho * (B^2\rho r)^T) = O(\epsilon(B^2\rho r)^{2T})$. The smaller order terms contribute no more than an additional multiplicative factor of $T$ leading to an error of $O(T\epsilon(B^2\rho r)^{2T})$. At any rate, the dominant

term is the exponential dependence on the number of layers. Finally, OptGNN's final layer is the evaluation of the loss $\mathcal{L}_\rho$. The lipschitzness of the loss $\mathcal{L}_\rho$ can be computed as follows. For any pair of matrices $V$ and $\hat{V}$ both in $\mathbb{R}^{r \times r}$ we have

$$|\mathcal{L}_\rho(V) - \mathcal{L}_\rho(\hat{V})| \leq \frac{\rho 2^k}{|\mathcal{P}|} \left\| V - \hat{V} \right\|_F^2 \tag{146}$$

where we used the fact that the loss is dominated by its quadratic penalty term $\rho$ times the square of the violations where each row in $V$ can be counted in up to $2^k$ constraints normalized by the number of predicates $|\mathcal{P}|$. Putting this together applying the LAYER function over $T$ layers we obtain

$$\left\| \text{OptGNN}_{M+U}(V) - \text{OptGNN}_M(V) \right\| \leq O(\epsilon B^{2T} \rho^{2T} r^{2T}) \tag{147}$$

$\square$

At this point we restate some elementary theorems in the study of PAC learning adapted for the unsupervised learning setting.

**Lemma E.4** (Agnostic PAC learnability (folklore)). Let $x_1, x_2, ..., x_N \sim \mathcal{D}$ be data drawn from a distribution $\mathcal{D}$. Let $\mathcal{H}$ be any finite hypothesis class comprised of functions $h \in \mathcal{H}$ where the range of $h$ in bounded in $[0, 1]$. Let $\hat{h} \in \mathcal{H}$ be the empirical loss minimizer

$$\arg\min_{h \in \mathcal{H}} \frac{1}{N} \sum_{i \in [N]} h(x_i)$$

Let $\delta$ be the failure probability and let $\epsilon$ be the approximation error. Then we say $\mathcal{H}$ is $(N, \epsilon, \delta)$-PAC learnable if

$$\Pr[|\frac{1}{N} \sum_{i \in [N]} \hat{h}(x_i) - \mathbb{E}_{x \sim \mathcal{D}}[h(x)]| \geq \epsilon] \leq \delta$$

Furthermore, $\mathcal{H}$ is $(N, \epsilon, \delta)$-PAC learnable so long as

$$N = O\left( \log |\mathcal{H}| \frac{\log(1/\delta)}{\epsilon^2} \right)$$

The proof is a standard epsilon net union bound argument which the familiar reader should feel free to skip.

*Proof.* For any fixed hypothesis $h$ the difference between the distributional loss between $[0, 1]$ and the empirical loss can be bounded by Hoeffding.

$$\Pr_{x_1, ..., x_N \sim \mathcal{D}} \left[ \mathbb{E}_{x \sim \mathcal{D}}[h(x)] - \frac{1}{N} \sum_{i \in [N]} h(x_i) > \epsilon \right] \leq \exp(-N\epsilon^2)$$

Let $\hat{h} \in \mathcal{H}$ be the empirical risk minimizer within the hypothesis class $\mathcal{H}$ on data $\{x_1, x_2, .., x_N\}$. What is the probability that the empirical risk deviates from the distributional risk by greater than $\epsilon$? i.e we wish to upper bound the quantity

$$\Pr_{x_1, ..., x_N \sim \mathcal{D}} \left[ \mathbb{E}_{x \sim \mathcal{D}}[\hat{h}(x)] - \frac{1}{N} \sum_{i \in [N]} \hat{h}(x_i) > \epsilon \right]$$

Of course the biggest caveat is that $\hat{h}$ depends on $x_1, ..., x_N$ and thus Hoeffding does not apply. Therefore we upper bound by the probability over draws $x_1, ..., x_N \sim \mathcal{D}$ that there exists ANY hypothesis in $\mathcal{H}$ that deviates from its distributional risk by more than $\epsilon$.

$$\Pr_{x_1, ..., x_N \sim \mathcal{D}} \left[ \mathbb{E}_{x \sim \mathcal{D}}[\hat{h}(x)] - \frac{1}{N} \sum_{i \in [N]} \hat{h}(x_i) > \epsilon \right] \leq \Pr_{x_1, ..., x_N \sim \mathcal{D}} \left[ \bigcup_{h \in \mathcal{H}} \left[ \mathbb{E}_{x \sim \mathcal{D}}[h(x)] - \frac{1}{N} \sum_{i \in [N]} h(x_i) > \epsilon \right] \right] \tag{148}$$

This follows because the event that $\hat{h}$ deviates substantially from its distributional loss is one of the elements of the union on the right hand side.

$$\left[ \mathbb{E}_{x \sim \mathcal{D}}[\hat{h}(x)] - \frac{1}{N} \sum_{i \in [N]} \hat{h}(x_i) > \epsilon \right] \subseteq \left[ \bigcup_{h \in \mathcal{H}} \left[ \mathbb{E}_{x \sim \mathcal{D}}[h(x)] - \frac{1}{N} \sum_{i \in [N]} h(x_i) > \epsilon \right] \right]$$

Then we have via union bound that

$$\Pr_{x_1,...,x_N \sim \mathcal{D}} \left[ \mathbb{E}_{x \sim \mathcal{D}}[\hat{h}(x)] - \frac{1}{N} \sum_{i \in [N]} \hat{h}(x_i) > \epsilon \right] \leq \Pr_{x_1,...,x_N \sim \mathcal{D}} \left[ \bigcup_{h \in \mathcal{H}} \left[ \mathbb{E}_{x \sim \mathcal{D}}[h(x)] - \frac{1}{N} \sum_{i \in [N]} h(x_i) > \epsilon \right] \right]$$

$$\leq \sum_{h \in \mathcal{H}} \Pr_{x_1,...,x_N \sim \mathcal{D}} \left[ \mathbb{E}_{x \sim \mathcal{D}}[h(x)] - \frac{1}{N} \sum_{i \in [N]} h(x_i) > \epsilon \right]$$

$$\leq \sum_{h \in \mathcal{H}} \exp(-\epsilon^2 N) = |\mathcal{H}| \exp(-\epsilon^2 N) \quad (149)$$

where the last line follows by Hoeffding. In particular if we want the failure probability to be $\delta$ then

$$|\mathcal{H}| \exp(-\epsilon^2 N) \leq \delta \quad (150)$$

which implies

$$N \geq \frac{1}{\epsilon^2} \ln(\frac{|\mathcal{H}|}{\delta})$$

Suffices for the $\hat{h} \in \mathcal{H}$ to deviate from its empirical risk by less than $\epsilon$. □

Finally putting our perturbation analysis Lemma E.3 together with the agnostic PAC learnability Lemma E.4

**Lemma E.5.** Let $\Lambda_1, \Lambda_2, ..., \Lambda_\Gamma \sim \mathcal{D}$ be Max-CSP instances drawn from a distribution over instances $\mathcal{D}$ with no more than $|\mathcal{P}|$ predicates. Let $M$ be a set of parameters $M = \{M_1, M_2, ..., M_T\}$ in a parameter space $\Theta$. Then for $T = O(\Phi^4)$, for $\Gamma = O(\frac{1}{\epsilon^4} \Phi^6 \log^4(\delta^{-1}))$, let $\hat{M}$ be the empirical loss minimizer

$$\hat{M} := \arg\min_{M \in \Theta} \frac{1}{\Gamma} \sum_{i \in [\Gamma]} \text{OptGNN}_{(M, \Lambda_i)}(V)$$

Then we have that OptGNN is $(\Gamma, \epsilon, \delta)$-PAC learnable

$$\Pr \left[ \left| \frac{1}{\Gamma} \sum_{i \in [\Gamma]} \text{OptGNN}_{(\hat{M}, \Lambda_i)}(V) - \mathbb{E}_{\Lambda \sim \mathcal{D}} \left[ \text{OptGNN}_{(\hat{M}, \Lambda)}(U) \right] \right| \leq \epsilon \right] \geq 1 - \delta$$

*Proof.* The result follows directly from the agnostic PAC learning Lemma E.4 and the perturbation analysis Lemma E.3. We have that for a net of interval size $\frac{\epsilon}{r^{2T}}$ suffices for an $\epsilon$ approximation. The cardinality of the net is then $r^{2T}/\epsilon$ per parameter raised the power of the number of parameters required for OptGNN to represent an $\epsilon$ approximate solution to SDP($\Lambda$). Each LAYER is comprised of a matrix $M_i$ of dimension $r \times 2r$ for $r = \Phi$. By Lemma E.1 we need a total of $T = O(\Phi^4 \epsilon^{-4} \log^4(\delta^{-1}))$ layers to represent an $\epsilon$ optimal solution with probability $1 - \delta$. Then the total number of parameters in the network is $O(\Phi^6 \epsilon^{-4} \log^4(\delta^{-1}))$ as desired. □

# F   Hessian Lemmas

This section is to prove the hessian $\nabla^2 \mathcal{L}_\rho$ is smooth. This is relevant for bounding the number of iterations required for algorithm 3.

**Lemma F.1** (Smooth Hessian Lemma). For a Max-CSP instance $\Lambda$, Let $\mathcal{L}_\rho(\mathbf{v})$ be the normalized loss defined in equation 72. Then the hessian satisfies

$$\left\| \nabla^2 \mathcal{L}_\rho(\mathbf{v}) - \nabla^2 \mathcal{L}_\rho(\hat{\mathbf{v}}) \right\| \leq 8B\rho \|V - \hat{V}\|_F.$$

In particular for $\rho = poly(k, \epsilon^{-1})$ we have that the hessian is $poly(B, k, \epsilon^{-1})$ smooth.

*Proof.* Next we verify this for the Max-CSP hessian. The form of its hessian is far more complex. To simplify matters we first consider the hessian of polynomials of the form $\mathcal{T}(v_i, v_j, v_k, v_\ell) = (\langle v_i, v_j \rangle - \langle v_k, v_\ell \rangle)^2$

$$\nabla^2 \mathcal{T}(v_i, v_j, v_k, v_\ell) := \begin{cases} \frac{\partial}{\partial v_{ia} \partial v_{kb}} \mathcal{T} = -2v_{ja}v_{\ell b}, & \text{for } a, b \in [r] \\ \frac{\partial}{\partial v_{ia} \partial v_{\ell b}} \mathcal{T} = -2v_{ja}v_{kb}, & \text{for } a, b \in [r] \\ \frac{\partial}{\partial v_{ja} \partial v_{kb}} \mathcal{T} = -2v_{ia}v_{\ell b}, & \text{for } a, b \in [r] \\ \frac{\partial}{\partial v_{ja} \partial v_{\ell b}} \mathcal{T} = -2v_{ia}v_{kb}, & \text{for } a, b \in [r] \\ \frac{\partial}{\partial v_{ia} \partial v_{jb}} \mathcal{T} = 2v_{ja}v_{ib}, & \text{for } a, b \in [r] \\ \frac{\partial}{\partial v_{ka} \partial v_{\ell b}} \mathcal{T} = 2v_{\ell a}v_{kb}, & \text{for } a, b \in [r] \\ \frac{\partial}{\partial v_{ia}^2} \mathcal{T} = v_{ja}^2, & \text{for } a \in [r] \\ \frac{\partial}{\partial v_{ja}^2} \mathcal{T} = v_{ia}^2, & \text{for } a \in [r] \\ \frac{\partial}{\partial v_{ka}^2} \mathcal{T} = v_{\ell a}^2, & \text{for } a \in [r] \\ \frac{\partial}{\partial v_{\ell a}^2} \mathcal{T} = v_{ka}^2, & \text{for } a \in [r] \\ 0, & \text{otherwise} \end{cases}$$

We can decompose the Hessian as a sum of 10 matrices corresponding to the cases enumerated above.

$$\begin{aligned} \nabla^2 \mathcal{T}(\mathbf{v}) = {}& \nabla^2 \mathcal{T}(\mathbf{v})|_{(i,k)} + \nabla^2 \mathcal{T}(\mathbf{v})|_{(i,\ell)} + \nabla^2 \mathcal{T}(\mathbf{v})|_{(j,k)} + \nabla^2 \mathcal{T}(\mathbf{v})|_{(j,\ell)} \\ & + \nabla^2 \mathcal{T}(\mathbf{v})|_{(i,j)} + \nabla^2 \mathcal{T}(\mathbf{v})|_{(k,\ell)} \\ & + \nabla^2 \mathcal{T}(\mathbf{v})|_{(i,i)} + \nabla^2 \mathcal{T}(\mathbf{v})|_{(j,j)} + \nabla^2 \mathcal{T}(\mathbf{v})|_{(k,k)} + \nabla^2 \mathcal{T}(\mathbf{v})|_{(\ell,\ell)}. \end{aligned} \tag{151}$$

Then we can compute

$$\begin{aligned} \|\nabla^2 \mathcal{T}(\mathbf{v}) - \nabla^2 \mathcal{T}(\mathbf{v})\|_F \leq {}& \\ & \|\nabla^2 \mathcal{T}(\mathbf{v})|_{(i,k)} - \nabla^2 \mathcal{T}(\hat{\mathbf{v}})|_{(i,k)}\|_F + \|\nabla^2 \mathcal{T}(\mathbf{v})|_{(i,\ell)} - \nabla^2 \mathcal{T}(\hat{\mathbf{v}})|_{(i,\ell)}\|_F \\ & + \|\nabla^2 \mathcal{T}(\mathbf{v})|_{(j,k)} - \nabla^2 \mathcal{T}(\hat{\mathbf{v}})|_{(j,k)}\|_F + \|\nabla^2 \mathcal{T}(\mathbf{v})|_{(i,k)} - \nabla^2 \mathcal{T}(\hat{\mathbf{v}})|_{(j,\ell)}\|_F \\ & + \|\nabla^2 \mathcal{T}(\mathbf{v})|_{(i,j)} - \nabla^2 \mathcal{T}(\hat{\mathbf{v}})|_{(i,j)}\|_F + \|\nabla^2 \mathcal{T}(\mathbf{v})|_{(k,\ell)} - \nabla^2 \mathcal{T}(\hat{\mathbf{v}})|_{(k,\ell)}\|_F \\ & + \|\nabla^2 \mathcal{T}(\mathbf{v})|_{(i,i)} - \nabla^2 \mathcal{T}(\hat{\mathbf{v}})|_{(i,i)}\|_F + \|\nabla^2 \mathcal{T}(\mathbf{v})|_{(j,j)} - \nabla^2 \mathcal{T}(\hat{\mathbf{v}})|_{(j,j)}\|_F \\ & + \|\nabla^2 \mathcal{T}(\mathbf{v})|_{(k,k)} - \nabla^2 \mathcal{T}(\hat{\mathbf{v}})|_{(k,k)}\|_F + \|\nabla^2 \mathcal{T}(\mathbf{v})|_{(\ell,\ell)} - \nabla^2 \mathcal{T}(\hat{\mathbf{v}})|_{(\ell,\ell)}\|_F. \end{aligned} \tag{152}$$

Noticing that the first four terms have the same upper bound, terms 5 and 6 have the same upper bound, and terms 7 through 10 share the same upper bound we obtain

$$\begin{aligned} \|\nabla^2 \mathcal{T}(\mathbf{v}) - \nabla^2 \mathcal{T}(\mathbf{v})\|_F \leq {}& 4\|\nabla^2 \mathcal{T}(\mathbf{v})|_{(i,k)} - \nabla^2 \mathcal{T}(\hat{\mathbf{v}})|_{(i,k)}\|_F \\ & + 2\|\nabla^2 \mathcal{T}(\mathbf{v})|_{(i,j)} - \nabla^2 \mathcal{T}(\hat{\mathbf{v}})|_{(i,j)}\|_F \\ & + 4\|\nabla^2 \mathcal{T}(\mathbf{v})|_{(i,i)} - \nabla^2 \mathcal{T}(\hat{\mathbf{v}})|_{(i,i)}\|_F. \end{aligned} \tag{153}$$

Now consider the first term

$$\|\nabla^2 \mathcal{T}(\mathbf{v})|_{(i,k)} - \nabla^2 \mathcal{T}(\hat{\mathbf{v}})|_{(i,k)}\|_F^2 = 4\Big( \sum_{a,b \in [r]} (v_{ja}v_{\ell b} - \hat{v}_{ja}\hat{v}_{\ell b})^2 \Big) \tag{154}$$

Let $\delta_j = \hat{v}_j - v_j$ and $\delta_\ell = \hat{v}_\ell - v_\ell$. We expand equation 154 to obtain

$$= 2\|v_j v_\ell^T - \hat{v}_j \hat{v}_\ell^T\|_F = 2\|v_j v_\ell^T - (v_j + \delta_j)(v_\ell + \delta_\ell)^T\|_F^2$$

$$= 2\|v_j v_\ell^T - (v_j v_\ell^T + \delta_j v_\ell^T + v_j \delta_j^T + \delta_j \delta_\ell^T)\|_F^2$$

$$= 2\|(\delta_j v_\ell^T + v_j \delta_\ell^T + \delta_j \delta_\ell^T)\|_F^2$$

$$\leq 16\big(\|\delta_j\|^2\|v_\ell\|^2 + \|v_j\|^2\|\delta_j\|^2 + \|\delta_j\|^2\|\delta_\ell\|^2\big) \leq 16B^2(\|\delta_j\|^2 + \|\delta_\ell\|^2),$$

where we apply Cauchy-Schwarz as needed. The second term is bounded similarly. The last term can be bounded

$$\|\nabla^2 \mathcal{T}(\mathbf{v})|_{(i,i)} - \nabla^2 \mathcal{T}(\hat{\mathbf{v}})|_{(i,i)}\|_F \leq 2\sqrt{\sum_{a \in [r]} (v_{ja}^2 - \hat{v}_{ja}^2)^2}.$$

Upper bounding $v_{ja} + \hat{v}_{ja} \leq 2B$ which applies for all $a \in [r]$ we obtain

$$= 2\sqrt{\sum_{a \in [r]} (v_{ja} - \hat{v}_{ja})^2 (v_{ja} + \hat{v}_{ja})^2} \leq 2\sqrt{B^2 \sum_{a \in [r]} (v_{ja} - \hat{v}_{ja})^2} \leq 2\sqrt{\|v_j - \hat{v}_j\|^2 B^2}.$$

Putting the terms together we obtain the following bound for

$$\|\nabla^2 \mathcal{T}(\mathbf{v}) - \nabla^2 \mathcal{T}(\mathbf{v})\|_F \leq$$
$$16B^2\Big[(\|\delta_j\|^2 + \|\delta_\ell\|^2) + (\|\delta_j\|^2 + \|\delta_k\|^2) + (\|\delta_i\|^2 + \|\delta_\ell\|^2) + (\|\delta_i\|^2 + \|\delta_\ell\|^2)$$
$$+ (\|\delta_i\|^2 + \|\delta_j\|^2) + (\|\delta_k\|^2 + \|\delta_\ell\|^2)$$
$$+ (\|\delta_i\|^2 + \|\delta_j\|^2 + \|\delta_k\|^2 + \|\delta_\ell\|^2)\Big]. \quad (155)$$

Now we can perform the analysis for the hessian of the entire Max-CSP loss

$$\mathcal{L}_\rho(\mathbf{v}) := \frac{1}{|\mathcal{P}|}\Bigg[\sum_{P_z \in \mathcal{P}} \sum_{s \subseteq z} y_s \frac{1}{|\mathcal{C}(s)|} \sum_{\substack{g,g' \subseteq s \\ \text{s.t } \zeta(g,g')=s}} \langle v_g, v_{g'}\rangle$$

$$+ \rho\Bigg[\sum_{P_z \in \mathcal{P}} \sum_{\substack{g,g',h,h' \subseteq z \\ \zeta(g,g')=\zeta(h,h')}} \big(\langle v_g, v_{g'}\rangle - \langle v_h, v_{h'}\rangle\big)^2$$

$$+ \sum_{v_s \in \mathbf{v}} (\|v_s\|^2 - 1)^2\Bigg]\Bigg]. \quad (156)$$

We break the loss into three terms

$$\mathcal{W}_1(\mathbf{v}) := \frac{1}{|\mathcal{P}|} \sum_{P_z \in \mathcal{P}} \sum_{s \subseteq z} y_s \frac{1}{|\mathcal{C}(s)|} \sum_{\substack{g,g' \subseteq s \\ \text{s.t } \zeta(g,g')=s}} \langle v_g, v_{g'}\rangle,$$

$$\mathcal{W}_2(\mathbf{v}) := \frac{\rho}{|\mathcal{P}|}\Bigg[\sum_{P_z \in \mathcal{P}} \sum_{\substack{g,g',h,h' \subseteq z \\ \zeta(g,g')=\zeta(h,h')}} \big(\langle v_g, v_{g'}\rangle - \langle v_h, v_{h'}\rangle\big)^2\Bigg],$$

$$\mathcal{W}_3(\mathbf{v}) := \frac{\rho}{|\mathcal{P}|} \sum_{v_s \in \mathbf{v}} (\|v_s\|^2 - 1)^2,$$

such that $\mathcal{L}_\rho(\mathbf{v}) := \mathcal{W}_1(\mathbf{v}) + \mathcal{W}_2(\mathbf{v}) + \mathcal{W}_3(\mathbf{v})$. We break the hessian apart into three terms

$$\|\nabla^2\mathcal{L}_\rho(\mathbf{v}) - \nabla^2\mathcal{L}_\rho(\hat{\mathbf{v}})\|_F \le \|\nabla^2\mathcal{W}_1(\mathbf{v}) - \nabla^2\mathcal{W}_1(\hat{\mathbf{v}})\|_F$$
$$+ \|\nabla^2\mathcal{W}_2(\mathbf{v}) - \nabla^2\mathcal{W}_2(\hat{\mathbf{v}})\| + \|\nabla^2\mathcal{W}_3(\mathbf{v}) - \nabla^2\mathcal{W}_3(\hat{\mathbf{v}})\|. \quad (157)$$

The first term is zero as its the hessian of a quadratic which is a constant. The difference is then zero. We bound the second term as follows.

$$\|\nabla^2\mathcal{W}_2(\mathbf{v}) - \nabla^2\mathcal{W}_2(\hat{\mathbf{v}})\| \le \frac{\rho}{|\mathcal{P}|}\Big\| \sum_{P_z\in\mathcal{P}} \sum_{\substack{g,g',h,h'\subseteq z \\ \zeta(g,g')=\zeta(h,h')}} \big(\nabla^2\mathcal{T}_{(g,g',h,h')}(\mathbf{v}) - \nabla^2\mathcal{T}_{(g,g',h,h')}(\hat{\mathbf{v}})\big)\Big\|_F$$

$$\le \frac{\rho}{|\mathcal{P}|}\sqrt{\sum_{P_z\in\mathcal{P}} \sum_{\substack{g,g',h,h'\subseteq z \\ \zeta(g,g')=\zeta(h,h')}} 64B^2\big(\|\delta_g\|^2 + \|\delta_{g'}\|^2 + \|\delta_h\|^2 + \|\delta_{h'}\|^2\big)}.$$

Noticing that each $\delta_g$ can be involved in no more than $|\mathcal{P}|$ sums by being variables in every single predicate.

$$\le \frac{8B\rho}{|\mathcal{P}|}\sqrt{|\mathcal{P}|\|V-\hat{V}\|_F^2} \le \frac{8B\rho}{\sqrt{|\mathcal{P}|}}\|V-\hat{V}\|_F.$$

Now we move on to bound $\mathcal{W}_3$

$$\|\mathcal{W}_3(\mathbf{v}) - \mathcal{W}_3(\hat{\mathbf{v}})\|_F = \frac{\rho}{|\mathcal{P}|}\Big\|\sum_{v_s\in\mathbf{v}} \big(\nabla^2(\|v_s\|^2-1)^2 - \nabla^2(\|\hat{v}_s\|^2-1)^2\big)\Big\|_F$$

$$\le \frac{\rho B}{|\mathcal{P}|}\sqrt{\sum_{v_s\in\mathbf{v}}\|v_s-\hat{v}_s\|^2} = \frac{\rho B}{|\mathcal{P}|}\|V-\hat{V}\|_F.$$

Putting all three terms together we obtain the smoothness of the hessian is dominated by $\mathcal{W}_2$.

$$\|\nabla^2\mathcal{L}_\rho(\mathbf{v}) - \nabla^2\mathcal{L}_\rho(\hat{\mathbf{v}})\|_F \le \Big(\frac{8B\rho}{\sqrt{|\mathcal{P}|}} + \frac{B\rho}{|\mathcal{P}|}\Big)\|V-\hat{V}\|_F \le \frac{8B\rho}{\sqrt{|\mathcal{P}|}}\|V-\hat{V}\|_F.$$

$\square$

## F.1  Miscellaneous Lemmas

**Theorem F.1** (perturbed-gd Jin et al. (2017))**.** *Let $f$ be $\ell$-smooth (that is, it's gradient is $\ell$-Lipschitz) and have a $\gamma$-Lipschitz Hessian. There exists an absolute constant $c_{max}$ such that for any $\delta \in (0,1), \epsilon \le \frac{\ell^2}{\gamma}, \Delta_f \ge f(X_0) - f^*$, and constant $c \le c_{max}$, $PGD(X_0, \ell, \gamma, \epsilon, c, \delta, \Delta_f)$ applied to the cost function $f$ outputs a $(\gamma^2, \epsilon)$ SOSP with probability at least $1 - \delta$ in*

$$O\left(\frac{(f(X_0) - f^*)\ell}{\epsilon^2}\log^4\left(\frac{d\ell\Delta_f}{\epsilon^2\delta}\right)\right)$$

*iterations.*

**Definition.** $[(\gamma, \epsilon)$-second order stationary point] A $(\gamma, \epsilon)$ second order stationary point of a function $f$ is a point $x$ satisfying

$$\|\nabla f(x)\| \le \epsilon$$
$$\lambda_{min}(\nabla^2 f(x)) \ge -\sqrt{\gamma\epsilon}.$$

**Theorem F.2.** *(Robustness Theorem 4.6 (Raghavendra & Steurer, 2009a) rephrased) Let $\mathbf{v}$ be a set of vectors satisfying the constraints of SDP 2 to additive error $\epsilon$ with objective $OBJ(\mathbf{v})$, then*

$$SDP(\Lambda) \ge OBJ(\mathbf{v}) - \sqrt{\epsilon}poly(kq).$$

**Corollary 2.** Given a Max-k-CSP instance $\Lambda$, there is an OptGNN$_{M,\Lambda}(\mathbf{v})$ with $T = O(\epsilon^{-4}\Phi^4\log(\delta^{-1}))$ layers, $r = O(\Phi)$ dimensional embeddings, with learnable parameters $M = \{M_1, ..., M_T\}$ that outputs a set of vectors $V$ satisfying the constraints of SDP 2 and approximating its objective, SDP$(\Lambda)$, to error $\epsilon$ with probability $1 - \delta$ over random noise injections.

*Proof.* The proof is by inspecting the definition of OptGNN in the context of Theorem 3.1. □

**Corollary 3.** The OptGNN of Corollary 2 , which by construction is equivalent to Algorithm 3, outputs a set of embeddings **v** such that the rounding of Raghavendra & Steurer (2009a) outputs an integral assignment $\mathcal{V}$ with a Max-k-CSP objective OBJ($\mathcal{V}$) satisfying $OBJ(\mathcal{V}) \geq S_\Lambda(\text{SDP}(\Lambda) - \epsilon) - \epsilon$ in time $\exp(\exp(\text{poly}(\frac{kq}{\epsilon})))$ which approximately dominates the Unique Games optimal approximation ratio.

*Proof.* The proof follows from the robustness theorem of Raghavendra & Steurer (2009a) which states that any solution to the SDP that satisfies the constraints approximately does not change the objective substantially Theorem F.2. □

# G  Definitions

In this section we introduce precise definitions for message passing, message passing GNN, and SDP's. First we begin with the SDP.

**Definition** (Standard Form SDP). An SDP instance $\Lambda$ is comprised of objective matrix $C \in \mathbb{R}^{N \times N}$ and constraint matrices $\{A_i\}_{i \in \mathcal{F}}$ and constants $\{b_i\}_{i \in \mathcal{F}}$ over a constraint set $\mathcal{F}$

$$\text{Minimize:} \quad \langle C, X \rangle \tag{158}$$
$$\text{Subject to:} \quad \langle A_i, X \rangle = b_i \quad \forall i \in [\mathcal{F}] \tag{159}$$
$$X \succeq 0. \tag{160}$$

For any standard form SDP, there is an equivalent vector form SDP with an identical optimum.

**Definition** (Vector Form SDP). Any standard form SDP $\Lambda$ is equivalent to the following vector form SDP.

$$\text{Minimize:} \quad \langle C, V^T V \rangle \tag{161}$$
$$\text{Subject to:} \quad \langle A_i, V^T V \rangle = b_i \quad \forall i \in [\mathcal{F}] \tag{162}$$
$$V = [v_1, v_2, ..., v_N] \in \mathbb{R}^{N \times N}. \tag{163}$$

For the SDP's corresponding to CSP's, there is a natural graph upon which the CSP instance is defined. Next we define what it means for a message passing algorithm on a graph to solve a SDP.

**Definition** (Message Passing). A $T$ iteration message passing algorithm denoted MP is a uniform circuit family that takes as input a graph $G = (V, E)$ and initial embeddings $U = \{u_i^0\}_{i \in [N]} \in \mathbb{R}^{r \times N}$ and outputs $\text{MP}(G, U^0) \in \mathbb{R}^{r \times N}$. The evaluation of MP involves the use of a uniform circuit family defined for each iteration UPDATE $= \{\text{UPDATE}_j\}_{j \in [T]}$. At iteration $\ell \in [T]$, for each node $i \in V$, the function $\text{UPDATE}_\ell : \mathbb{R}^{r \times (|N(i)|+1)} \to \mathbb{R}^r$ takes the embeddings of nodes $i$ and its neighbors at iteration $\ell - 1$, denoted $\{u_j^{\ell-1}\}_{j \in N(i) \cup i} \in \mathbb{R}^r$, and outputs embedding $u_i^\ell \in \mathbb{R}^r$.

We additionally require some mild restrictions on the UPDATE function to be polynomially smooth in its inputs and computable in polynomial time. That is for any $\|U\| \leq B$ in the $B$ norm ball we have

$$\text{UPDATE}_r(U - U') \leq poly(B, N) \|U - U'\|, \tag{164}$$

and the $\text{UPDATE}_\ell$ circuit is polynomial time computable $\text{poly}(|N(i)|, r)$.

We impose some mild restrictions on the form of the message passing algorithm to capture algorithms that could reasonably be executed by a Message Passing GNN. In practice the UPDATE circuit for OptGNN is a smooth cubic polynomial similar to linear attention and therefore of practical value. Next we define Message Passing GNN.

**Definition** (Message Passing GNN). A $T$ layer Message Passing GNN is a uniform circuit family that takes as input a graph $G = (V, E)$, a set of parameters $M := \{M^1, M^2, ..., M^T\}$, and initial embeddings $U^0 = \{u_i^0\}_{i \in V} \in \mathbb{R}^{r \times N}$ and outputs $\text{GNN}(G, M, U^0) \in \mathbb{R}^{r \times N}$. The GNN circuit is evaluated as follows. For each node $i \in V$, each layer $\ell \in [T]$, there is a uniform circuit family $\text{AGG}_\ell : \mathbb{R}^{|M^\ell|} \times \mathbb{R}^{r \times |N(i)|} \to \mathbb{R}^r$ that takes as input a set of parameters $M^\ell \in M$ and a set of

embeddings of node $i$ and its neighbors at layer $\ell - 1$ denoted $\{u_j^{\ell-1}\}_{j \in N(i) \cup i}$, and outputs an embedding $u_i^\ell$. This update equation is represented as follows

$$u_i^\ell := \text{AGG}(M^\ell, U^\ell) = \text{UPD}(u_i^{\ell-1}, \text{MSG}(\{u_j^{\ell-1}\}_{j \in N(i)})). \tag{165}$$

For some functions UPD and MSG parameterized by $M^\ell$. To capture meaningful models of GNNs, we require $\text{AGG}(M, U)$ to be polynomially lipschitz and computable in polynomial time. That is, for weights $\hat{M}$ and $M'$ and for inputs $\hat{U}$ and $U'$ all in the $B$ norm ball, we require

$$\|\text{AGG}(\hat{M}, \hat{U}) - \text{AGG}(M', U')\| \leq \text{poly}(B, N) \left( \|\hat{M} - M'\| + \|\hat{U} - U'\| \right), \tag{166}$$

and we require $\text{AGG}(\hat{M}, \hat{U})$ to be computable in $poly(|N(i)|, r)$ time.

In practice the AGG circuit for OptGNN is a smooth cubic polynomial similar to linear attention and therefore of practical value. In almost the same way, we define a message passing algorithm see Definition G.

**Definition** (Output of GNN solves SDP). We say a GNN $\epsilon$-approximately solves an SDP with optimum OPT if its output embeddings $U^\ell = \{u_i\}_{i \in [N]}$ approximately optimize the vector form SDP objective $\text{SDP}(\{u_i\}_{i \in [N]}) \geq \text{OPT} - \epsilon$ and approximately satisfy the vector form constraints $\left| \langle A_i, V^T V \rangle - b_i \right| \leq \epsilon$.

