# OpenReview forum: "Are Graph Neural Networks Optimal Approximation Algorithms?"
_NeurIPS.cc/2024/Conference — NeurIPS 2024 spotlight_

### Official Review · Reviewer_Ubhy · 2024-07-04

**Soundness:** 3
**Presentation:** 4
**Contribution:** 3
**Rating:** 7
**Confidence:** 2

**Summary:**

This paper draws analogy between the optimization of, e.g., Max-Cut and Max-SAT problems, and the message-passing algorithm, and accordingly constructs OptGNN to implement the message-passing algorithms towards the problems. The generated optimal solutions are shown with provable bounds. Finally, empirical studies show the effectiveness and efficiency of the proposed OptGNN.

**Strengths:**

1. The logic of the paper in writing/presentation are absolutely easy to read and understand, and the project in design are reasonable.
2. The proposal is presented with theoretical guarantees, i.e. the bounds of the optimal solutions are shown.
3. The proposed framework OptGNN is easy to implement and the empirical studies are consistent and demonstrate the effectiveness of the proposal.
4. The discussion of related work is extensive and satisfactory.

**Weaknesses:**

1. Take the example of Max-Cut problem as an example, how to show the optimization problem are the same in equation (2) and the message-passing diagram shown in equation (3)? Same for other optimization problems discussed in the paper.

**Questions:**

1. In Table 2, what's the usual setting for the clause? Since the ratio is relatively high. What's the relationship between the total computational complexity and this ratio?

**Limitations:**

Based on my review, this part is not well addressed by the authors. But since myself is not an expert in optimization approximation algorithms, please the authors/ACs kindly refer to the weaknesses and questions raised by pother reviewers.

---

> ### Author Rebuttal · Authors · 2024-08-05
>
> We thank the reviewer for their feedback. We are glad that the reviewer finds that the paper is well-presented and explained and that they value both the empirical and theoretical aspects of this work!
>
> Next, we address the weaknesses/questions in order:
>
> >Take the example of Max-Cut problem as an example, how to show the optimization problem are the same in equation (2) and the message-passing diagram shown in equation (3)? Same for other optimization problems discussed in the paper.
>
>
>  Let's take the Max-Cut problem as an example (equation 1). To update the embedding $v_i$ corresponding to node $i$ we take the gradient of the Lagrangian $\frac{\partial{\mathcal{L}(\mathbf{v})}}{\partial v_i}$.  The update will be as follows:
> $$ v_i' = v_i - \eta \frac{\partial{\mathcal{L}(\mathbf{v})}}{\partial v_i}.$$
> For the Max-Cut problem, we can enforce the constraint by normalizing the embeddings. So we just need derivatives with respect to the objective from equation 1. It is hopefully easy to see that computing  $\frac{\partial{\mathcal{L}(\mathbf{v})}}{\partial v_i}$  of the objective in equation 1 leads to the expression in the parenthesis in equation 2. The normalization for equation 2 enforces the constraint in equation 1. The central observation is that this kind of approach to minimizing the Lagrangian using gradient steps will lead to message-passing steps on the constraint graph for constraint satisfaction problems. In equation 3 we show a simple way to parameterize this message passing by adding learnable matrices.
>
> For the question:
> >In Table 2, what's the usual setting for the clause? Since the ratio is relatively high. What's the relationship between the total computational complexity and this ratio?
>
> The ratio is the clauses-to-variables ratio. This means that the higher the ratio the more constrained the instance will be, which will typically make it harder or potentially impossible to solve. More specifically, it is known that there is a phase transition point at a ratio of 4.26. Formulae with higher ratios than that become increasingly more likely to be unsatisfiable.  Formulae with ratios at this phase transition point are typically considered to be hard.
>
> >Limitations:
> Based on my review, this part is not well addressed by the authors. But since myself is not an expert in optimization approximation algorithms, please the authors/ACs kindly refer to the weaknesses and questions raised by pother reviewers.
>
>  The main limitation of this work has to do with its applicability to different combinatorial optimization problems. The theoretical connection to optimal approximability results only holds for Max-CSPs. Of course, our method can be applied to essentially any SDP/polynomial optimization problem in practice. The main practical challenge in that case is enforcing the feasibility of the solutions. For example, consider the Travelling Salesperson Problem (TSP). In that case, our model requires a way of enforcing that the solutions produced by the model are valid tours. That would have to be solved with a post-processing step. Covering such cases is no trivial task and would require additional arguments and empirical work to be done in a clean and mathematically coherent way. We believe this is certainly a promising avenue for future work.
>
> Again, we thank the reviewer for the comments and we will make sure to include an extensive discussion of the limitations in the final version of the paper. We would also like to encourage the reviewer to have a look at our responses to the other reviewers as well, in case that helps further clarify some of their concerns. We hope our response addresses the concerns of the reviewer! If the reviewer finds our answers satisfactory, we would be grateful if they could further increase their score.

---

> > ### Comment · Reviewer_Ubhy · 2024-08-11
> > **Appreciate the authors' responses**
> >
> > I believe my questions are well addressed by the response, and I am happy to increase my score to 7 accordingly.

---

> ### Author Response · Authors · 2024-08-09
> **minor remark**
>
> Regarding the question:
>
> >In Table 2, what's the usual setting for the clause? Since the ratio is relatively high. What's the relationship between the total computational complexity and this ratio?
>
> We wanted to mention that higher ratios means larger constraint graphs (because the number of clauses increases) which implies also larger memory costs. As we said in our response to reviewer WeYu, for a graph  with vertices $|V|$  and edges $|E|$ , for embedding dimension
> $d$, and depth $L$, the total runtime of OptGNN is $O(Ld^\omega |V| + Ld|E|)$,
>  where $\omega$
>  is the matrix multiplication constant.

---

### Official Review · Reviewer_857L · 2024-07-13

**Soundness:** 4
**Presentation:** 4
**Contribution:** 3
**Rating:** 7
**Confidence:** 4

**Summary:**

The papers propose graph neural architectures that can be used to capture optimal approximation algorithms for a large class of combinatorial optimization problems.

**Strengths:**

- The paper for the most part is well written with clear motivation.
- The contributions made in the paper are manifold across various optimization problems like Max-cut, Min-vertex cover etc. and have shown commendable performance.
- The paper also showcases a good theoretical basis.

**Weaknesses:**

No significant weakness as such.
Fig1. and Fig. 2 the visibility could be improved.

**Questions:**

NA.

**Limitations:**

Yes, the authors has provided limitations section.

---

> ### Author Rebuttal · Authors · 2024-08-04
>
> We thank the reviewer for the positive feedback and for appreciating both the empirical and theoretical elements of the paper's contributions.

---

### Official Review · Reviewer_zqcP · 2024-07-13

**Soundness:** 3
**Presentation:** 3
**Contribution:** 3
**Rating:** 7
**Confidence:** 3

**Summary:**

The paper establishes that polynomial-sized message-passing GNNs, can learn and replicate the optimal approximation capabilities of traditional algorithms based on SDP relaxations for Max-CSP under the assumption of the UGC.  The authors propose OptGNN, which effectively integrates the theoretical framework of SDP to produce high-quality solutions for combinatorial problems such as Max-Cut, Min-Vertex-Cover, and Max-3-SAT.

**Strengths:**

The paper explores integrating semidefinite programming into GNNs under the Unique Games Conjecture to optimally approximate combinatorial optimization problems. It introduces the OptGNN, a model designed to utilize polynomial-time message-passing algorithms and low-rank SDP relaxations to effectively solve problems such as Max-Cut and Max-3-SAT. The paper establishes OptGNN's potential through PAC learning arguments and empirical validation against traditional solvers and neural baselines. The paper is well-structured and mathematically sound.

**Weaknesses:**

- The paper lacks a comprehensive analysis of how well OptGNN scales with increasing graph sizes and complexity.
- Although the out-of-distribution generalization of OptGNN has been tested, an analysis of its generalization over problem size is missing.

**Questions:**

1. Could you clarify how the proposed approach scales with increasingly large problem instances, particularly regarding computational complexity and performance stability?
2. What are the generalization capabilities over problem scale of the proposed approach?
3. What are the limitations of OptGNN? please explain in the text.

**Limitations:**

Authors need to discuss the limitation of their work.

---

> ### Author Rebuttal · Authors · 2024-08-06
>
> We would like to thank the reviewer for the detailed feedback. We appreciate their positive comments on the soundness and the structure of the paper!
>
> To address the weaknesses:
>
> >The paper lacks a comprehensive analysis of how well OptGNN scales with increasing graph sizes and complexity.
>
> The cost of OptGNN scales linearly with the size of the graph (number of nodes/edges) as it is a message passing GNN. Empirically, we show that OptGNN runs on graphs of several thousands of nodes (GSET and 3-SAT experiments) without scalability issues.
>
> >Although the out-of-distribution generalization of OptGNN has been tested, an analysis of its generalization over problem size is missing.
>
> Our Max-Cut GSET results act also as both size and OOD generalization tests since the training was done on ER graphs of 500 nodes with a fixed edge density, and the testing was done on the GSET instances which can have tens of thousands of nodes and are not from the same ER distribution. The model performs competitively on up to an order of magnitude larger graphs than the ones it was trained on.
>
> >Could you clarify how the proposed approach scales with increasingly large problem instances, particularly regarding computational complexity and performance stability?
>
> Please see answers above and our response to reviewer WeYu.
>
> >What are the generalization capabilities over problem scale of the proposed approach?
>
> It is not quite clear what is meant by this question. Is this is about generalization to other problems, e.g., training on Max-Cut and testing on Vertex Cover? If that is what is meant by the reviewer,  OptGNN in that case will typically not be able to transfer its performance to different problems without further training. It is certainly an interesting question to explore whether pre-training on certain problems and fine-tuning on others can impact performance in a positive way.
>
> >What are the limitations of OptGNN? please explain in the text.
>
> Thank you for pointing this out! We will make sure to discuss limitations and scaling in more detail in the paper.  Briefly, the main two limitations are that the applicability of our theoretical result only covers CSPs and not all CO problems.  Empirically, there is also the issue of "rounding" a convex relaxation to a integral solution, which is typically undertaken by a branch and bound tree search (i.e for producing a valid tour in the TSP). Tackling this issue is an important avenue for future work and would require non-trivial additions to our framework.  Please also see our response to reviewer Ubhy for the same question.
>
>
> Again, we thank the reviewer for their comments. If this answer has addressed your concerns please consider raising your score. In any case, we will gladly respond to any further questions!

---

> > ### Comment · Reviewer_zqcP · 2024-08-12
> >
> > Thank you for the detailed response. Please update the paper in the camera-ready version with these explanations. Your response meets my expectations, and I have increased my score to 7.

---

### Official Review · Reviewer_WeYu · 2024-07-26

**Soundness:** 4
**Presentation:** 3
**Contribution:** 4
**Rating:** 8
**Confidence:** 4

**Summary:**

This work presents a significant advancement in the field of combinatorial optimization by developing a graph neural network (GNN) architecture named OptGNN. The authors demonstrate that OptGNN can capture optimal approximation algorithms for a broad class of combinatorial optimization problems, leveraging the power of semidefinite programming (SDP) and the Unique Games Conjecture (UGC). They prove that polynomial-sized message-passing GNNs can learn the most powerful polynomial-time algorithms for Max Constraint Satisfaction Problems, resulting in high-quality approximate solutions for challenges such as Max-Cut, Min-Vertex-Cover, and Max-3-SAT. Additionally, OptGNN provides a method for generating provable bounds on the optimal solution from the learned embeddings. The empirical results show that OptGNN outperforms classical heuristics, solvers, and state-of-the-art neural baselines across various datasets, making it a robust tool for efficient and optimal combinatorial optimization.

**Strengths:**

1. The paper is theoretically solid. It builds on well-established concepts in approximation algorithms and semidefinite programming, providing a strong theoretical basis for its claims.
2. The development of OptGNN, a graph neural network that can capture optimal message-passing algorithms for combinatorial optimization problems, seems to be a significant innovation. This architecture leverages message-passing updates use SDP relaxations to prove its high-quality approximate solutions.
3. The paper includes a variety of evaluations, such as out-of-distribution tests and ablation studies, which help validate the practical utility and robustness of OptGNN.

**Weaknesses:**

1. Although OptGNN shows strong performance on several benchmarks, it does not outperform all state-of-the-art methods. This indicates room for improvement in the model’s optimization and training processes.
2. The theoretical guarantees provided by OptGNN rely on the truth of the UGC, which, while widely believed, remains unproven. This dependence could limit the certainty of the results. That being said, I consider this more like a limitation than weakness.
3. The writing needs to be improved, there are multiple typos and inconsist notations.

**Questions:**

1. Can you elaborate on the scalability of OptGNN for very large graphs or more complex real-world problems? What are the practical limitations in terms of computational resources and time?
2. How adaptable is OptGNN to new combinatorial optimization problems that were not directly addressed in the paper? What modifications would be necessary to extend its applicability?
3. Can you also point out some future directions for this research? Both application and theoretical wise are good.

**Limitations:**

Yes.

---

> ### Author Rebuttal · Authors · 2024-08-06
>
> We thank the reviewer for all their detailed comments and review of our work! To address the
> concerns in order.
> 1. For a graph $G$ with vertices $V$ and edges $E$, for embedding dimension $d$, and depth $L$, the total runtime of OptGNN is $O(L d^\omega |V| + Ld|E|)$ where $\omega$ is the matrix multiplication constant $2.37$.  That is, the computational cost scales linearly in the size of the graph just as it does with message passing neural networks.  In practice, the max cut GSET instances serve as a scalability test.
> The GSET dataset is comprised of graphs with up to tens of thousands of nodes, whereas the OptGNN training set was comprised of graphs with up to 500 nodes.  We observe the model maintains strong performance for instances that are up to 3000 nodes and subsequently tapers off when the number of nodes substantially exceeds 3000.
> 2. The OptGNN construction can be performed for any polynomial optimization subject to polynomial constraints (polynomial system solving).  Any polynomial system, of which MaxCSP is a prominent example, admits a vector form SDP relaxation corresponding to the degree-2 Lasserre hierarchy relaxation. A prominent example of a problem that does not have a well known formulation as a polynomial optimization is the traveling salesman problem.
> 3. Promising avenues of research include constructing tighter bounds on optimality for the outputs of neural architectures/OptGNN.  Tighter bounds directly translate to superior performance for branch and bound tree searches that require a certificate of optimality for termination.  This would be a promising avenue for empirical investigation.
> 4. Weaknesses: Indeed, further empirical work is required to achieve state of the art across a wide variety of benchmarks.  With respect to the UGC, although its truth is yet undetermined, another way to think about it is from the perspective of algorithms.  OptGNN captures the algorithms with the best approximation ratios for Max CSP that are known in the literature.
>
> We once again thank the reviewer for their questions and time in reviewing our work, and hope our response addresses their concerns!

---

> ### Comment · Reviewer_WeYu · 2024-08-13
>
> I have read the authors response and I would like to keep my score.

---

### Decision · Program_Chairs · 2024-09-25

**Decision:**

Accept (spotlight)

**Comment:**

This work introduces OptGNN, a novel graph neural network (GNN) architecture designed to tackle combinatorial optimization problems. By integrating semidefinite programming (SDP) and the Unique Games Conjecture (UGC), OptGNN can learn optimal approximation algorithms for Max Constraint Satisfaction Problems like Max-Cut, Min-Vertex-Cover, and Max-3-SAT. The authors demonstrate that polynomial-sized message-passing GNNs can replicate the performance of traditional SDP-based algorithms, producing high-quality approximate solutions. Empirical results show that OptGNN outperforms classical heuristics, solvers, and state-of-the-art neural models, making it a powerful tool for combinatorial optimization.

All the reviewers recognized the significance of this work and provided positive feedback. The paper is technically solid and merits presentation at NeurIPS.